# Fast $k$-means Seeding Under The Manifold Hypothesis

**Poojan Shah** [1]   **Shashwat Agrawal** [1]   **Ragesh Jaiswal** [1]

## Abstract

We study *beyond worst case analysis* for the $k$-means problem where the goal is to model typical instances of $k$-means arising in practice. Existing theoretical approaches provide guarantees under certain assumptions on the optimal solutions to $k$-means, making them difficult to validate in practice. We adopt the *manifold hypothesis*, where data obtained in ambient dimension $D$ concentrates around a low dimensional manifold of intrinsic dimension $d \ll D$, as a reasonable assumption to model real world clustering instances. We identify key geometric properties of datasets which have theoretically predictable scaling laws depending on the *quantization exponent $\varepsilon = 2/d$* using techniques from optimum quantization theory. We show how to exploit these regularities to design a fast seeding method called Qkmeans[1] which provides $O(\varrho^{-2} \log k)$ approximate solutions to the $k$-means problem in time $O(nD \log k) + \widetilde{O}(\varrho^{-1} k^{1+\gamma})$; where the exponent $\gamma = \varepsilon + \varrho$ for an input parameter $\varrho < 1$. This allows us to obtain new runtime - quality trade-offs. We perform a large scale empirical study [2] across various domains to validate our theoretical predictions and algorithm performance to bridge theory and practice for beyond worst case data clustering.

## 1. Introduction

Data clustering is a fundamental task in machine learning and data science. The goal is to partition a dataset into subsets so that points within the same subset (or cluster) are more similar to each other than to points in other clusters. Such partitions help uncover latent structure and compress

data into a small number of interpretable groups. Among the many approaches to clustering, the $k$-means formulation is perhaps the most widely used. In this formulation, we are given a dataset $\mathcal{X} = \{x_1, \ldots, x_n\} \subset \mathbb{R}^D$, and our task is to find a set of $k$ centers $C = \{c_1, \ldots, c_k\} \subset \mathbb{R}^D$ which minimize the total within-cluster variance:

$$\texttt{cost}(\mathcal{X}, C) \overset{\text{def}}{=} \sum_{x \in \mathcal{X}} \min_{c \in C} \|x - c\|^2.$$

These centers induce a partition $\mathcal{X} = \bigcup_{j=1}^{k} C_j$, where $C_j$ contains the points in the dataset $\mathcal{X}$ closest to $c_j$. An optimal set $C_k^*$ achieves cost $\texttt{opt}_k(\mathcal{X}) = \texttt{cost}(\mathcal{X}, C_k^*)$. Optimizing the $k$-means cost then serves as a proxy for uncovering some natural clustering of the dataset. This formulation has become a key component of algorithm design in several fields such as vector quantization and compression (Linde et al., 1980), approximate nearest neighbor search (Jégou et al., 2011a; Johnson et al., 2017) along with numerous others.

Given the immense importance of $k$-means in modern data processing, it is natural that the problem has been thoroughly investigated from both theoretical and empirical perspectives. The $k$-means problem is known to be NP-hard (Dasgupta, 2008; Mahajan et al., 2009), so it is unlikely to have a polynomial time algorithm. Much research has been done on designing polynomial time *approximation schemes*[3] for the $k$-means problem. However, the algorithm most popularly used in practice[4] to solve $k$-means instances is a heuristic known as the $k$-means algorithm (*not to be confused with the $k$-means problem*). This heuristic, also known as Lloyd's iterations (Lloyd, 1982), iteratively improves the solution in several rounds. The heuristic starts with an arbitrarily chosen set of $k$ centers. In every iteration, it (i) partitions the points based on the nearest center and (ii) updates the center set to the centroids of the $k$ partitions. In the classical computational model, it is easy to see that every Lloyd iteration costs $O(nkD)$ time. This hill-climbing approach may get stuck at a local

---

[1]Department of Computer Science and Engineering, Indian Institute of Technology Delhi. Correspondence to: Poojan Shah <shahpoojan2004@gmail.com>.

*Proceedings of the $43^{rd}$ International Conference on Machine Learning*, Seoul, South Korea. PMLR 306, 2026. Copyright 2026 by the author(s).

[1]The Q stands for Quantization, as the analysis of our algorithm was motivated by results from optimum quantization theory.

[2]Our code is available at This Repository

[3]These were developed in a long line of research by (Kumar et al., 2010; Jaiswal et al., 2014; 2015; Cohen-Addad, 2018; Friggstad et al., 2019; Cohen-Addad et al., 2019; Bhattacharya et al., 2020)

[4]Ranked as the second most popular data mining algorithm in (Wu et al., 2008).

minimum or take a huge amount of time to converge, and hence, it does not give provable guarantees on the quality of the final solution or the running time (Dasgupta, 2003; Har-Peled & Sadri, 2005; Arthur & Vassilvitskii, 2006a;b). In practice, Lloyd's iterations are usually preceded by the $k$-means++ algorithm (Arthur & Vassilvitskii, 2007), a fast sampling-based approach for picking the initial $k$ centers that also gives an approximation guarantee. So, Lloyd's iterations, preceded by the $k$-means++ algorithm, give the best of both worlds, theory and practice. Hence, it is unsurprising that a lot of work has been done on these two algorithms. The work ranges from efficiency improvements in specific settings to implementations in distributed and parallel models.

**Beyond worst case analysis.** A large fraction of theoretical algorithms for $k$-means are designed to be used as *off the shelf* algorithms in a *data oblivious* manner. This means that they assume that the input dataset $\mathcal{X}$ can be any arbitrary set of points in $\mathbb{R}^D$ and provide worst case guarantees. The pervasive issue is that algorithms designed to resist pathological instances, i.e. inputs that maximize resource usage-often yield complexity bounds far removed from their typical empirical performance. The realization that real datasets rarely resemble these theoretical worst-case examples provides the core motivation for moving beyond pessimistic, data-oblivious constraints (Roughgarden, 2019; Balcan, 2021; Agarwal et al., 2015). This *data driven* paradigm acts as a necessary bridge between theoretical analysis and practical applicability and so has gained attention in the algorithm design community.

For $k$-means in particular, this comprises of several approaches. The performance of Lloyd-type methods and $k$-means++ under separability conditions was analyzed by (Ostrovsky et al., 2013; Jaiswal & Garg, 2012). An instance $(\mathcal{X}, k)$ of the $k$-means problem is said to be $\alpha$-separated if $\frac{\texttt{opt}_k(\mathcal{X})}{\texttt{opt}_{k-1}(\mathcal{X})} \leq \alpha$. This definition is motivated by the usual practice of using the *elbow method* to determine a good value of $k$ to be used for clustering $\mathcal{X}$. Another line of research studies the notion of *approximation stability* in (Balcan et al., 2009; Blum, 2021; Agarwal et al., 2015). Approximation stability is the idea that any clustering whose cost is close to optimal must be similar to the ground truth clustering. An instance $(\mathcal{X}, k)$ is called $(c, \alpha)$-approximation stable with respect to the cost function cost and a ground truth target clustering $C_T$ if every $c$ approximate solution to cost is atmost $\alpha$ far from $C_T$ in the sense of fraction of points misclustered. Other proposals include distributional stability (Bilu & Linial, 2010) , spectral stability (Kumar & Kannan, 2010) and perturbation resilience (Awasthi et al., 2010). (Cohen-Addad & Schwiegelshohn, 2017) studied the structural consequences of these notions. A common theme

across these proposals is assuming some properties of the optimal $k$-means clustering, which makes an empirical study of these notions quite difficult. Indeed, it is a non trivial problem to figure out how to check if a given dataset instance follows these assumptions. Although these notions have been studied theoretically, not much work has been done in studying whether they hold (even approximately) true for real world datasets, in part due to the fact that these assumptions bestow constraints on the optimal cost, which are computationally hard to verify.

---

***Motivating Questions*** *The above discussion naturally leads to the following questions:*

1. *Can we identify structural properties of real-world datasets that are easily verifiable, well motivated in theory and empirically hold true for diverse modern datasets ?*

2. *If so, can we exploit the existence of such structural properties to design better algorithms for $k$-means clustering?*

---

**Contributions.** In order to tackle these questions, we present the following contributions, which are discussed in detail in the next section :

1. We adopt the manifold hypothesis - a well-established and empirically validated model for high-dimensional data - as a structural assumption for beyond-worst-case analysis of $k$-means.

2. Under this framework, we identify geometrical parameters which have predictable scaling law behaviour. We use this structure to design a fast seeding procedure through analyzing a *perturbed* version of $k$-means++ sampling.

3. We perform a large scale empirical study to validate our assumptions and the performance of our seeding algorithm.

### 1.1. Our Contributions

We propose the idea that the *manifold hypothesis* is a good candidate to derive such structural properties. The manifold hypothesis states that although real-world data often reside in high-dimensional ambient spaces, their probability mass tends to concentrate near low-dimensional, smooth submanifolds $\mathcal{M} \subset \mathbb{R}^D$ with intrinsic dimension $d \ll D$. This hypothesis forms the conceptual bedrock for explaining the success of modern deep-learning systems (Bengio et al., 2013) and has subsequently attracted thorough theoretical and experimental investigation (Fefferman et al., 2016; Brown et al., 2023; Bordt et al., 2023) and hence is widely accepted and established. We formally state this:

**Assumption 1.** *(Manifold hypothesis assumption) The dataset $\mathcal{X} = \{x_1, \ldots, x_n\} \subset \mathbb{R}^D$ is generated by drawing $n$ independent and identically distributed samples from a density function $f : \mathcal{M} \to \mathbb{R}_+$ whose support $\mathrm{supp}(f)$ is a measurable subset of a smooth, compact $d$-dimensional submanifold $\mathcal{M} \subset \mathbb{R}^D$.*

From the point of view of beyond–worst-case analysis, the manifold hypothesis offers a different type of structural regularity: instead of assuming properties about the *optimal solution*, it assumes geometric regularity of the *input distribution*. This makes it attractive for algorithm design, as the assumption is intrinsic to the data rather than to the algorithm's output.

Moreover, it is compatible with the increasing body of empirical evidence that high-dimensional machine learning data has low intrinsic dimension (Tempczyk et al., 2022; Horvat & Pfister, 2022). Finally, from a practical standpoint, the manifold hypothesis is more aligned with modern applications than separation- or stability-based assumptions. While the latter are tailored to idealized clustering scenarios, the manifold hypothesis is already the implicit structural model underlying essentially all deep learning architectures used in vision, NLP, physics, and generative tasks (Bengio et al., 2013; Loaiza-Ganem et al., 2024)

What does this imply for the $k$-means problem ? A fundamental question is to study how the optimal $k$ means cost decreases with the number of clusters $k$. This question was explored for Euclidean spaces in an early work of (Zador, 1982), who showed asymptotic laws for the optimal *quantization cost* given by $\Delta_k(f) = \inf_{|C|=k} \Delta(f, C)$ where $\Delta(f, C) = \int_{\mathcal{M}} \min_{c \in C} \|x - c\|^2 f(x) dx$. This led to the development of the field now known as *optimum quantization theory*, for a detailed exposition of which we point the readers to (Gray & Neuhoff, 2006; Graf & Luschgy, 2000). Zador's results were generalized to manifolds by (Gruber, 2004), which are of interest to us :

**Fact 2.** *(Gruber, 2004) Suppose $f$ is a probability density according to Assumption 1 and $S_k = \{s_1, \ldots, s_k\} \subset \mathbb{R}^D$ be an optimal $k$-quantizer of $f$ i.e., $S_k \in \arg\min_{|C|=k} \Delta(f, C)$.*
*Define the parameters $\beta_k(f)$ and $\eta_k(f)$ as follows :*

$$\beta_k(f) \stackrel{\text{def}}{=} \frac{\Delta_1(f)}{\Delta_k(f)} \qquad \eta_k(f) \stackrel{\text{def}}{=} \frac{\max_{i \neq j \in [k]} \|s_i - s_j\|}{\min_{i \neq j \in [k]} \|s_i - s_j\|}$$

*Then, the following holds :*

$$\beta_k(f) \in O_f(1) \cdot k^\varepsilon \qquad \eta_k(f) \in O_f(1) \cdot k^{\varepsilon/2}$$

*where $\varepsilon = 2/d$ is referred to as the quantization exponent and $O_f(\cdot)$ hides constants depending only on $f$.*

**Remark 3.** *We mention that such scaling laws are prevalent in several research areas. Notable examples include the*

scaling law between word frequency and word rank in large text documents, referred to as Zipf's law (Zipf, 1949) and the power law distribution of node degrees in large scale networks (Barabasi & Albert, 1999). These are frequently used to guide algorithm design and algorithm analysis.

We now define these parameters for finite datasets :

**Definition 4.** *For a dataset $\mathcal{X} = \{x_1, \ldots, x_n\}$ and a set of centers $C = \{c_1, \cdots, c_k\}$ we define the following:*

$$\beta(\mathcal{X}, C) := \frac{\mathrm{cost}(\mathcal{X}, \mu)}{\mathrm{cost}(\mathcal{X}, C)} \quad \beta_k(\mathcal{X}) := \frac{\mathrm{opt}_1(\mathcal{X})}{\mathrm{opt}_k(\mathcal{X})}$$

$$\eta(\mathcal{X}, C) := \frac{\max_{i \neq j} \|c_i - c_j\|}{\min_{i \neq j} \|c_i - c_j\|} \quad \eta(\mathcal{X}) := \frac{\max_{i \neq j} \|x_i - x_j\|}{\min_{i \neq j} \|x_i - x_j\|}$$

*Here, $\mu = \mu(\mathcal{X})$ is the centroid of the dataset $\mathcal{X}$.*

Note that we always have $\beta(\mathcal{X}, C) \leq \beta_k(\mathcal{X})$ and $\eta(\mathcal{X}, C) \leq \eta(\mathcal{X})$ for all $C = \{c_1, \ldots, c_k\}$, as long as the centers belong to the dataset itself i.e.,$C \subset \mathcal{X}$. We show the following finite sample bounds :

**Theorem 5.** *(Scaling laws) Suppose $\mathcal{X}$ is sampled according to Assumption 1 then the following hold with probability at least $1 - \frac{1}{\mathrm{poly}(n)}$:*

$$\beta_k(\mathcal{X}) \in \left(1 + O_f\left(\sqrt{\frac{kD \log n}{n}}\right)\right) \cdot k^\varepsilon$$

$$\eta(\mathcal{X}) \in O_f(1) \cdot n^{3\varepsilon/2}$$

Next, we design a fast rejection-sampling based seeding scheme to accelerate $k$-means++ with novel runtime-approximation tradeoffs. Our algorithms are designed in a way which allows us to exploit the scaling laws. This allows us to improve upon existing approaches aimed at speeding up $k$-means++ under the manifold hypothesis. We formally state our main results as follows :

**Theorem 6.** *(Main Theorem) Let $\mathcal{X} = \{x_1, \ldots, x_n\} \subset \mathbb{R}^D$ be a dataset. Let $\beta_k(\mathcal{X})$ and $\eta(\mathcal{X})$ be the data dependent parameters introduced in Definition 4. Then, there exists an algorithm which additionally takes as input parameters $\varrho < 1, m \in \mathbb{N}$ and outputs a set $C$ of $k$ centers such that*

$$\mathbf{E}[\mathrm{cost}(\mathcal{X}, C)] \in$$

$$O(\varrho^{-2}) \cdot \left( \underbrace{\log k \cdot \mathrm{opt}_k(\mathcal{X})}_{k\text{-means++ guarantee}} + \underbrace{k^{-\varrho m / \beta_k(\mathcal{X})} \cdot \mathrm{opt}_1(\mathcal{X})}_{exponentially\ decaying\ additive\ term} \right)$$

*using a one-time preprocessing time of $O(nD \log k)$, with each seeding taking time $\widetilde{O}(m(k \log \eta(\mathcal{X}))^{1+\varrho} \log \beta_k(\mathcal{X}))$.*

The above theorem holds true for *all* datasets without any assumptions and hence the bounds are in terms of the data dependent parameters $\beta = \beta_k(\mathcal{X})$ and $\eta = \eta(\mathcal{X})$. We can see that for fixed $\eta, \beta$, the runtime consists of an $O(nD \log k)$

term, which is required just for reading the input $\mathcal{X}$, with a $\log k$ overhead for dimensionality reduction as part of the preprocessing, and an additional term which is *almost* linear in $k$. The approximation guarantee consists of the standard $O(\log k)$ guarantee along with an additive, scale invariant and exponentially decaying term depending on the variance of the dataset. Due to the strong decaying of the additive term, it can be removed as follows :

**Corollary 7.** *(Rejection sampling for $k$-means++) Let $\mathcal{X} = \{x_1, \ldots, x_n\} \subset \mathbb{R}^D$ be a dataset. Then there exists an algorithm which additionally takes as input a parameter $\varrho < 1$ and outputs a set $C$ of $k$ centers such that*

$$\mathbf{E}[\mathtt{cost}(\mathcal{X}, C)] \in O(\varrho^{-2} \log k)\mathtt{opt}_k(\mathcal{X})$$

*using a one-time preprocessing time of $O(nD \log k)$, with each seeding taking expected time $\widetilde{O}\left(\varrho^{-1}\beta_k(\mathcal{X})(k \log \eta(\mathcal{X}))^{1+\varrho} \log \beta_k(\mathcal{X})\right)$, and $\widetilde{O}(\cdot)$ hides $\log k, \log n$ factors*

Finally, we can combine the scaling laws from Theorem 5 along with Theorem 6 to obtain the following result for $k$-means clustering under the manifold hypothesis:

**Corollary 8.** *(Rejection sampling under manifold hypothesis) Let $\mathcal{X} = \{x_1, \ldots, x_n\} \subset \mathbb{R}^D$ be a dataset generated according to Assumption 1. Then there exists an algorithm which additionally takes as input a parameter $\varrho < 1$ and outputs a set $C$ of $k$ centers such that*

$$\mathbf{E}[\mathtt{cost}(\mathcal{X}, C)] \in O(\varrho^{-2} \log k)\mathtt{opt}_k(\mathcal{X})$$

*using a one-time preprocessing time of $O(nD \log k)$, with each seeding taking expected time $\widetilde{O}\left(\varrho^{-1}k^{1+\gamma}\right)$, where $\gamma = \varepsilon + \varrho$, and $\widetilde{O}(\cdot)$ hides $\log k, \log n$ factors*

**Remark 9** (Estimating the data-dependent parameters). *The parameters $\beta_k(\mathcal{X})$ and $\eta(\mathcal{X})$ appearing in Theorem 6 and Corollaries 7 -8 are in general, not known before the execution of the algorithm. We address this issue by discussing how to estimate them prior to execution.*

1. *The parameters can be upper bounded by the dataset aspect ratio $\eta(\mathcal{X})$, $\beta_k(\mathcal{X}) \leq \eta(\mathcal{X})^2$. The aspect ratio can be approximated in $O(n \log n)$ time (Shamos & Hoey, 1975), providing worst-case bounds that hold for any dataset without any assumptions. However, our experiments show that these are often too pessimistic.*

2. *Under Assumption 1, the parameters exhibit predictable scaling behavior. In particular, one can estimate the intrinsic dimension $d$ using the maximum likelihood estimator of (Levina & Bickel, 2005), and obtain an estimate of the quantization exponent $\varepsilon = 2/d$. This yields $\beta_k(\mathcal{X}) \lesssim k^{2/d}$, which is significantly tighter and more informative than the worst-case bound. As shown in Section 3, these estimates are stable across datasets and domains.*

3. *A practical alternative is to compute k-means costs for a small set of values $k \in O(1)$, fit the scaling laws, and extrapolate to larger $k$. Our experiments show that the resulting fits are highly consistent (typically $R^2 \geq 0.95$).*

*Finally, given any estimate of $\beta_k(\mathcal{X})$ obtained via the above approaches, one can set $m = O(\beta_k(\mathcal{X}))$ to ensure a negligible contribution from the additive term in Theorem 6.*

**Empirical results.** We perform extensive empirical analysis to test whether our theoretical assumptions are reasonable for practical datasets. Our study captures cross-domain datasets covering image data, image and text embeddings, as well as tabular data captured from sensors and scientific experiments. Our assumptions can be easily checked by computing the cost for various values of $k$, then evaluating the geometric properties of the resulting clusters. We find that the scaling laws strongly manifest in these datasets. We also analyze the effect of intrinsic dimension on these laws using independent ID estimates. Finally, we observe that our seeding method gives about an order of magnitude speedup for $k$-means++ seeding on high dimensional datasets when compared to prior work.

### 1.2. Comparison with prior work

**Speeding up k-means++.** Let us discuss the methods proposed to accelerate $k$-means++ so as to position our results in the literature. Directly implementing $k$-means++ provides an $O(\log k)$ approximation guarantee in $O(nkD)$ time, which quickly becomes prohibitively slow even for moderate sized datasets. The problem is that $k$-means++ is inherently a sequential procedure where the new center chosen depends on the previously chosen centers. $k$-means++ has also been extended to distributed settings (Bahmani et al., 2012; Bachem et al., 2017a; 2018), but our results are for clustering on a single node.

**MCMC methods.** In the line of work (Bachem et al., 2016b;a), the authors propose to approximate the $k$-means++ distribution using an appropriately designed markov chain, whose limiting distribution is equal to the $D^2$ distribution. Both of these algorithms have runtime of $\Omega(k^2)$, thus making them slow for use-cases when a dataset is required to be partitioned into thousands of clusters [5]. In comparison, Qkmeans does not suffer from this $\Omega(k^2)$ dependence, but has $O(k^{1+\gamma})$ dependence where the exponent $\gamma := \varepsilon + \varrho$. As we shall see, $\varepsilon \sim 0.1 - 0.2$ in real large scale datasets. Moreover, the MCMC methods often have an additive term which decays as $O(1/m)$ where $m$ is the markov chain length, while we obtain an exponential $O(k^{-\varrho m/\beta})$ decay in our results.

---

[5]This was empirically demonstrated in Section 6 of (Cohen-Addad et al., 2020a).

**ANNS techniques.** In (Cohen-Addad et al., 2020a), the authors provide a speedup for $k$-means++ using a variety of algorithmic techniques which include hierarchical tree embeddings, approximate nearest neighbor search and rejection sampling. However, constructing this embedding brings up dependence on the aspect ratio of the dataset $\Delta$, which in general may be unbounded in terms of the input size and requires special pre-processing. They then use rejection sampling to convert samples from the $D^2$ distribution on the tree metric to a sample in the euclidean metric, which introduces an additional factor of $O(D^2)$. In comparison, our algorithm does not require such metric embeddings and instead depends on simply sampling from an $\ell_2$-norm distribution which can be computed in a lightweight pre-processing step. Moreover, we have better dependence on the ambient dimension $O(D)$ vs $O(D^3)$ and prove improved approximation guarantee $O(\varrho^{-2})$ vs $O(\varrho^{-3})$. We also mention several works which try to speed up $k$-means++ or Lloyd's iterations by maintaining some nearest neighbor data structure (Pelleg & Moore, 1999; Kanungo et al., 2002; Elkan, 2003; Hamerly, 2010; Wang et al., 2012; Ding et al., 2015; Bottesch et al., 2016; Newling & Fleuret, 2016; Curtin, 2017), but do not provide any theoretical guarantee on the solution quality or runtime for their proposed methods.

**Dimensionality reduction.** In (Charikar et al., 2023), the authors propose to project the dataset onto a randomly sampled unit vector and then perform $k$-means++ in 1D in $O(n \log n)$ time. They show that when this 1D solution is lifted back to $D$ dimensions, it provides an $O(k^4 \log k)$ approximation guarantee. Although this is extremely fast, they report clustering costs 10-100$\times$ larger than those provided by $k$-means++. They also propose a coreset-based method which theoretically runs in $O(nD + n \log n) + O(k^5 D)$ time to provide $O(\log k)$ approximation. In comparison, our method has a much lower dependence on $k$. There is a line of research (see (Makarychev et al., 2019) and the references therein) on dimensionality reduction for $k$-means which show that the ambient dimension can be reduced to $O(\log k)$ while preserving the $k$-means cost.

**Coresets.** Coresets are a way to compress the dataset by computing a weighted subset of the data. We refer the reader to (Bachem et al., 2017b; Feldman, 2020) for becoming familiar with coresets, which have now become ubiquitous tools for $k$-means clustering. This long line of research culminated in (Draganov et al., 2024), where a coreset construction in near-linear time $\tilde{O}(nD \log \log \Delta)$ of size $O(k/\varepsilon^4)$ is proposed which can then be used in conjunction with standard $k$-means++ to obtain an $O(nD \log \log \Delta + k^2 D)$ algorithm having $O(\log k)$ guarantee. We conclude by providing a summary of this discussion in Table 1. We mention that our results provide new theoretical tradeoffs as well as insights into the structure of input data for $k$-means clustering, along with providing a method to exploit them.

---

**Algorithm 1** PREPROCESS($\mathcal{X}$)

1. For $\varepsilon_{\mathrm{JL}} \in (0, 1/4)$, apply the JL-like dimensionality reduction from (Makarychev et al., 2019) in $O(nD \log k)$ time to obtain $D \in O(\log(k/\varepsilon_{\mathrm{JL}})/\varepsilon_{\mathrm{JL}}^2)$
2. Compute the mean $\mu(\mathcal{X}) := \frac{1}{n} \sum_i x_i$ of the dataset
3. Center each $x_i \leftarrow x_i - \mu(\mathcal{X}), 1 \le i \le n$
4. Compute $\|\mathcal{X}\|_{\mathrm{F}}^2 := \sum_i \|x_i\|^2$
5. Compute $\kappa(x_i) \leftarrow \|x_i\|^2 / \|\mathcal{X}\|_{\mathrm{F}}^2, 1 \le i \le n$

---

**Algorithm 2** SAMPLE($\mathscr{L}, C, m$)

1. Initialize $iter \leftarrow 0$ and sample $s \sim$ UNIFORM($\mathcal{X}$)
2. **while** $iter < m \ln k$ **do**
   - $iter \leftarrow iter + 1$
   - Sample $x \sim \kappa(\cdot \mid C)$, where
   $$\kappa(x \mid C) := \frac{\|x\|^2 + \|c_1\|^2}{\|\mathcal{X}\|_{\mathrm{F}}^2 + n\|c_1\|^2}$$
   - Sample $R \sim$ UNIFORM($0, 1$)
   - Compute $r \leftarrow \frac{\texttt{cost}(x, \texttt{Query}(\mathscr{L}, x))}{2\varrho^{-1}\left(\|x\|^2 + \|c_1\|^2\right)}$
   - **if** $R \le r$ **then** $s \leftarrow x$ and **break**
3. **Return** $s$

---

## 2. Seeding Algorithm

At a high level, our algorithm works by *simulating* the $D^2$ sampling step of $k$-means++ using rejection sampling. The idea is simple: instead of sampling directly from the $D^2$ distribution - which is expensive to update after every new center - we sample from a much easier distribution and correct the bias using a rejection step. The efficiency of this correction is controlled by a single quantity: the max Rényi divergence between the two distributions.

Suppose $\mu$ is a target distribution over a finite domain $\mathcal{X}$, and $\nu$ is a proposal distribution from which sampling is easy. Rejection sampling allows us to draw exact samples from $\mu$ using only samples from $\nu$, provided that $\mu$ is not too "peaked" relative to $\nu$.

**Definition 10.** *Let $\mu$ and $\nu$ be probability distributions supported on $\mathcal{X}$. The max Rényi divergence of $\mu$ from $\nu$ is defined as*

$$D_\infty(\mu \| \nu) \overset{\text{def}}{=} \sup_{x \in \mathcal{X}} \log \frac{\mu(x)}{\nu(x)}.$$

Intuitively, $\exp(D_\infty(\mu \| \nu))$ measures the worst-case mismatch between $\mu$ and $\nu$, and directly determines how many proposal samples we need before one is accepted.

**Lemma 11** (Rejection sampling guarantee). *Let $\mu$ and $\nu$ be distributions over a finite set $\mathcal{X}$ with $D_\infty(\mu \| \nu) < \infty$, and*

*Table 1.* Comparison of time complexity and approximation guarantees for accelerating $k$-means++ seeding. For our method, we empirically show $\beta \sim k^{\varepsilon}$ and $\log \eta \sim \varepsilon \log k$, where $\varepsilon$ is the *quantization exponent*. In the table, $\Delta$ is the aspect ratio of the dataset and $\beta, \eta$ are the parameters from Theorem 5. Note that the relation $\eta \leq \Delta$ always holds true. Moreover, under the manifold hypothesis the overall exponent $\gamma$ is defined as $\gamma = \varepsilon + \varrho$.

| APPROACH | TIME COMPLEXITY | APPROXIMATION GUARANTEE |
|---|---|---|
| *Prior work* | | |
| (Bachem et al., 2016a) | $O(nD) + O\big(\varepsilon^{-1} k^2 D \log k \, \varepsilon^{-1}\big)$ | $O(\log k)\, \mathtt{opt}_k + \varepsilon \, \mathtt{opt}_1$ |
| (Bachem et al., 2018) | $O(nD) + O\big(\varepsilon^{-2} k^2 D^2 \log k\big)$ | $O(\log k)\, \mathtt{opt}_k + \varepsilon \, \mathtt{opt}_1$ |
| (Cohen-Addad et al., 2020a) | $O(n(D + \log n) \log(\Delta D))$ $+ O\big(\varrho^{-1} k D^3 \log \Delta (n \log \Delta)^{O(\varrho)}\big)$ | $O\big(\varrho^{-3} \log k\big)\, \mathtt{opt}_k$ |
| (Charikar et al., 2023) | $O(nD) + O(n \log n)$ | $O\big(k^4 \log k\big)\, \mathtt{opt}_k$ |
| (Charikar et al., 2023) | $O(nD) + O(n \log n)$ $+ O(\varepsilon^{-2} k^5 D \log^2 k)$ | $O(\log k)\, \mathtt{opt}_k$ |
| (Draganov et al., 2024) | $O(nD \log \log \eta) + O(\varepsilon^{-4} k^2 D)$ | $O(\log k)\, \mathtt{opt}_k$ |
| (Shah & Jaiswal, 2025) | $O(nD) + O(\Delta^2 k^2 D)$ | $O(\log k)\, \mathtt{opt}_k$ |
| *Our results* | | |
| Theorem 6 | $O(nD) + O\big(mD \log k (k \log \eta)^{1+\varrho}\big)$ | $O(\varrho^{-2})\big(\log k \, \mathtt{opt}_k + k^{-\Omega(m/\beta)} \mathtt{opt}_1\big)$ |
| Corollary 7 | $O(nD) + O\big(\varrho^{-1} \beta D (k \log \eta)^{1+\varrho}\big)$ | $O\big(\varrho^{-2} \log k\big)\, \mathtt{opt}_k$ |
| Corollary 8 | $O(nD) + O\big(\varrho^{-1} (k \log n)^{1+\gamma} D\big)$ | $O\big(\varrho^{-2} \log k\big)\, \mathtt{opt}_k$ |

---

**Algorithm 3** QKMEANS$(\mathcal{X}, k, m)$

1. PREPROCESS$(\mathcal{X})$
2. Initialize the data structure INIT$(\mathcal{L}, \varrho)$
3. Sample $c_1 \sim$ UNIFORM$(\mathcal{X})$
4. Set $C \leftarrow \{c_1\}$, insert $c_1$ into $\mathcal{L}$, and set $C_1 \leftarrow \{c_1\}$
5. **for** $t = 2$ **to** $k$ **do**
   - Sample $c_t \leftarrow$ SAMPLE$(\mathcal{L}, C, m)$
   - Insert $c_t$ into $\mathcal{L}$
   - Update $C_t \leftarrow C_{t-1} \cup \{c_t\}$
6. **Return** $C_k$

---

*set $M = \exp(D_\infty(\mu \| \nu))$. Then:*

1. *Conditioned on acceptance, Algorithm 4 outputs an exact sample from $\mu$.*

2. *Each iteration succeeds with probability at least $1/M$.*

3. *Running the algorithm for $M \ln k$ iterations succeeds with probability at least $1 - 1/k$.*

*Proof.* In one iteration, the probability of accepting a point $x$ is $\nu(x) \cdot \mu(x)/(M\nu(x)) = \mu(x)/M$. Thus, conditioned on acceptance, the output distribution is exactly $\mu$. The total acceptance probability is $1/M$, and repeating for $M \ln k$ iterations fails with probability at most $(1 - 1/M)^{M \ln k} \leq 1/k$. $\square$

For performing $k$-means++ seeding, we are interested in sampling from the $D^2$ distribution given by $\pi(x|C) =$

---

**Algorithm 4** REJECTSAMPLE$(\mu, \nu, M)$

1. **for** $i = 1$ **to** $M \ln k$ **do**
   (a) Sample $x \sim \nu$
   (b) Sample $R \sim$ UNIFORM$(0, 1)$
   (c) **if** $R \leq \frac{\mu(x)}{M\nu(x)}$ **then**
      i. **return** $x$
2. **Fail**

---

$\mathtt{cost}(x, C)/\mathtt{cost}(\mathcal{X}, C)$ for each $x \in \mathcal{X}$, which is our target. What remains to be done is to select a suitable proposal distribution, which we now define as follows :

**Definition 12.** *Let $c_1$ be the first center which is chosen uniformly at random from $\mathcal{X}$ by Algorithm 3. The distribution $\kappa(\cdot|C)$ is defined as follows :*

$$\kappa(x|C) = \frac{\|x\|^2 + \|c_1\|^2}{\|\mathcal{X}\|_F^2 + n\|c_1\|^2}$$

**Remark 13.** *The motivation behind selecting this particular distribution comes from a line of work in the theoretical computer science community called "dequantized quantum machine learning" which often uses similar $\ell^2$ norm distributions over the data. More recently, this distribution was introduced in the context of clustering algorithms by works such as (Cornelissen et al., 2025; Shah & Jaiswal, 2025).*

Using the Cauchy-Schwarz inequality, we can provide an upper bound on the max-Rényi divergence as follows :

**Lemma 14.** *Let $\pi(\cdot|C)$ be the target $D^2$ distribution and $\kappa(\cdot|C)$ be the proposal distribution defined in Definition 12. Then the following holds :*

$$\mathbf{E}[D_\infty\big(\pi(\cdot|C)\|\kappa(\cdot|C)\big)] \leq \log\left(4 \cdot \frac{\mathtt{opt}_1(\mathcal{X})}{\mathtt{opt}_k(\mathcal{X})}\right)$$

Notice that the expression inside the $\log(\cdot)$ is exactly the $\beta_k(\mathcal{X})$ parameter which we introduced earlier. Under Assumption 1 we can use Theorem 5 to obtain the following:

**Lemma 15.** *Suppose the dataset $\mathcal{X}$ is generated according to Assumption 1, then the following holds :*

$$\mathbf{E}[D_\infty\big(\pi(\cdot|C)\|\kappa(\cdot|C)\big)] \in O\left(\frac{\log k}{d}\right)$$

Thus we can perform $k$-means seeding by first sampling a uniformly random point as a centre, and subsequently calling Algorithm 4 $k-1$ times with $\mu, \nu$ being the distributions $\pi(\cdot|C), \kappa(\cdot|C)$ respectively, where $C$ is the current set of centers. In order to employ the rejection sampling idea, the probability of accepting a sampled point $x \in \mathcal{X}$ needs to be proportional to $\mathtt{cost}(x, C)^2$. However, computing this quantity takes $\Theta(kD)$ time, which contributes $\Theta(k^2 D)$ time on performing this sampling $k$ times. To mitigate this quadratic dependence on $k$, we apply another optimization technique; we use an approximate nearest neighbors (ANNS) data structure to approximate the distance between $x$ and the closest center. We use one such ANNS data structure developed by (Cohen-Addad et al., 2020a) which is based on locality sensitive hashing (details in Appendix C.2). This data structure is specified by the following properties:

**Lemma 16.** *(LSH-based ANNS data structure) For any set of points $P = \{p_1, \ldots, p_t\} \subset \mathbb{R}^D$ and parameter $\varrho < 1$, there exists a data structure $\mathscr{L}$ with operations* Insert *and* Query *such that with probability at least $1 - 1/t$, the following guarantees hold* [6]

1. Insert$(\mathscr{L}, p)$ *inserts the point $p \in P$ into $\mathscr{L}$ in time $O\left(D \log \eta (t \log \eta)^\varrho\right)$.*

2. Query$(\mathscr{L}, p)$ *returns a point such that $\|p - \mathtt{Query}(\mathscr{L}, p)\| \leq \frac{1}{\sqrt{\varrho}} \cdot \min_{q \in \mathscr{L}} \|p - q\|$ and the query time is $O\left(D \log \eta (t \log \eta)^\varrho\right)$.*

3. $\mathscr{L}$ *is monotone under insertions: the distance between $p$ and* Query$(\mathscr{L}, p)$ *is non-increasing after inserting more points.*

Here, $\eta$ is the aspect ratio of the set $P$ defined as $\eta(P) = \frac{\max_{x \neq y \in P} \|x - y\|}{\min_{x \neq y \in P} \|x - y\|}$.

We take one more step. We are passing a parameter $m \in \mathbb{N}$ in our input that is used to give a threshold on the time complexity of our seeding algorithm. We use this parameter to control the number of iterations in Rejection Sampling; if

we do not output a sample after $m \ln k$ iterations of sampling from $\kappa(\cdot|C)$, we terminate and instead output a sample from the uniform distribution on $\mathcal{X}$. With these add-ons, our idea for $D^2$ sampling via rejection sampling is given in Algorithm 2.

It is important to observe that both these add-ons to our original rejection sampling strategy of Algorithm 4 distort the output distribution from $D^2$. As a consequence, we no longer have the $O(\log k)$ approximation guarantee readily available to us. We instead have an approximation guarantee which is dependent on $m$ and $\varrho$ as a part of Theorem 6. We give a detailed proof of the new approximation guarantee in Appendix C.3; to model the new distribution of cluster centers, we analyze an abstract version of $k$-means++ which we call $(\varrho, \delta)$-$k$-means++, where instead of sampling from the standard $D^2$ distribution $\pi(\cdot|C)$, we instead have to sample from a perturbed distribution defined as follows .

**Definition 17** (($(\varrho, \delta)$ perturbed $D^2$ distribution). *Let $\mathcal{X} = \{x_1, \ldots, x_n\} \subset \mathbb{R}^d$ be a dataset and $S \subset X$ be a set of centers sampled from $X$. Suppose that the data structure $\mathscr{L}$ with parameter $\varrho < 1$ is successful and each $c \in S$ is inserted into the data structure $\mathscr{L}$. Then for each $x \in \mathcal{X}$, we define the $(\varrho, \delta)$ perturbed $D^2$ distribution as :*

$$\pi_{(\varrho,\delta)}(x|S) = (1 - \delta) \cdot \pi^{\mathscr{L}}(x|S) + \delta \cdot \frac{1}{n}$$

Our technical contribution is to analyze the seeding quality of our algorithm under this distribution. This requires a careful potential based analysis to control the effect of the parameters $(\varrho, \delta)$ without introducing any large constant factors. We emphasize that this is a non trivial task since it is not known whether perturbing the $D^2$ distribution by a small amount translates to a similar guarantee in the solution cost or not (Grunau et al., 2023). We leverage the specific structure of the perturbation to show the following.

**Theorem 18** (Approximation guarantee for $(\varrho, \delta)$-$k$-means++). *Let $\mathcal{X} \subset \mathbb{R}^D$ be any dataset which is to be partitioned into $k$ clusters. Let $C$ be the set of centers returned by Algorithm 6 for any $\delta \in (0, 0.5)$ and $\varrho < 1$. The expected cost $\mathbf{E}[\mathtt{cost}(\mathcal{X}, C)]$ equals:*

$$O(\varrho^{-2} \log k)\mathtt{opt}_k(\mathcal{X}) + \delta \cdot O(k + \varrho^{-2} \log k)\mathtt{opt}_1(\mathcal{X})$$

If we let $m = \infty$, we can analyze the expected time of our algorithm and the approximation guarantee only depends on $\varrho$, which is given in Corollary 7. Finally, by plugging in the manifold hypothesis assumption, we obtain Corollary 8.

## 3. Empirical Study

In this section, we empirically evaluate the extent to which the theoretical predictions developed in this paper are reflected in real-world data. The complete details are in Appendix D.

*Compute environment.* All experiments were conducted on a dual-socket machine equipped with 2.20 GHz

| DATASET | $n$ | $d$ |
|---|---|---|
| MNIST | 60,000 | 784 |
| FMNIST | 60,000 | 784 |
| CIFAR10 | 50,000 | 3,072 |
| CIFAR100 | 50,000 | 3,072 |
| MNIST (CLIP) | 60,000 | 512 |
| FMNIST (CLIP) | 60,000 | 512 |
| CIFAR10 (CLIP) | 50,000 | 512 |
| CIFAR100 (CLIP) | 50,000 | 512 |
| REDDIT | 100,000 | 384 |
| STACKEXCHANGE | 20,000 | 384 |
| SUSY | 5,000,000 | 18 |
| HAR | 10,299 | 561 |

*Table 2.* Datasets used in experiments with sample size $n$ and ambient dimension $d$.

IntelXeonGold5220R running Linux on x86-64 architecture.

*Datasets.* We include popular image datasets, both as raw pixels as well as image embeddings which are produced through joint vision-language models such as CLIP-ViT-B/32 (Radford et al., 2021); text datasets in the form of text embeddings generated by the All-MiniLM-L6-v2 model (Wang et al., 2020); tabular datasets which record sensor data as well as data generated by scientific experiments. The details of the datasets used are presented in Table 2.

**Scaling Laws.** We test the hypothesis that the data-dependent parameters $\beta = \beta(\mathcal{X}, C)$ and $\eta = \eta(\mathcal{X}, C)$ grow polynomially with the number of clusters $k$. For each dataset, we compute $k$-means clusterings for $k \in \{5, 10, 50, 100, 250, 500, 750, 1000\}$ using 10 independent runs of standard $k$-means++ followed by Lloyd's iterations. We fit linear regressions to $\log \beta$ and $\log \eta$ versus $\log k$ and report slopes, $R^2$, and 95% confidence intervals. Figure 1 shows that across datasets and modalities, both quantities exhibit reasonable linear behavior in log–log coordinates. It is important to note that although we are not able to provide bounds on $\eta$ in term of $k$, the experiments show that $\log \eta$ scales atmost linearly with $\log k$. The estimated quantization exponent $\widehat{\varepsilon}$ is the slope of the best fit line for the $\log \beta$ vs $\log k$ plot.

**Intrinsic dimension.** We test the hypothesis that the empirically estimated quantization exponent $\widehat{\varepsilon}$ is inversely proportional to the intrinsic dimension. Independent intrinsic dimension estimates $\widehat{d}_{\text{MLE}}$ are obtained by using the popular maximum likelihood method (MLE) (Levina & Bickel, 2005). For a dataset $\mathcal{X} = \{x_1, \ldots, x_n\}$, let $T_j(x_i)$ denote the distance of the $j$th nearest neighbor of $x_i$ from $x_i$. For a

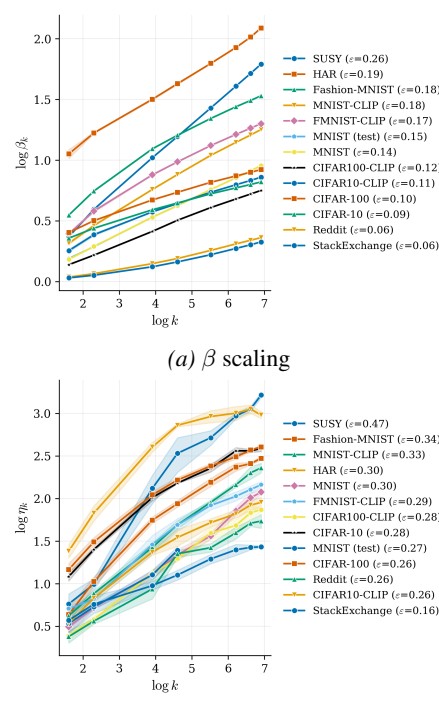

*(a) $\beta$ scaling*

*(b) $\eta$ scaling*

*Figure 1.* Scaling behavior across datasets

hyperparameter $k$, the ID estimate is calculated as :

$$\widehat{d}_{\text{MLE}}(\mathcal{X}, k) := \frac{1}{n} \sum_{i=1}^{n} \left( \frac{1}{k-1} \sum_{j=1}^{k-1} \log \frac{T_k(x_i)}{T_j(x_i)} \right)^{-1}$$

For each dataset we take 10 independent samples each of size $n' = 10,000$ to estimate the ID for a particular value of $k$. This is repeated for $k \in \{5, 10, 20, 50, 100\}$ and the average is reported. We plot the estimates $\widehat{\varepsilon}$ of the quantization exponent against $\widehat{d}_{\text{MLE}}$ in Figure 2.

**Seeding performance.** We compare our seeding algorithm Qkmeans with AFKMC$^2$ (Bachem et al., 2016a), RejectionSampling (Cohen-Addad et al., 2020a), PRONECoreset (Charikar et al., 2023) and FastKMeans (Draganov et al., 2024). All the algorithms were implemented in C++ without using any dimensionality reduction to ensure a fair comparison. Across a wide range of datasets and data modalities, the experiments show that Qkmeans consistently provides the fastest seeding procedure in practically relevant regimes, particularly as the number of clusters $k$ and the ambient dimension $D$ increase. RejectionSampling runtimes are almost independent of $k$, but it along with FastKMeans suffers a significant overhead due to constructing the multi-tree embeddings. AFKMC$^2$ is competitive only in low-dimensional tabular settings, but

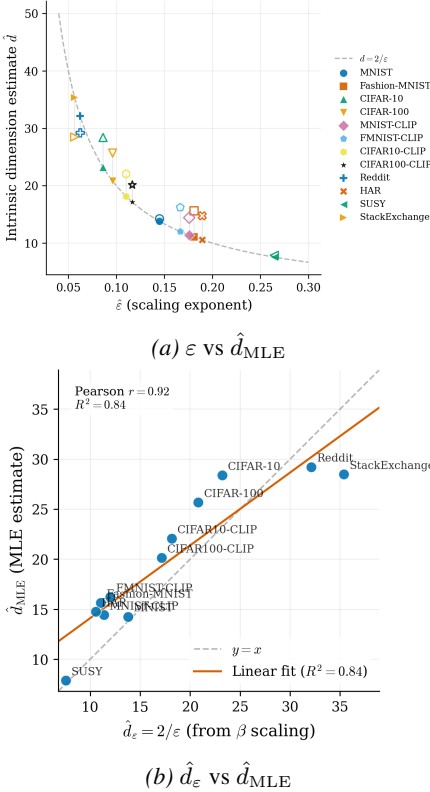

*(a) $\varepsilon$ vs $\hat{d}_{\mathrm{MLE}}$*

*(b) $\hat{d}_\varepsilon$ vs $\hat{d}_{\mathrm{MLE}}$*

*Figure 2.* Dependence of quantization exponent on ID

loses its advantage on image, text, and embedding data. From the detailed benchmarks in Appendix D, `Qkmeans` achieves consistent speedups of $5$–$10\times$ over `AFKMC`$^2$ and `PRONECoreset` and up to two orders of magnitude over `RejectionSampling`, while maintaining comparable seeding cost. Additional experiments on the effect of noisy manifolds and studying the rejection rate vs $m$ are presented in Appendix D.

## 4. Conclusion, Limitations, Future Directions

In this work, we modelled typical $k$-means clustering instances from a manifold hypothesis perspective. Using techniques from optimum quantization theory, we identified key geometric properties of the dataset that follow predictable scaling laws. We also showed how to exploit these regularities to design a fast rejection sampling-based seeding algorithm with new tradeoffs. Our theoretical results were backed by extensive empirical evaluations. We feel that due to its broad impact on the machine learning community, studying algorithms under the manifold hypothesis is an appealing and exciting direction for future research.

## Impact Statement

This paper presents work whose goal is to advance the field of machine learning. There are many potential societal consequences of our work, none of which we feel must be

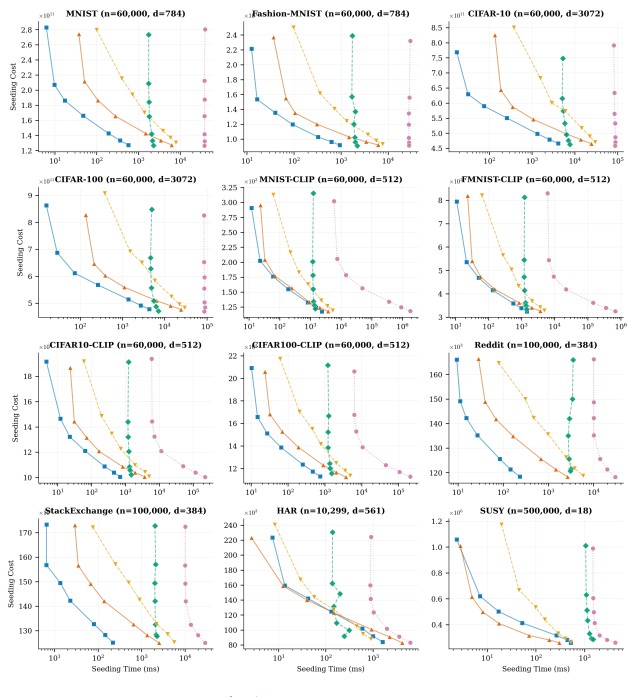

*Figure 3.* Seeding quality - runtime tradeoff plots

specifically highlighted here.

## Acknowldegements

RJ would like to thank IIT Delhi - Abu Dhabi for hosting him during the later stages of this work. PS would like to acknowledge that this work was partially supported by the CSE Research Acceleration Fund of IIT Delhi.

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

# Appendix

The appendices expand upon the content of the main paper. In Appendix A, we provide a theoretical proof of the scaling laws presented in Theorem 5. In Appendix C we provide the technical proofs for our seeding algorithm along with the analysis of its runtime and approximation guarantees. Finally, in Appendix D we provide the details of aour empirical study along with additional experiments and data.

## A. Theoretical Analysis of Scaling Laws

In this section we provide theoretical proofs for the scaling laws presented in Theorem 5. We mention that the techniques used here are well developed in optimum quantization theory (Zador, 1982; Gray & Neuhoff, 2006; Graf & Luschgy, 2000; Gruber, 2004) and we provide an elementary proof for the sake of being self-contained. Since we work under the manifold assumption, we require definitions and techniques from differential geometry for our proofs. We refer the reader to the book by (Lee, 2019) for obtaining familiarity with such notions. Recall that we are given a dataset $\mathcal{X} = \{x_1, \ldots, x_n\}$ where the $x_i$'s are sampled i.i.d. from a probability density function $f : \mathcal{M} \to \mathbb{R}_+$ supported on a compact, connected and smooth subset of a manifold $\mathcal{M} \subset \mathbb{R}^D$. The proof can be broken down into the following two steps :

- *Step 1* Showing that the optimum distortion $\Delta_k(f) = \inf_{|C|=k} \int_{\mathcal{M}} \min_{c \in C} \|x - c\|^2 f(x) \, \mathrm{dvol}_{\mathcal{M}}(x)$ follows the scaling law $\Delta_k(f) \in \Theta_f(k^{-2/d})$.

- *Step 2* Showing how to translate this in the finite sample case by studying how $\frac{1}{n}\mathtt{opt}_k(\mathcal{X}) \to \Delta_k(f)$ as $n \to \infty$.

**Preliminaries.** A function $\varphi : A \to B$ between two metric spaces $A$ and $B$ is called a *homeomorphism* if $\varphi$ is bijective and both $\varphi$ and $\varphi^{-1}$ are continuous. A set $\mathcal{M} \subset \mathbb{R}^D$ is called a smooth $d$-dimensional manifold embedded in $\mathbb{R}^D$ if for each point $p \in \mathcal{M}$ there exists an open neighborhood $W \subset \mathbb{R}^D$ of $p$ and a smooth diffeomorphism $\varphi : W \to \mathbb{R}^D$ such that $\varphi(\mathcal{M} \cap W) = \mathbb{R}^d \times \{0\}^{D-d}$. Equivalently, $\mathcal{M}$ admits a collection of local coordinate maps as follows. A *chart* on $\mathcal{M}$ is a pair $(U, \varphi)$, where $U \subset \mathcal{M}$ is open and $\varphi : U \to V \subset \mathbb{R}^d$ is a homeomorphism onto an open subset of $\mathbb{R}^d$. A collection of charts $\mathcal{A} = \{(U_\alpha, \varphi_\alpha)\}_{\alpha \in I}$ is called an *atlas* if $\mathcal{M} = \bigcup_{\alpha \in I} U_\alpha$. The atlas is said to be *smooth* if for every pair of overlapping charts $(U_\alpha, \varphi_\alpha)$ and $(U_\beta, \varphi_\beta)$, the transition maps $\varphi_\beta \circ \varphi_\alpha^{-1} : \varphi_\alpha(U_\alpha \cap U_\beta) \to \varphi_\beta(U_\alpha \cap U_\beta)$ are smooth diffeomorphisms. Throughout, we assume that $\mathcal{M}$ is equipped with the smooth atlas induced by its embedding in $\mathbb{R}^D$. This atlas endows $\mathcal{M}$ with the structure of a smooth manifold and allows us to perform local analysis by identifying sufficiently small neighborhoods of $\mathcal{M}$ with open subsets of $\mathbb{R}^d$.

**Lemma 19** (Zador bounds for manifolds). *Let $\mathcal{M} \subset \mathbb{R}^D$ be a compact, $d$-dimensional $C^2$ submanifold and let $J \subset \mathcal{M}$ be compact, measurable, and satisfy $\mathrm{vol}_{\mathcal{M}}(J) > 0$. Let $f : J \to R_+$ be a probability density function on $\mathcal{M}$ such that $f_{min} < f(x) < f_{max}$ and $\int_{\mathcal{M}} f(x) \, \mathrm{dvol}_{\mathcal{M}}(x) = 1$. Define*

$$\Delta_k(f) := \inf_{\substack{S \subset \mathbb{R}^D \\ |S|=k}} \int_J \min_{p \in S} \|x - p\|^2 \, f(x) \, \mathrm{dvol}_{\mathcal{M}}(x).$$

*Then there exist constants $0 < c < C < \infty$ depending only on $\mathcal{M}, J$ and $f$ such that*

$$c \, k^{-2/d} \leq \Delta_k(f) \leq C \, k^{-2/d} \quad \text{for all } k \geq 1.$$

*Proof. Upper bound.* Since $J$ is compact and $\mathcal{M}$ is a smooth embedded submanifold, there exists a finite atlas $\{(U_\ell, \varphi_\ell)\}_{\ell=1}^m$ such that

$$J = \bigcup_{\ell=1}^m J_\ell, \qquad J_\ell \subset U_\ell,$$

where each $\varphi_\ell : U_\ell \to V_\ell \subset \mathbb{R}^d$ is a diffeomorphism. By compactness and smoothness, shrinking charts if necessary, there exists a constant $C_0 \geq 1$ such that for all $\ell$ and all $x, y \in U_\ell$,

$$C_0^{-1}\|\varphi_\ell(x) - \varphi_\ell(y)\| \leq \|x - y\| \leq C_0\|\varphi_\ell(x) - \varphi_\ell(y)\|, \tag{1}$$

and the induced volume form satisfies

$$C_0^{-1} \, du \leq \mathrm{dvol}_{\mathcal{M}}(x) \leq C_0 \, du \quad \text{under } u = \varphi_\ell(x). \tag{2}$$

Fix $k \geq m$ and set $k_\ell := \lfloor k/m \rfloor$. The set $\varphi_\ell(J_\ell) \subset \mathbb{R}^d$ is compact, hence contained in a cube of side length $L_\ell < \infty$. Partition this cube into $\lceil k_\ell^{1/d} \rceil^d$ congruent subcubes, each of side length at most $2L_\ell k_\ell^{-1/d}$. Selecting one point from each subcube intersecting $\varphi_\ell(J_\ell)$ yields a set $\{u_{\ell,i}\}_{i=1}^{k_\ell} \subset \varphi_\ell(J_\ell)$ such that

$$\varphi_\ell(J_\ell) \subset \bigcup_{i=1}^{k_\ell} B_{\mathbb{R}^d}(u_{\ell,i}, Ck^{-1/d}),$$

for a constant $C$ depending only on $J$ and $d$.

Define $p_{\ell,i} := \varphi_\ell^{-1}(u_{\ell,i}) \in J_\ell$ and let

$$S := \bigcup_{\ell=1}^{m} \{p_{\ell,1}, \ldots, p_{\ell,k_\ell}\},$$

so that $|S| \leq k$. By (1), for every $x \in J$,

$$\min_{p \in S} \|x - p\| \leq C'k^{-1/d},$$

and hence

$$\min_{p \in S} \|x - p\|^2 \leq C''k^{-2/d}.$$

Integrating over $J$ and using the finiteness of $\mathrm{vol}_{\mathcal{M}}(J)$ gives

$$\int_J \min_{p \in S} \|x - p\|^2 f(x) \, d\mathrm{vol}_{\mathcal{M}}(x) \leq Ck^{-2/d},$$

which implies $\Delta_k(f) \leq Ck^{-2/d}$.

*Lower bound.* Since $\mathrm{vol}_{\mathcal{M}}(J) > 0$ and $\mathcal{M}$ is smooth, there exists a point $x_0 \in J$ and constants $\varrho > 0$ and $c_0 > 0$ such that the geodesic ball $B_{\mathcal{M}}(x_0, r) \subset J$ for all $0 < r \leq \varrho$, and

$$\mathrm{vol}_{\mathcal{M}}(B_{\mathcal{M}}(x_0, r)) \geq c_0 r^d. \tag{3}$$

Let $S \subset \mathbb{R}^D$ with $|S| = k$ be arbitrary and let $r > 0$. For each $p \in S$, the intersection $B(p, r) \cap \mathcal{M}$ is contained in a geodesic ball of radius $Cr$, hence by (3),

$$\mathrm{vol}_{\mathcal{M}}(B(p, r) \cap \mathcal{M}) \leq Cr^d.$$

Therefore,

$$\mathrm{vol}_{\mathcal{M}}\left(\bigcup_{p \in S} B(p, r) \cap \mathcal{M}\right) \leq Ckr^d.$$

Choosing $r = ck^{-1/d}$ with $c > 0$ sufficiently small ensures that

$$\mathrm{vol}_{\mathcal{M}}\left(\bigcup_{p \in S} B(p, r) \cap \mathcal{M}\right) < \mathrm{vol}_{\mathcal{M}}(B_{\mathcal{M}}(x_0, \varrho)).$$

Hence there exists a measurable subset

$$A \subset B_{\mathcal{M}}(x_0, \varrho)$$

of positive volume such that $\|x - p\| \geq r$ for all $x \in A$ and all $p \in S$. For such $x$,

$$\min_{p \in S} \|x - p\|^2 \geq r^2 = c^2 k^{-2/d}.$$

Integrating over $A$ yields

$$\int_J \min_{p \in S} \|x - p\|^2 f(x) \, d\mathrm{vol}_{\mathcal{M}}(x) \geq c'k^{-2/d},$$

where $c' > 0$ depends only on $\mathcal{M}$ and $J$. Since $S$ was arbitrary, this implies $\Delta_k(f) \geq ck^{-2/d}$.

Combining the upper and lower bounds completes the proof. $\qquad\square$

Next, we show finite sample bounds on the $k$-means cost to quantify how $\text{opt}_k(\mathcal{X}) \to \Delta_k(f)$ as the number of samples $n \to \infty$. By re-scaling, we can assume without loss of generality that $\|x\| \le 1$ for each $x \in J$. For this, we would require two standard results :

**Lemma 20** (Hoeffding's Inequality). *Let $X_1, \ldots, X_m$ be independent random variables such that $a_j \le X_j \le b_j$ almost surely for each $j \in [m]$. Let $S = \sum_{j=1}^m X_j$ then :*

$$\mathbf{Pr}[|S_m - \mathbf{E}[S_m]| \ge t] \le 2 \exp\left(-\frac{2t^2}{\sum_{j=1}^m (b_j - a_j)^2}\right)$$

**Lemma 21** (Covering Lemma). *Let $A \subset \mathbb{R}^m$ be a bounded set. Then for any $\delta > 0$, there exists a finite set $\mathcal{N}_\delta(A) \subset A$, called a $\delta$-net for $A$, such that:*

1. *For every $x \in A$, there exists $x' \in \mathcal{N}_\delta(A)$ with $\|x - x'\| \le \delta$.*

2. *$|\mathcal{N}_\delta(A)| \le \left(\frac{\text{diam}(A)}{\delta} + 1\right)^m$.*

Using these, we can show the following :

**Lemma 22** (Finite sample bound on $k$-means cost). *For any $\delta \in (0, 1)$, the following holds with probability atleast $1 - \delta$ :*

$$\left|\frac{1}{n}\text{opt}_k(\mathcal{X}) - \Delta_k(f)\right| \in O_f\left(\sqrt{\frac{kD \log n + \log(1/\delta)}{n}}\right)$$

*Proof.* Let us define the set of possible centers $\mathcal{C}_k := \{(c_1, \ldots, c_k) \subset \mathbb{R}^D : \|c_j\| \le 1, 1 \le j \le k\}$. For a fixed set of centers $C = (c_1, \ldots, c_k) \in \mathcal{C}_k$, let us define the following quantities:

$$\ell(x|C) := \min_{1 \le j \le k} \|x - c_j\|^2$$

$$L_n(C) := \frac{1}{n}\sum_{i \le i \le n} \ell(x_i|C)$$

$$L(C) := \mathbf{E}_{x \sim f}\ell(x|C)$$

It can be easily seen that the function $\ell(x|C)$ is bounded since:

$$\ell(x|C) = \min_{1 \le j \le k}\|x - c_j\|^2 \le \|x - c_1\|^2 \le 2(\|x\|^2 + \|c_1\|^2) \le 4$$

Notice that by definition, $L_n(C)$ is the average of $n$ i.i.d. random variables $\ell(x_i|C)$ each of which is bounded in $[0, 4]$. Hence, applying Lemma 20 implies that for any $t > 0$,

$$\mathbf{Pr}\big(|L_n(C) - L(C)| \ge t\big) \le 2\exp(-nt^2/8)$$

We next control the dependence of the loss on the centers. Let $C = (c_1, \ldots, c_k)$ and $C' = (c'_1, \ldots, c'_k)$ be two elements of $\mathcal{C}_k$ such that

$$\max_{j \in [k]} \|c_j - c'_j\| \le \delta.$$

For any $x \in \mathbb{R}^D$, we have

$$\left|\ell(x \mid C) - \ell(x \mid C')\right| = \left|\min_{1 \le j \le k}\|x - c_j\|^2 - \min_{1 \le j \le k}\|x - c'_j\|^2\right|$$
$$\le \max_{1 \le j \le k}\left|\|x - c_j\|^2 - \|x - c'_j\|^2\right|.$$

For a fixed $j$, we may expand

$$\left|\|x - c_j\|^2 - \|x - c'_j\|^2\right| = \left|\langle c'_j - c_j, 2x - c_j - c'_j\rangle\right|$$
$$\le \|c_j - c'_j\| \cdot \|2x - c_j - c'_j\|.$$

Since $\|x\| \leq 1$ and $\|c_j\|, \|c_j'\| \leq 1$, it follows that

$$\|2x - c_j - c_j'\| \leq 2\|x\| + \|c_j\| + \|c_j'\| \leq 4.$$

Consequently,

$$\left|\ell(x \mid C) - \ell(x \mid C')\right| \leq 4 \max_{j \in [k]} \|c_j - c_j'\| \leq 4\delta.$$

Averaging over the samples and taking expectation yields

$$|L_n(C) - L_n(C')| \leq 4\delta, \qquad |L(C) - L(C')| \leq 4\delta.$$

Let $\varepsilon > 0$ be a parameter to be chosen later. By Lemma 21, there exists an $\varepsilon$-net $\mathcal{N}_\varepsilon \subset \mathcal{C}_k$ such that

$$|\mathcal{N}_\varepsilon| \leq \left(\frac{2}{\varepsilon} + 1\right)^{kD} \leq \left(\frac{3}{\varepsilon}\right)^{kD}.$$

For any $C \in \mathcal{C}_k$, let $\Pi(C) \in \mathcal{N}_\varepsilon$ denote a nearest net point. Then

$$|L_n(C) - L(C)| \leq |L_n(\Pi(C)) - L(\Pi(C))| + 8\varepsilon.$$

Applying Hoeffding's inequality and a union bound over $\mathcal{N}_\varepsilon$, we obtain for any $t > 0$,

$$\mathbf{Pr}\left(\sup_{C' \in \mathcal{N}_\varepsilon} |L_n(C') - L(C')| \geq t\right) \leq 2|\mathcal{N}_\varepsilon| \exp(-nt^2/8).$$

Thus, with probability at least $1 - \delta$,

$$\sup_{C' \in \mathcal{N}_\varepsilon} |L_n(C') - L(C')| \lesssim \sqrt{\frac{kD \log(1/\varepsilon) + \log(1/\delta)}{n}}.$$

Combining the above bounds, we conclude that with probability at least $1 - \delta$,

$$\sup_{C \in \mathcal{C}_k} |L_n(C) - L(C)| \lesssim \sqrt{\frac{kD \log(1/\varepsilon) + \log(1/\delta)}{n}} + \varepsilon.$$

Choosing $\varepsilon = 1/n$ gives $\log(1/\varepsilon) = \log n$, and hence

$$\sup_{C \in \mathcal{C}_k} |L_n(C) - L(C)| \in O\left(\sqrt{\frac{kD \log n + \log(1/\delta)}{n}}\right).$$

Finally, observing that

$$\frac{1}{n}\mathsf{opt}_k(\mathcal{X}) = \inf_{C \in \mathcal{C}_k} L_n(C), \qquad \Delta_k(f) = \inf_{C \in \mathcal{C}_k} L(C),$$

we obtain

$$\left|\frac{1}{n}\mathsf{opt}_k(\mathcal{X}) - \Delta_k(f)\right| \leq \sup_{C \in \mathcal{C}_k} |L_n(C) - L(C)|,$$

which completes the proof. $\qquad\square$

We are now ready to provide a proof for Theorem 5.

**Theorem 23** (Scaling laws). *Suppose $\mathcal{X} = \{x_1, \ldots, x_n\} \subset \mathbb{R}^D$ consists of i.i.d. samples drawn from a distribution $f$ supported on a compact, smooth $d$-dimensional manifold $\mathcal{M} \subset \mathbb{R}^D$, with density bounded above and below on its support. Let $\varepsilon := 2/d$. Then, with probability at least $1 - \frac{1}{\mathrm{poly}(n)}$, the following hold:*

$$\beta_k(\mathcal{X}) = \left(1 + O_f\left(\sqrt{\frac{kD \log n}{n}}\right)\right) k^\varepsilon, \qquad \eta(\mathcal{X}) \in O_f(1)\, n^{3\varepsilon/2}.$$

*Proof.* We prove the two statements separately.

*Analysis of $\beta_k(\mathcal{X})$.* By definition,

$$\beta_k(\mathcal{X}) = \frac{\mathsf{opt}_1(\mathcal{X})}{\mathsf{opt}_k(\mathcal{X})}, \qquad \mathsf{opt}_k(\mathcal{X}) = \min_{C:|C|=k} \sum_{i=1}^{n} \min_{c \in C} \|x_i - c\|^2.$$

Define the normalized empirical distortion

$$\frac{1}{n} \mathsf{opt}_k(\mathcal{X}).$$

By Lemma 22, for any fixed $k$, with probability at least $1 - \frac{1}{\mathrm{poly}(n)}$,

$$\left| \frac{1}{n} \mathsf{opt}_k(\mathcal{X}) - \Delta_k(f) \right| \leq C_f \sqrt{\frac{kD \log n}{n}}.$$

In particular, the same bound holds for $k = 1$ with the factor $k$ removed.

On the other hand, by Zador's theorem on manifolds (Lemma 19 and its matching lower bound),

$$\Delta_1(f) = \Theta_f(1), \qquad \Delta_k(f) = \Theta_f(k^{-2/d}).$$

Combining these estimates yields, with high probability,

$$\frac{1}{n} \mathsf{opt}_1(\mathcal{X}) = \Theta_f(1) + O_f\left( \sqrt{\frac{D \log n}{n}} \right),$$

and

$$\frac{1}{n} \mathsf{opt}_k(\mathcal{X}) = \Theta_f(k^{-2/d}) \left( 1 + O_f\left( \sqrt{\frac{kD \log n}{n}} \right) \right),$$

where the error term is of lower order provided $n \gg k^{1+2/d} \log n$.

Taking the ratio, we obtain

$$\beta_k(\mathcal{X}) = \frac{\mathsf{opt}_1(\mathcal{X})}{\mathsf{opt}_k(\mathcal{X})} = \frac{\frac{1}{n} \mathsf{opt}_1(\mathcal{X})}{\frac{1}{n} \mathsf{opt}_k(\mathcal{X})} = \left( 1 + O_f\left( \sqrt{\frac{kD \log n}{n}} \right) \right) k^{2/d},$$

which proves the first claim.

*Analysis of $\eta(\mathcal{X})$.* By definition,

$$\eta(\mathcal{X}) = \frac{\max_{i \neq j} \|x_i - x_j\|}{\min_{i \neq j} \|x_i - x_j\|}.$$

Since $f$ is supported on a compact set, we have

$$\max_{i \neq j} \|x_i - x_j\| \leq \mathrm{diam}(\mathrm{supp}\, f) = O_f(1) \quad \text{almost surely.}$$

It remains to lower bound the minimum interpoint distance.

By the volume growth property of a smooth $d$-dimensional manifold and the lower bound on the density of $f$, there exists a constant $c_f > 0$ such that for all sufficiently small $r$ and all $x \in \mathcal{M}$,

$$\mathbf{Pr}(\|X - x\| \leq r) \leq c_f r^d.$$

Applying a union bound over all unordered pairs $(i, j)$ with $i \neq j$, we obtain

$$\mathbf{Pr}\left( \min_{i \neq j} \|x_i - x_j\| \leq r \right) \leq n^2 c_f r^d.$$

Choosing $r = n^{-3/d}$ yields

$$\mathbf{Pr}\left( \min_{i \neq j} \|x_i - x_j\| \leq r \right) \leq c_f n^2 \cdot \frac{1}{n^3} = \frac{1}{\mathrm{poly}(n)},$$

for all sufficiently large $n$. Therefore, with probability at least $1 - \frac{1}{\mathrm{poly}(n)}$,

$$\min_{i \neq j} \|x_i - x_j\| \geq c'_f n^{-3/d}$$

for some constant $c'_f > 0$. Combining this with the diameter bound yields

$$\eta(\mathcal{X}) = \frac{O_f(1)}{\Omega_f(n^{-3/d})} = O_f\left(n^{3/d}\right) = O_f\left(n^{3\varepsilon/2}\right).$$

$\square$

# B. Notation

| # | Symbol | Meaning |
|---|--------|---------|
| 1 | $n$ | number of points in the dataset |
| 2 | $D$ | ambient dimension of the dataset |
| 3 | $k$ | number of clusters into which the dataset is to be partitioned |
| 4 | $\mathcal{X} = \{x_1, \ldots, x_n\}$ | dataset in consideration |
| 5 | $\mu = \mu(\mathcal{X})$ | mean of the dataset $\mathcal{X}$ |
| 6 | $\|\mathcal{X}\|_{\mathrm{F}}$ | frobenius norm of the dataset defined as $\|\mathcal{X}\|_{\mathrm{F}}^2 = \sum_{i=1}^{n} \|x_i\|^2$ |
| 7 | $C = \{c_1, \ldots, c_k\}$ | a set of $k$ cluster centers |
| 8 | $\mathtt{cost}(\mathcal{X}, C)$ | $k$-means cost of dataset $\mathcal{X}$ with respect to $C$ |
| 9 | $\pi(\cdot\|C)$ | $D^2$ distribution with respect to centers $C$ given by $\pi(x\|C) = \frac{\mathtt{cost}(x,C)}{\mathtt{cost}(\mathcal{X},C)}$ |
| 10 | $\kappa(\cdot\|C)$ | proposal distribution with respect to center $c_1$ given by $\kappa(x\|C) = \frac{\|x\|^2 + \|c_1\|^2}{\|\mathcal{X}\|_{\mathrm{F}}^2 + n\|c_1\|^2}$ |
| 11 | $\mathtt{opt}_k(\mathcal{X})$ | optimal $k$-means cost for dataset $\mathcal{X}$ |
| 12 | $\{s_1, \ldots, s_k\}$ | an optimal set of cluster centers |
| 13 | $\mathcal{X} = \bigcup_{j=1}^{k} \mathcal{X}_j$ | optimal clustering partition |
| 14 | $\mathcal{L} = \mathcal{L}_\varrho(C)$ | LSH data structure initialized by parameter $\varrho$ into which the centers $C$ have been insert |
| 15 | $\mathtt{Insert}(c, \mathcal{L})$ | operation to insert a center $c$ in $\mathcal{L}$ |
| 16 | $\mathtt{Query}(x, \mathcal{L})$ | operation to query an approximate nearest neighbor of $x$ from $\mathcal{L}$ |
| 17 | $\mathtt{cost}^{\mathcal{L}}(\mathcal{X}, C)$ | defined as $\sum_{x \in \mathcal{X}} \mathtt{cost}(x, \mathtt{Query}(x, \mathcal{L}))$ |
| 18 | $\pi^{\mathcal{L}}(\cdot\|C)$ | $\pi^{\mathcal{L}}(x\|C) = \frac{\mathtt{cost}^{\mathcal{L}}(x,C)}{\mathtt{cost}^{\mathcal{L}}(\mathcal{X},C)}$ |
| 19 | $t$ | during the $t$th iteration $t$ centers have been chosen |
| 20 | $C_t = \{c_1, \ldots, c_t\}$ | centers chosen uptill $t$ iterations |
| 21 | $H_t, U_t, W_t$ | number of covered clusters, uncovered clusters and wasted iterations uptill iteration $t$ |
| 22 | $\mathcal{X} = \mathcal{H}_t \cup \mathcal{U}_t$ | partitioning of $\mathcal{X}$ into covered and uncovered clusters |
| 23 | $\mathtt{cost}_t(\mathcal{P})$ | defined as $\mathtt{cost}(\mathcal{P}, C_t)$ |
| 24 | $d$ | dimension of the underlying manifold $\mathcal{M}$ |
| 25 | $\varepsilon$ | quantization exponent defined as $\varepsilon := 2/d$ |
| 26 | $\varrho$ | LSH parameter |
| 27 | $\gamma$ | overall exponent defined as $\gamma := \varepsilon + \varrho$ |
| 28 | $f$ | probability density function supported on the underlying manifold $\mathcal{M}$ |

*Table 3.* Index of notation used throughout the appendix

## C. Theoretical Analysis of `Qkmeans`

The goal of this section is to describe our proof techniques for the theoretical analysis of the `Qkmeans` seeding algorithm, which we reproduce here for convenience.

---

**Algorithm 5** QKMEANS$(\mathcal{X}, k, m)$

---

1. **Preprocessing:**

    (a) (Optional) For $\varepsilon_{\mathrm{JL}} \in (0, 1/4)$, apply the JL-like dimensionality reduction from (Makarychev et al., 2019) to $\mathcal{X}$, reducing to dimension
    $$D \in O\left(\frac{\log(k/\varepsilon_{\mathrm{JL}})}{\varepsilon_{\mathrm{JL}}^2}\right)$$
    with distortion $(1 + \varepsilon_{\mathrm{JL}})$.

    (b) Compute the mean $\mu \leftarrow \frac{1}{n}\sum_{x \in \mathcal{X}} x$.

    (c) Center each point: $x \leftarrow x - \mu$ for all $x \in \mathcal{X}$.

    (d) Compute $\|\mathcal{X}\|_{\mathrm{F}}^2 \leftarrow \sum_{x \in \mathcal{X}} \|x\|^2$.

    (e) Compute $\kappa(x) \leftarrow \|x\|^2/\|\mathcal{X}\|_{\mathrm{F}}^2$ for all $x \in \mathcal{X}$.

2. **Initialization:**

    (a) Initialize the data structure INIT$(\mathscr{L}, \varrho)$.

    (b) Sample $c_1 \sim$ UNIFORM$(\mathcal{X})$.

    (c) Set $C_1 \leftarrow \{c_1\}$ and insert $c_1$ into $\mathscr{L}$.

3. **Main loop:**

    (a) **for** $t = 2$ **to** $k$ **do**

        i. Initialize $iter \leftarrow 0$ and sample $s \sim$ UNIFORM$(\mathcal{X})$.

        ii. **while** $iter < m \ln k$ **do**

            A. $iter \leftarrow iter + 1$.

            B. Sample $x \sim \kappa(\cdot \mid C_{t-1})$, where
    $$\kappa(x \mid C_{t-1}) := \frac{\|x\|^2 + \|c_1\|^2}{\|\mathcal{X}\|_{\mathrm{F}}^2 + n\|c_1\|^2}.$$

            C. Sample $R \sim$ UNIFORM$(0, 1)$.

            D. Compute
    $$r \leftarrow \frac{\mathtt{cost}(x, \mathtt{Query}(\mathscr{L}, x))}{2\varrho^{-1}\big(\|x\|^2 + \|c_1\|^2\big)}.$$

            E. **if** $R \leq r$ **then**

                Set $s \leftarrow x$. **break**.

        iii. Set $c_t \leftarrow s$, insert $c_t$ into $\mathscr{L}$, and update
    $$C_t \leftarrow C_{t-1} \cup \{c_t\}.$$

4. **Return** $C_k$.

---

### C.1. Preprocessing

The Preprocessing step (PREPROCESS$(\mathcal{X})$, Algorithm 1) clearly takes a time of $O(nD)$ as it involves a JL-like linear transformation on the dataset, followed by a couple of for loops iterating over the transformed $D$-dimensional dataset of $n$ points.

Consider the vector $v_{\mathcal{X}} \in \mathbb{R}^n$ given by $v_{\mathcal{X}}(x) = \|x\|$. Define $\kappa(x) = \frac{\|x\|^2}{\|\mathcal{X}\|_{\mathrm{F}}^2}$ as a distribution over $\mathcal{X}$. We will use a (complete) binary tree data structure to sample from $\kappa(\cdot)$. The leaves of the binary tree correspond to the entries of $v_{\mathcal{X}}$ and

store weight $v_{\mathcal{X}}(x)^2$ along with the sign of $v_{\mathcal{X}}(x)$. The internal nodes also store a weight that is equal to the sum of weights of its children. To sample from $\kappa(\cdot)$, we traverse the tree, choosing either to go left or right at each node with probability proportional to the weight of its two children until reaching the leaves. The binary tree similarly supports querying and updating the entries of $v_{\mathcal{X}}$.

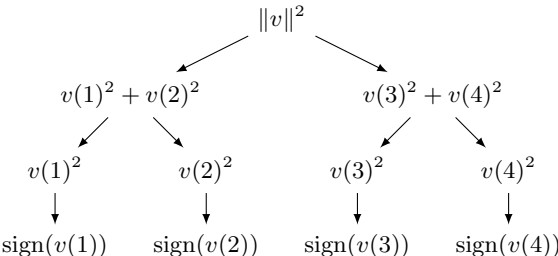

*Figure 4.* Data structure for sampling from a vector $v \in \mathbb{R}^4$

We state this formally following (Tang, 2019), in which such data structures, called *sample and query access data structures* were introduced.

**Lemma 24.** *(Lemma 3.1 in (Tang, 2019)) There exists a data structure storing a vector $v \in \mathbb{R}^n$ with $\nu$ nonzero entries in $O(\nu \log n)$ space which supports the following operations:*

- *Reading and updating an entry of $v$ in $O(\log n)$ time*

- *Finding $\|v\|^2$ in $O(1)$ time*

- *Generating an independent sample $i \in \{1, \ldots, n\}$ with probability $\frac{v(i)^2}{\sum_{j=1}^{n} v(j)^2}$ in $O(\log n)$ time.*

Note that if $n$ is not a perfect power of 2 then we can find a $n' \in \mathbf{N}$ which is a perfect power of 2 such that $n' < n < 2n'$. We can then set the remaining $2n' - n$ data points to have zero norm and use this dataset instead to construct the complete binary tree. Thus the following corollary is immediate.

**Corollary 25.** *There is a data structure that can be prepared in $O(nD)$ time which enables generating a sample from $\kappa(\cdot)$ in $O(\log n)$ time.*

Looking ahead, the purpose of preprocessing is to enable fast sampling from the distribution $\kappa(\cdot|C)$ used in SAMPLE($\mathscr{L}, C, m$), Algorithm 2.

### C.2. ANNS

In order to employ the rejection sampling idea, the probability of accepting a sampled point $x \in \mathcal{X}$ needs to be proportional to $\mathrm{cost}(x, S)$, where $S$ is the current set of centers. Again, computing this quantity takes $\Theta(kD)$ time, which contributes $\Theta(k^2 D)$ time on performing this sampling $k$ times. Thus, we apply another optimization technique; we use an approximate nearest neighbor data structure to approximate the distance between $x$ and the closest center.

In particular, we utilize the following data structure, developed by (Cohen-Addad et al., 2020a), which is based on locality-sensitive hash functions (Andoni & Indyk, 2008).

**Lemma 26.** *(LSH-based ANNS data structure) For any set of points $P = \{p_1, \ldots, p_t\} \subset \mathbb{R}^D$ and parameter $\varrho < 1$, there exists a data structure $\mathscr{L}$ with operations* Insert *and* Query *such that with probability atleast $1 - 1/t$, the following guarantees hold* [7]

1. Insert($\mathscr{L}, p$) *inserts the point $p \in P$ into $\mathscr{L}$ in time $O\left(D \log \eta (t \log \eta)^\varrho\right)$.*

2. Query($\mathscr{L}, p$) *returns a point such that $\|p - \text{Query}(\mathscr{L}, p)\| \leq \frac{1}{\sqrt{\varrho}} \cdot \min_{q \in \mathscr{L}} \|p - q\|$ and the query time is $O\left(D \log \eta (t \log \eta)^\varrho\right)$.*

3. $\mathscr{L}$ *is monotone under insertions: the distance between $p$ and* Query($\mathscr{L}, p$) *is non-increasing after inserting more points.*

*Here, $\eta$ is the aspect ratio of the set $P$ defined as $\eta(P) = \frac{\max_{x \neq y \in P} \|x - y\|}{\min_{x \neq y \in P} \|x - y\|}$.*

We shall be using this data structure $\mathscr{L}$ to store the sampled centers $S$ and to query the approximate nearest center from $S$. To this end, we define $\text{cost}^{\mathscr{L}}(P, S) = \sum_{p \in P} \|p - \text{Query}(\mathscr{L}, p)\|^2$ and $\pi^{\mathscr{L}}(p|S) = \frac{\text{cost}^{\mathscr{L}}(p,S)}{\text{cost}^{\mathscr{L}}(P,S)}$ for a set of points $P$ and a set of centers $S \subset \mathbb{R}^D$. Conditioned on the data structure being successful we have the following :

**Lemma 27** (LSH approximation bounds). *Let $X = \{x_1, \ldots, x_n\} \subset \mathbb{R}^D$ be a dataset and $S \subset X$ be a set of centers sampled from $X$. Suppose that the data structure $\mathscr{L}$ with parameter $\varrho < 1$ is successful and each $c \in S$ is inserted into the data structure $\mathscr{L}$. Then for each $x \in X$, the following holds*

$$\text{cost}(x, S) \leq \text{cost}^{\mathscr{L}}(x, S) \leq \varrho^{-1} \cdot \text{cost}(x, S)$$

*from which we also have*

$$\varrho \cdot \pi(x|S) \leq \pi^{\mathscr{L}}(x|S) \leq \varrho^{-1} \cdot \pi(x|S)$$

**Remark 28** (Failure Probability of the LSH data structure). *For the rest of the analysis, we condition on the event that the data structure is* successful. *To understand the effect of the failure probability of the LSh data structure on the approximation guarantee, we refer the reader to a similar discussion from (Cohen-Addad et al., 2020b).*

*Given the dataset, the hash tables which constitute the LSH data structure are chosen in the beginning in such a manner that with probability $1 - 1/n$, they follow certain desirable property (farther pairs of points map to distinct hashes, closer points map to same hash). See Corollary D.6 of previous work (Cohen-Addad et al., 2020b).*

*Assuming this property successfully holds for all $\binom{n}{2}$ points in the dataset, the Insert and Query operations will always be successful. So the failure probability of $1/n$ is valid only during the initialization phase; henceforth we can work with the assumption that all operations are successful with probability $1 - 1/n$. Also, see the footnote on Page 6 of previous work (Cohen-Addad et al., 2020b).*

*Suppose we repeat our LSH initialization followed by the seeding procedure $M$ times and take the best clustering. Then with probability atleast $1 - p = 1 - 1/n^M$, one of the intializations would be successful. In that case, the expected cost is bounded by $O(\log k)\text{opt}_k$. In the case of failure, since we choose the first center uniformly at random, the expected cost is bounded by $2\text{opt}_1$. Hence, the total expected cost can be bounded as : $\mathbb{E}[ALG] \leq (1 - p) \cdot O(\log k)\text{opt}_k + 2p \cdot \text{opt}_1 \leq O(\log k) \cdot \text{opt}_k(1 + 2p\beta_k/\log k).$*

*Choosing $M = \log_n(2n\beta_k)$ shows that $\mathbb{E}[ALG] \leq O(\log k)\text{opt}_k(1 + O(1/n))$. Hence, the effect of LSH failure has negligible impact on the output quality of our method.*

### C.3. Analysis of $(\varrho, \delta)$-$k$-means++

The speedup in our algorithm can be attributed to two things; the first being using rejection sampling for sampling for the $D^2$ distribution and second, using the approximate nearest center for fast computation of the rejection sampling probability. Our goal is to bound the expected cost of the output of our algorithm. To do this, we analyze the following *abstract* version of $k$-means++, which we call $(\varrho, \delta)$-$k$-means++.

---

**Algorithm 6** $(\varrho, \delta)$-$k$-means++

---

**Input:** dataset $\mathcal{X} = \{x_1, \ldots, x_n\} \subset \mathbb{R}^D$, parameters $0 < \varrho, \delta < 1$ and $k \in \mathbb{N}$

1: $c_1 \leftarrow \text{Uniform}(\mathcal{X})$
2: $\mathscr{L} \leftarrow$ ANNS data structure with parameter $\varrho$
3: $\text{Insert}(c_1, \mathscr{L})$ ; $C \leftarrow \{c_1\}$
4: **for** $i = 2$ to $k$ **do**
5:      Sample $y$ with probability $\pi_{(\varrho, \delta)}(y|C) := (1 - \delta) \cdot \pi^{\mathscr{L}}(y|C) + \delta \cdot \frac{1}{n}$
6:      $c_i \leftarrow y$; $\text{Insert}(c_i, \mathscr{L})$; $C \leftarrow C \cup \{c_i\}$
7: **end for**
8: **Output** $C$

---

#### C.3.1. THE PERTURBED DISTRIBUTION

**Definition 29** (($(\varrho, \delta)$ perturbed $D^2$ distribution). *Let $\mathcal{X} = \{x_1, \ldots, x_n\} \subset \mathbb{R}^d$ be a dataset and $S \subset X$ be a set of centers sampled from $X$. Suppose that the data structure $\mathscr{L}$ with parameter $\varrho < 1$ is successful and each $c \in S$ is inserted into the*

*data structure $\mathcal{L}$. Then for each $x \in \mathcal{X}$, we define the $(\varrho, \delta)$ perturbed $D^2$ distribution as :*

$$\pi_{(\varrho,\delta)}(x|S) = (1 - \delta) \cdot \pi^{\mathcal{L}}(x|S) + \delta \cdot \frac{1}{n}$$

Now, we provide some useful bounds that a point is chosen according to the perturbed distribution, conditioned on the event that it belongs to a specified set of points beforehand. These bounds will be useful later in several places.

**Lemma 30.** *(Bounds on perturbed sampling probabilities) Let $\mathcal{X} = \{x_1, \ldots, x_n\} \subset \mathbb{R}^d$ be a dataset and $\mathcal{Y} \subseteq \mathcal{X}$ be an arbitrary non-empty subset of $\mathcal{X}$. Let $C \subset \mathcal{X}$ be a set of centers and suppose the data structure $\mathcal{L}$ with parameter $\varrho < 1$ is successful and each $c \in C$ is inserted into $\mathcal{L}$. For any point $y \in \mathcal{Y}$ and parameter $\delta \in (0, 1)$, the following hold:*

1. $\mathbf{Pr}_{y \sim \pi_{(\varrho,\delta)}(\cdot|C)}[y \mid y \in \mathcal{Y}] \leq \frac{\mathsf{cost}^{\mathcal{L}}(y, C)}{\mathsf{cost}^{\mathcal{L}}(\mathcal{Y}, C)} + \frac{\delta}{1-\delta} \cdot \frac{1}{n} \cdot \frac{\mathsf{cost}^{\mathcal{L}}(\mathcal{X}, C)}{\mathsf{cost}^{\mathcal{L}}(\mathcal{Y}, C)}$

2. $\mathbf{Pr}_{y \sim \pi_{(\varrho,\delta)}(\cdot|C)}[y \mid y \in \mathcal{Y}] \geq \frac{\mathsf{cost}^{\mathcal{L}}(y, C)}{\mathsf{cost}^{\mathcal{L}}(\mathcal{Y}, C)} - \frac{\delta}{1-\delta} \cdot \frac{|\mathcal{Y}|}{n} \cdot \frac{\mathsf{cost}^{\mathcal{L}}(y, C) \cdot \mathsf{cost}^{\mathcal{L}}(\mathcal{X}, C)}{\mathsf{cost}^{\mathcal{L}}(\mathcal{Y}, C)^2}$

*Proof.* Let us first compute the probability that a point sampled from the perturbed $D^2$ distribution belongs to $\mathcal{Y}$ :

$$\mathbf{Pr}_{y \sim \pi_{(\varrho,\delta)}(\cdot|C)}[y \in \mathcal{Y}] = \sum_{y' \in \mathcal{Y}} \left( (1 - \delta) \frac{\mathsf{cost}^{\mathcal{L}}(y', C)}{\mathsf{cost}^{\mathcal{L}}(\mathcal{X}, C)} + \delta \cdot \frac{1}{n} \right) = (1 - \delta) \frac{\mathsf{cost}^{\mathcal{L}}(\mathcal{Y}, C)}{\mathsf{cost}^{\mathcal{L}}(\mathcal{X}, C)} + \delta \cdot \frac{|\mathcal{Y}|}{n}$$

Hence the required conditional probability is

$$\mathbf{Pr}_{y \sim \pi_{(\varrho,\delta)}(\cdot|C)}[y \mid y \in \mathcal{Y}] = \frac{(1 - \delta) \frac{\mathsf{cost}^{\mathcal{L}}(y, C)}{\mathsf{cost}^{\mathcal{L}}(\mathcal{X}, C)} + \delta \cdot \frac{1}{n}}{(1 - \delta) \frac{\mathsf{cost}^{\mathcal{L}}(\mathcal{Y}, C)}{\mathsf{cost}^{\mathcal{L}}(\mathcal{X}, C)} + \delta \cdot \frac{|\mathcal{Y}|}{n}}$$

For part 1, we have:

$$\frac{(1 - \delta) \frac{\mathsf{cost}^{\mathcal{L}}(y, C)}{\mathsf{cost}^{\mathcal{L}}(\mathcal{X}, C)} + \delta \cdot \frac{1}{n}}{(1 - \delta) \frac{\mathsf{cost}^{\mathcal{L}}(\mathcal{Y}, C)}{\mathsf{cost}^{\mathcal{L}}(\mathcal{X}, C)} + \delta \cdot \frac{|\mathcal{Y}|}{n}} \leq \frac{(1 - \delta) \frac{\mathsf{cost}^{\mathcal{L}}(y, C)}{\mathsf{cost}^{\mathcal{L}}(\mathcal{X}, C)} + \delta \cdot \frac{1}{n}}{(1 - \delta) \frac{\mathsf{cost}^{\mathcal{L}}(\mathcal{Y}, C)}{\mathsf{cost}^{\mathcal{L}}(\mathcal{X}, C)}}$$

$$= \frac{\mathsf{cost}^{\mathcal{L}}(y, C)}{\mathsf{cost}^{\mathcal{L}}(\mathcal{Y}, C)} + \frac{\delta}{1 - \delta} \cdot \frac{1}{n} \cdot \frac{\mathsf{cost}^{\mathcal{L}}(\mathcal{X}, C)}{\mathsf{cost}^{\mathcal{L}}(\mathcal{Y}, C)}$$

For part 2, we have:

$$\frac{(1 - \delta) \frac{\mathsf{cost}^{\mathcal{L}}(y, C)}{\mathsf{cost}^{\mathcal{L}}(\mathcal{X}, C)} + \delta \cdot \frac{1}{n}}{(1 - \delta) \frac{\mathsf{cost}^{\mathcal{L}}(\mathcal{Y}, C)}{\mathsf{cost}^{\mathcal{L}}(\mathcal{X}, C)} + \delta \cdot \frac{|\mathcal{Y}|}{n}} = \frac{\mathsf{cost}^{\mathcal{L}}(y, C)}{\mathsf{cost}^{\mathcal{L}}(\mathcal{Y}, C)} \left( \frac{1 + \frac{\delta}{1-\delta} \cdot \frac{\mathsf{cost}^{\mathcal{L}}(\mathcal{X}, C)}{n \cdot \mathsf{cost}^{\mathcal{L}}(y, C)}}{1 + \frac{\delta}{1-\delta} \cdot \frac{|\mathcal{Y}| \cdot \mathsf{cost}^{\mathcal{L}}(\mathcal{X}, C)}{n \cdot \mathsf{cost}^{\mathcal{L}}(\mathcal{Y}, C)}} \right)$$

$$\geq \frac{\mathsf{cost}^{\mathcal{L}}(y, C)}{\mathsf{cost}^{\mathcal{L}}(\mathcal{Y}, C)} \left( 1 + \frac{\delta}{1 - \delta} \cdot \frac{|\mathcal{Y}| \cdot \mathsf{cost}^{\mathcal{L}}(\mathcal{X}, C)}{n \cdot \mathsf{cost}^{\mathcal{L}}(\mathcal{Y}, C)} \right)^{-1}$$

$$\geq \frac{\mathsf{cost}^{\mathcal{L}}(y, C)}{\mathsf{cost}^{\mathcal{L}}(\mathcal{Y}, C)} \left( 1 - \frac{\delta}{1 - \delta} \cdot \frac{|\mathcal{Y}| \cdot \mathsf{cost}^{\mathcal{L}}(\mathcal{X}, C)}{n \cdot \mathsf{cost}^{\mathcal{L}}(\mathcal{Y}, C)} \right)$$

where in the last step we used $(1 + x)^{-1} \geq 1 - x$ for any $x \geq 0$. $\qquad\square$

### C.3.2. SOME USEFUL LEMMAS

Suppose $S = \{s_1, \ldots, s_k\}$ is am optimal set of centers for the dataset $\mathcal{X}$ so that $\mathsf{cost}(\mathcal{X}, S) = \mathsf{opt}_k(\mathcal{X})$. Let $\mathcal{X}_j$ denote the set of points from $\mathcal{X}$ whose nearest center is $s_j$. This induces a partition $\mathcal{X} = \bigcup_{j \in [k]} \mathcal{X}_j$ where $\mathcal{X}_i \cap \mathcal{X}_j = \emptyset$ and $s_j = \mu(\mathcal{X}_j)$. Notice that we can decompose the optimal $k$-means cost as $\mathsf{opt}_k(\mathcal{X}) = \sum_{j \in [k]} \mathsf{opt}_1(\mathcal{X}_j)$ Let us state a few results from (Arthur & Vassilvitskii, 2007) that will be useful.

**Lemma 31.** *(First center cost, Lemma 3.1 of (Arthur & Vassilvitskii, 2007)) Let $c_1$ be the first center chosen by Algorithm 6. For an optimal choice of cluster centers $S$, let $\mathcal{X}_j$ denote the cluster corresponding to the $j$th center $s_j$ for $j \in [k]$. Then:*

$$\mathbf{E}[\mathsf{cost}(\mathcal{X}_j, c_1) | c_1 \in \mathcal{X}_j] = 2\mathsf{opt}_1(\mathcal{X}_j)$$

**Lemma 32** (Standard $k$-means++ bound (Lemma 3.2 of (Arthur & Vassilvitskii, 2007))). *For an optimal choice of cluster centers $S$, let $\mathcal{X}_j$ denote the cluster corresponding to the $j$th center for $j \in [k]$. Let $C$ be a set of chosen centers, and let $y$ be a new center chosen according to the (standard) $D^2$ distribution such that $y \in \mathcal{X}_j$. Then:*

$$\mathbf{E}[\mathsf{cost}(\mathcal{X}_j, C \cup \{y\}) \mid C, \{y \in \mathcal{X}_j\}] \leq 8\mathsf{opt}_1(\mathcal{X}_j)$$

The following is an analogous version of Lemma 3.2 of (Arthur & Vassilvitskii, 2007), but we need additional analysis since we use the perturbed distribution instead of the exact $D^2$ distribution:

**Lemma 33** (Perturbed center selection bound). *Consider an iteration of Algorithm 6 after the first one and let $C$ be the current set of centers. Let $y$ be the center chosen in the current iteration. For an optimal choice of cluster centers $S$, let $\mathcal{X}_j$ denote the cluster corresponding to the $j$th center for $j \in [k]$. Then for any $C \neq \emptyset$:*

$$\mathbf{E}[\mathsf{cost}(\mathcal{X}_j, C \cup \{y\}) \mid C, \{y \in \mathcal{X}_j\}] \leq \varrho^{-1}\left(8\mathsf{opt}_1(\mathcal{X}_j) + \frac{\delta}{1-\delta}\frac{|\mathcal{X}_j|}{n}\mathsf{cost}(\mathcal{X}, C)\right)$$

*Proof.* When the new center $y$ is added, each point $x \in \mathcal{X}_j$ contributes $\mathsf{cost}(x, C \cup \{y\}) = \min(\|x - y\|^2, \mathsf{cost}(x, C))$ to the overall cost, where $\mathscr{L}$ now contains $C \cup \{y\}$. The expected cost of the cluster $\mathcal{X}_j$ can hence be written as:

$$\mathbf{E}[\mathsf{cost}(\mathcal{X}_j, C \cup \{y\}) \mid C, \{y \in \mathcal{X}_j\}] \qquad \Big\rangle \text{\textit{Expanding expectation}}$$

$$= \sum_{y \in \mathcal{X}_j} \mathbf{Pr}_{y \sim \pi_{(\varrho,\delta)}(\cdot|C)}[y \mid y \in \mathcal{X}_j] \cdot \mathsf{cost}(\mathcal{X}_j, C \cup \{y\}) \qquad \Big\rangle \text{\textit{Expanding} } \mathsf{cost}(\mathcal{X}_j, C \cup \{y\})$$

$$= \sum_{y \in \mathcal{X}_j} \mathbf{Pr}_{y \sim \pi_{(\varrho,\delta)}(\cdot|C)}[y \mid y \in \mathcal{X}_j] \cdot \left(\sum_{x \in \mathcal{X}_j} \mathsf{cost}(x, C \cup \{y\})\right) \qquad \Big\rangle \text{\textit{Lemma 30 item (1)}}$$

$$\leq \sum_{y \in \mathcal{X}_j} \left(\frac{\mathsf{cost}^{\mathscr{L}}(y, C)}{\mathsf{cost}^{\mathscr{L}}(\mathcal{X}_j, C)} + \frac{\delta}{1-\delta}\frac{1}{n}\frac{\mathsf{cost}^{\mathscr{L}}(\mathcal{X}, C)}{\mathsf{cost}^{\mathscr{L}}(\mathcal{X}_j, C)}\right) \cdot \left(\sum_{x \in \mathcal{X}_j} \mathsf{cost}(x, C \cup \{y\})\right)$$

$$\leq \underbrace{\sum_{y \in \mathcal{X}_j} \frac{\mathsf{cost}^{\mathscr{L}}(y, C)}{\mathsf{cost}^{\mathscr{L}}(\mathcal{X}_j, C)} \sum_{x \in \mathcal{X}_j} \min(\|x - y\|^2, \mathsf{cost}(y, C))}_{(I)}$$

$$+ \underbrace{\sum_{y \in \mathcal{X}_j} \left(\frac{\delta}{1-\delta}\frac{1}{n}\frac{\mathsf{cost}(\mathcal{X}, C)}{\mathsf{cost}^{\mathscr{L}}(\mathcal{X}_j, C)}\right) \sum_{x \in \mathcal{X}_j} \min(\|x - y\|^2, \mathsf{cost}(x, C))}_{II}$$

where in the third step we used part 1 of Lemma 30. Using Lemma 27 and Lemma 3.2 of (Arthur & Vassilvitskii, 2007), we have for $(I)$:

$$\sum_{y \in \mathcal{X}_j} \frac{\mathsf{cost}^{\mathscr{L}}(y, C)}{\mathsf{cost}^{\mathscr{L}}(\mathcal{X}_j, C)} \sum_{x \in \mathcal{X}_j} \min(\|x - y\|^2, \mathsf{cost}(x, C)) \leq 8\varrho^{-1}\mathsf{opt}_1(\mathcal{X}_j)$$

For the second term $(II)$, noting that $\sum_{x \in \mathcal{X}_j} \min(\|x - y\|^2, \mathsf{cost}(x, C)) \leq \sum_{x \in \mathcal{X}_j} \mathsf{cost}(x, C) = \mathsf{cost}(\mathcal{X}_j, C)$, we have:

$$\sum_{y \in \mathcal{X}_j} \left(\frac{\delta}{1-\delta}\frac{1}{n}\frac{\mathsf{cost}^{\mathscr{L}}(\mathcal{X}, C)}{\mathsf{cost}^{\mathscr{L}}(\mathcal{X}_j, C)}\right) \sum_{x \in \mathcal{X}_j} \min(\|x - y\|^2, \mathsf{cost}(x, C))$$

$$\leq \varrho^{-1} \cdot \sum_{y \in \mathcal{X}_j} \left(\frac{\delta}{1-\delta}\frac{1}{n}\frac{\mathsf{cost}(\mathcal{X}, C)}{\mathsf{cost}(\mathcal{X}_j, C)}\right) \mathsf{cost}(\mathcal{X}_j, C) = \frac{\varrho^{-1}\delta}{1-\delta}\frac{|\mathcal{X}_j|}{n}\mathsf{cost}(\mathcal{X}, C)$$

Combining both terms completes the proof. $\qquad \square$

### C.3.3. POTENTIAL ANALYSIS

Following (Dasgupta, 2003), we first setup some notation. Let $t \in \{0, \ldots, k\}$ denote the number of centers already chosen by Algorithm 6. Let $C_t := \{c_1, \ldots, c_t\}$ be the set of centers after $t$ iterations . We say that an optimal cluster $\mathcal{X}_j$ is *covered* by $C_t$ if at least one of the chosen centers is in $\mathcal{X}_j$. If not, then it is *uncovered*. We denote $H_t = \{j \in \{1, \ldots, k\} : \mathcal{X}_j \cap C_t \neq \emptyset\}$ and $U_t = \{1, \ldots, k\} \backslash H_t$. Similarly, the dataset $\mathcal{X}$ can be partitioned into two parts: $\mathcal{H}_t \subset \mathcal{X}$ being the points belonging to *covered* clusters and $\mathcal{U}_t = \mathcal{X} \backslash \mathcal{H}_t$ being the points belonging to *uncovered* clusters. Let $W_t = t - |H_t|$ denote the number of *wasted* iterations so far, i.e., the number of iterations in which no new cluster was covered. Note that we always have $|H_t| \leq t$ and hence $|U_t| \geq k - t$. For any $\mathcal{Y} \subset \mathcal{X}$, we use the notation $\text{cost}_t(\mathcal{Y}) = \text{cost}(\mathcal{Y}, C_t)$ and $\text{cost}_t^{\mathscr{L}}(\mathcal{Y}) = \text{cost}^{\mathscr{L}}(\mathcal{Y}, C_t)$ for brevity. The total cost can be decomposed as:

$$\text{cost}_t(\mathcal{X}) = \text{cost}_t(\mathcal{H}_t) + \text{cost}_t(\mathcal{U}_t) \leq \text{cost}_t(\mathcal{H}_t) + \text{cost}_t^{\mathscr{L}}(\mathcal{U}_t)$$

We can bound the first term directly using the previous lemmas.

**Lemma 34.** *(Cost bound for covered clusters) For each $t \in \{1, \ldots, k\}$ the following holds:*

$$\mathbf{E}[\text{cost}_t(\mathcal{H}_t)] \leq \varrho^{-1}\left(8\text{opt}_k(\mathcal{X}) + \frac{2\delta}{1-\delta}\text{opt}_1(\mathcal{X})\right)$$

*Proof.* Consider any cluster $j \in H_t$. This means that at some iteration $t_j$, a point $y \in \mathcal{X}_j$ would have been selected as the new center $c_{t_j}$ so that $j$ becomes covered. This observation allows us to use Lemma 33 in the following manner :

$$\mathbf{E}[\text{cost}_t(\mathcal{H}_t)] = \mathbf{E}\left[\sum_{j \in H_t} \text{cost}_t(\mathcal{X}_j)\right] \qquad \left.\begin{array}{c}\\\\\end{array}\right\rbrace \text{\textit{Using linearity of expectation}}$$

$$= \sum_{j \in H_t} \mathbf{E}[\text{cost}_t(\mathcal{X}_j)] \qquad \left.\begin{array}{c}\\\\\end{array}\right\rbrace \text{\textit{Using Lemma 33}}$$

$$\leq \sum_{j \in H_t} \varrho^{-1}\left(8\text{opt}_1(\mathcal{X}_j) + \frac{\delta}{1-\delta}\frac{|\mathcal{X}_j|}{n}\text{cost}(\mathcal{X}, C_{t_j-1})\right) \qquad \left.\begin{array}{c}\\\\\\\end{array}\right\rbrace \begin{array}{l}\text{\textit{Using}}\\ \mathbf{E}[\text{cost}(\mathcal{X}, C_{t_j-1})] \leq\\ \mathbf{E}[\text{cost}(\mathcal{X}, c_1)] =\\ 2\text{opt}_1(\mathcal{X})\end{array}$$

$$\leq \sum_{j \in H_t} \varrho^{-1}\left(8\text{opt}_1(\mathcal{X}_j) + \frac{\delta}{1-\delta}\frac{|\mathcal{X}_j|}{n} \cdot 2\text{opt}_1(\mathcal{X})\right) \qquad \left.\begin{array}{c}\\\\\end{array}\right\rbrace \begin{array}{l}\text{\textit{Expanding the sum}}\\ \text{\textit{over }} j \in H_t \text{ \textit{to}}\\ j \in [k]\end{array}$$

$$= \varrho^{-1}\left(8\sum_{j \in [k]} \text{opt}_1(\mathcal{X}_j) + \frac{2\delta}{1-\delta} \cdot \frac{\sum_{j \in [k]}|\mathcal{X}_j|}{n} \cdot \text{opt}_1(\mathcal{X})\right) \qquad \left.\begin{array}{c}\\\\\end{array}\right\rbrace \begin{array}{l}\text{\textit{Using }} \text{opt}_k(\mathcal{X}) =\\ \sum_{j \in [k]} \text{opt}_1(\mathcal{X}_j)\end{array}$$

$$= \varrho^{-1}\left(8\text{opt}_k(\mathcal{X}) + \frac{2\delta}{1-\delta}\text{opt}_1(\mathcal{X})\right)$$

This completes the proof.

$\square$

**Potential function.** To bound the second term, i.e., the cost of the uncovered clusters, we use the potential function method introduced in (Dasgupta, 2013), with a slightly modified potential function:

$$\varphi_t = \frac{W_t}{|U_t|}\text{cost}_t^{\mathscr{L}}(\mathcal{U}_t)$$

Instead of *paying* the complete clustering cost of the uncovered clusters at once, we make sure that at the end of iteration $t$, we have paid an amount of at least $\varphi_t$. Observe that when $t = k$, we have $W_k = |U_k|$, so the potential becomes $\text{cost}_k^{\mathscr{L}}(\mathcal{U}_k)$, which is atleast $\text{cost}_k(\mathcal{U}_k)$, i.e. the total cost of the uncovered clusters returned by Algorithm 3. We now show how to bound the expected increase in the potential, i.e., $\varphi_{t+1} - \varphi_t$ at each iteration. The total potential bound will then follow by expressing $\varphi_k$ as a telescoping sum of the increments.

Suppose $t$ centers have been chosen. The next center $c_{t+1}$ is chosen which belongs to some optimal cluster $\mathcal{X}_j$. We consider two cases: the first case is when $j \in U_t$, i.e., a new cluster is covered, and the second case is when $j \in H_t$, i.e., the center is chosen from an already covered cluster. We shall denote the set of all random variables after the end of iteration $t$ by $\mathbf{F}_t$.

**Lemma 35.** *(Potential increment for uncovered clusters) For any $t \in \{1, \ldots, k-1\}$, the following holds:*

$$\mathbf{E}[\varphi_{t+1} - \varphi_t \mid \mathsf{F}_t, \{j \in U_t\}] \leq \frac{2\delta}{1-\delta} \frac{t}{\max(1, k-t-1)^2} \mathsf{opt}_1(\mathcal{X})$$

*Proof.* When $j \in U_t$, we have $W_{t+1} = W_t$, $H_{t+1} = H_t \cup \{j\}$, and $U_{t+1} = U_t \setminus \{j\}$. Thus using the monotonicity of the data structure $L$, we have

$$\varphi_{t+1} = \frac{W_{t+1}}{|U_{t+1}|} \mathsf{cost}_{t+1}^{\mathscr{L}}(U_{t+1}) \leq \frac{W_t}{|U_t| - 1} \left( \mathsf{cost}_t^{\mathscr{L}}(U_t) - \mathsf{cost}_t^{\mathscr{L}}(\mathcal{X}_j) \right)$$

We can use part 2 of Lemma 30 for getting a lower bound on the second term:

$$\mathbf{E}[\mathsf{cost}_t^{\mathscr{L}}(\mathcal{X}_j) \mid \mathsf{F}_t, \{j \in U_t\}]$$
$$\geq \sum_{j \in U_t} \left( \frac{\mathsf{cost}_t^{\mathscr{L}}(\mathcal{X}_j)}{\mathsf{cost}_t^{\mathscr{L}}(U_t)} - \frac{\delta}{1-\delta} \frac{|\mathcal{X}_j|}{n} \frac{\mathsf{cost}_t^{\mathscr{L}}(\mathcal{X}_j)\mathsf{cost}_t^{\mathscr{L}}(X)}{\mathsf{cost}_t^{\mathscr{L}}(U_t)^2} \right) \mathsf{cost}_t^{\mathscr{L}}(\mathcal{X}_j)$$
$$\geq \left( 1 - \frac{\delta}{1-\delta} \frac{\mathsf{cost}_t^{\mathscr{L}}(X)}{\mathsf{cost}_t^{\mathscr{L}}(U_t)} \right) \sum_{j \in U_t} \frac{\mathsf{cost}_t^{\mathscr{L}}(\mathcal{X}_j)^2}{\mathsf{cost}_t^{\mathscr{L}}(U_t)}$$

where in the second step we used the fact that $|\mathcal{X}_j| \leq n$ for each $j \in U_t$. We can use the Cauchy-Schwarz inequality[8] to simplify the last expression as follows:

$$|U_t|^2 \sum_{j \in U_t} \mathsf{cost}_t^{\mathscr{L}}(\mathcal{X}_j)^2 \geq |U_t| \sum_{j \in U_t} \mathsf{cost}_t^{\mathscr{L}}(\mathcal{X}_j) = |U_t| \mathsf{cost}_t^{\mathscr{L}}(U_t)$$

This shows that

$$\mathbf{E}[\mathsf{cost}_t^{\mathscr{L}}(\mathcal{X}_j) \mid \mathsf{F}_t, \{j \in U_t\}] \geq \frac{\mathsf{cost}_t^{\mathscr{L}}(U_t)}{|U_t|} - \frac{\delta}{1-\delta} \frac{\mathsf{cost}_t^{\mathscr{L}}(X)}{|U_t|}$$

Now,

$$\mathbf{E}[\varphi_{t+1} \mid \mathsf{F}_t, i \in U_t] \leq \frac{W_t}{|U_t| - 1} \left( \mathsf{cost}_t^{\mathscr{L}}(U_t) - \mathbf{E}[\mathsf{cost}_t^{\mathscr{L}}(\mathcal{X}_j) \mid \mathsf{F}_t, i \in U_t] \right)$$
$$\leq \frac{W_t}{|U_t| - 1} \left( \mathsf{cost}_t^{\mathscr{L}}(U_t) - \frac{\mathsf{cost}_t^{\mathscr{L}}(U_t)}{|U_t|} + \frac{\delta}{1-\delta} \frac{\mathsf{cost}_t^{\mathscr{L}}(X)}{|U_t|} \right)$$
$$= \varphi_t + \frac{\delta}{1-\delta} \frac{W_t}{|U_t|(|U_t| - 1)} \mathsf{cost}_t^{\mathscr{L}}(\mathcal{X})$$

Recall that $W_t \leq t$ and $|U_t| \geq k - t$. So for $t \leq k - 2$, the following holds after taking expectation:

$$\mathbf{E}[\varphi_{t+1} - \varphi_t \mid \mathsf{F}_t, \{j \in U_t\}] \leq \frac{\delta}{1-\delta} \frac{t}{(k-t-1)^2} \mathbf{E}[\mathsf{cost}_t^{\mathscr{L}}(X) \mid \mathsf{F}_t, j \in U_t]$$
$$\leq \frac{2\delta}{1-\delta} \frac{t}{(k-t-1)^2} \mathsf{opt}_1(X)$$

Now consider the case when $t = k-1$. We cannot use the above argument directly because it may so happen that $|U_{k-1}| = 1$. If this happens, the potential of the uncovered clusters is always 0. This only happens when a new cluster is covered in each iteration. Let this event be $\mathsf{A}$ (for *All Clusters* being covered). Denoting $\mathsf{E}$ to be the event: $\{\mathsf{F}_{k-1}, \{j \in U_{k-1}\}\}$, we have the following:

$$\mathbf{E}[\varphi_k - \varphi_{k-1} \mid \mathsf{E}] = \mathbf{E}[\varphi_k - \varphi_{k-1} \mid \mathsf{E}, \mathsf{A}]\mathbf{Pr}[\mathsf{A} \mid \mathsf{E}] + \mathbf{E}[\varphi_k - \varphi_{k-1} \mid \mathsf{E}, \neg\mathsf{A}]\mathbf{Pr}[\neg\mathsf{A} \mid \mathsf{E}] \quad \begin{smallmatrix} Using \\ \mathbf{E}[\varphi_k - \varphi_{k-1} \mid \\ \mathsf{E}, \mathsf{A}] \leq 0 \end{smallmatrix}$$
$$\leq \mathbf{E}[\varphi_k - \varphi_{k-1} \mid \mathsf{E}, \neg\mathsf{A}]\mathbf{Pr}[\neg\mathsf{A} \mid \mathsf{E}]$$
$$\leq \mathbf{E}[\varphi_k - \varphi_{k-1} \mid \mathsf{E}, \neg\mathsf{A}]$$
$$\leq \frac{2\delta t}{1-\delta} \mathsf{opt}_1(X)$$

where in the last line we used the fact that $|U_{k-1}| > 1$ if all clusters are not covered. We also used the fact that $\mathbf{E}[\varphi_k - \varphi_{k-1} \mid \mathsf{E}, \mathsf{A}] \leq 0$ since in this case there are no uncovered clusters .Combining both cases completes the proof. $\qquad\square$

---

[8] For lists of numbers $a_1, \ldots, a_m$ and $b_1, \ldots, b_m$ we have $\left( \sum_i a_i b_i \right)^2 \leq \left( \sum_i a_i^2 \right) \left( \sum_i b_i^2 \right)$.

**Lemma 36.** *(Potential increment for covered clusters) For any $t \in \{1, \ldots, k-1\}$, the following holds:*

$$\mathbf{E}[\varphi_{t+1} - \varphi_t \mid \mathbf{F}_t, \{i \in H_t\}] \leq \frac{\text{cost}_t^{\mathscr{L}}(\mathcal{U}_t)}{k - t}$$

*Proof.* When $i \in H_t$, we have $H_{t+1} = H_t$, $\mathcal{W}_{t+1} = \mathcal{W}_t + 1$, and $U_{t+1} = U_t$. Thus,

$$\begin{aligned}
\varphi_{t+1} - \varphi_t &= \frac{W_{t+1}}{|U_{t+1}|}\text{cost}_{t+1}^{\mathscr{L}}(\mathcal{U}_{t+1}) - \frac{W_t}{|U_t|}\text{cost}_t^{\mathscr{L}}(\mathcal{U}_t) \\
&\leq \frac{W_t + 1}{|U_t|}\text{cost}_t^{\mathscr{L}}(\mathcal{U}_t) - \frac{W_t}{|U_t|}\text{cost}_t^{\mathscr{L}}(\mathcal{U}_t) \\
&= \frac{\text{cost}_t^{\mathscr{L}}(\mathcal{U}_t)}{|U_t|} \leq \frac{\text{cost}_t^{\mathscr{L}}(\mathcal{U}_t)}{k - t}
\end{aligned}$$

$\square$

We can now combine the two cases to get:

**Lemma 37** (Combined potential increment bound). *For any $t \in \{1, \ldots, k-1\}$, the following holds:* $\mathbf{E}[\varphi_{t+1} - \varphi_t \mid \mathbf{F}_t] \leq (1 - \delta)\frac{\mathbf{E}[\text{cost}_t^{\mathscr{L}}(\mathcal{H}_t)]}{k - t} + \delta\left(\frac{2}{k-t} + \frac{2t}{\max(1, k-t-1)^2}\right)\text{opt}_1(\mathcal{X})$

*Proof.* To compute the overall expectation, we have:

$$\mathbf{E}[\varphi_{t+1} - \varphi_t \mid \mathbf{F}_t] = \mathbf{E}[\varphi_{t+1} - \varphi_t \mid \mathbf{F}_t, \{i \in U_t\}]\mathbf{Pr}[i \in U_t] + \mathbf{E}[\varphi_{t+1} - \varphi_t \mid \mathbf{F}_t, \{i \in H_t\}]\mathbf{Pr}[i \in H_t]$$

We can bound the first term using Lemma 35:

$$\begin{aligned}
&\mathbf{E}[\varphi_{t+1} - \varphi_t \mid \mathbf{F}_t, i \in U_t]\mathbf{Pr}[i \in U_t] \\
&\leq \mathbf{E}[\varphi_{t+1} - \varphi_t \mid \mathbf{F}_t, i \in U_t] \\
&\leq \frac{2\delta}{1 - \delta}\frac{t}{\max(1, k-t-1)^2}\text{opt}_1(\mathcal{X})
\end{aligned}$$

and the second term using Lemma 36:

$$\begin{aligned}
&\mathbf{E}[\varphi_{t+1} - \varphi_t \mid \mathbf{F}_t, i \in H_t]\mathbf{Pr}[i \in H_t] \\
&\leq \frac{\text{cost}_t^{\mathscr{L}}(\mathcal{U}_t)}{k - t}\left((1 - \delta)\frac{\text{cost}_t^{\mathscr{L}}(\mathcal{H}_t)}{\text{cost}_t^{\mathscr{L}}(\mathcal{X})} + \delta\frac{|\mathcal{H}_t|}{n}\right) \\
&\leq (1 - \delta)\frac{\text{cost}_t^{\mathscr{L}}(\mathcal{H}_t)}{k - t} + \delta\frac{\text{cost}_t^{\mathscr{L}}(\mathcal{X})}{k - t}
\end{aligned}$$

where in the last step we used $\text{cost}_t^{\mathscr{L}}(\mathcal{U}_t) \leq \text{cost}_t^{\mathscr{L}}(\mathcal{X})$ and $|\mathcal{H}_t| \leq n$. Combining both terms completes the proof. $\square$

### C.3.4. FINAL APPROXIMATION GUARANTEE

We are now ready to provide a proof for the main theorem regarding $(\varrho, \delta)$-$k$-means++:

**Theorem 38** (Main approximation guarantee for $(\varrho, \delta)$-$k$-means++). *Let $\mathcal{X} \subset \mathbb{R}^D$ be any dataset which is to be partitioned into $k$ clusters. Let $C$ be the set of centers returned by Algorithm 6 for any $\delta \in (0, 0.5)$ and $\varrho < 1$. The following approximation guarantee holds:*

$$\mathbf{E}[\text{cost}(\mathcal{X}, C)] \in O(\varrho^{-2}\log k)\text{opt}_k(\mathcal{X}) + \delta \cdot O(k + \varrho^{-2}\log k)\text{opt}_1(\mathcal{X})$$

*Proof.* At the end of $k$ iterations, we have $\texttt{cost}(\mathcal{X}, C) \leq \texttt{cost}_k(\mathcal{H}_k) + \texttt{cost}_k^{\mathcal{L}}(\mathcal{U}_k) = \texttt{cost}_k(\mathcal{H}_k) + \varphi_k$. The first term can be bounded using Lemma 34. For the second term, we can express $\varphi_k$ as a telescopic sum:

$$\mathbf{E}[\texttt{cost}(\mathcal{X}, C)]$$

$$\leq \mathbf{E}[\texttt{cost}_k(\mathcal{H}_k)] + \sum_{t=0}^{k-1} \mathbf{E}[\varphi_{t+1} - \varphi_t \mid \mathsf{F}_t]$$

$$\leq \mathbf{E}[\texttt{cost}_k(\mathcal{H}_k)] + \sum_{t=0}^{k-1} (1-\delta) \frac{\mathbf{E}[\texttt{cost}_t^{\mathcal{L}}(\mathcal{H}_t)]}{k-t} + \sum_{t=0}^{k-1} \delta \left( \frac{2}{k-t} + \frac{2t}{(1-\delta)\max(1, k-t-1)^2} \right) \texttt{opt}_1(\mathcal{X})$$

$$\leq \mathbf{E}[\texttt{cost}_k(\mathcal{H}_k)] + \sum_{t=0}^{k-1} \frac{(1-\delta)}{\varepsilon} \frac{\mathbf{E}[\texttt{cost}_t(\mathcal{H}_t)]}{k-t} + \sum_{t=0}^{k-1} \delta \left( \frac{2}{k-t} + \frac{2t}{(1-\delta)\max(1, k-t-1)^2} \right) \texttt{opt}_1(\mathcal{X})$$

$$\leq \varrho^{-1} \left( 8\texttt{opt}_k(\mathcal{X}) + \frac{2\delta}{1-\delta} \texttt{opt}_1(\mathcal{X}) \right) \left( 1 + (1-\delta) \cdot \varepsilon^{-1} \cdot \sum_{t=0}^{k-1} \frac{1}{k-t} \right)$$

$$+ \frac{2\delta}{1-\delta} \texttt{opt}_1(X) \left( k + (1-\delta)(1 + \ln k) + \sum_{t=0}^{k-2} \frac{t}{(k-t-1)^2} \right)$$

To simplify this, note that $\sum_{t=0}^{k-1} \frac{1}{k-t} \leq 1 + \ln k$ and $\sum_{t=0}^{k-2} \frac{t}{(k-t-1)^2} \leq k \sum_{t=1}^{\infty} t^{-2} = k\pi^2/6$.

$$\mathbf{E}[\texttt{cost}(\mathcal{X}, C)]$$

$$\leq \varrho^{-1} \left( 8\texttt{opt}_k(\mathcal{X}) + \frac{2\delta}{1-\delta} \texttt{opt}_1(\mathcal{X}) \right) \left( 1 + \varepsilon^{-1}(1 + \ln k) \right) + 2\delta \texttt{opt}_1(\mathcal{X}) \left( k + 2 + 2\ln k + k\pi^2/3 \right)$$

For sufficiently large $k$, we obtain :

$$\mathbf{E}[\texttt{cost}(\mathcal{X}, C)] \in O(\varrho^{-2} \log k) \texttt{opt}_k(\mathcal{X}) + \frac{\delta}{1-\delta} \cdot O(k + \varrho^{-2} \log k) \texttt{opt}_1(\mathcal{X})$$

This completes the proof of the theorem. $\qquad \square$

### C.4. Rejection Sampling

The naive implementation of $k$-means++ seeding involves iteratively selecting each cluster center using the $D^2$ distribution with respect to the current set of centers. Sampling a point from this distribution naively is expensive as it takes $O(nkd)$ time. To obtain a faster implementation, we sample each center from a distribution that *oversamples* the $D^2$ distribution (but is faster to sample from), and then perform rejection sampling to ensure that the overall distribution remains close to $D^2$. Let us formally define what we mean by *oversampling* a distribution.

**Definition 39.** *($\tau$-oversampling) Suppose $\pi$, $\kappa$ define probability distributions over $\mathcal{X}$. The distribution $\kappa$ is said to $\tau$-oversample $\pi$ for $\tau > 0$ if $\pi(x) \leq \tau \cdot \kappa(x)$ for each $x \in \mathcal{X}$.*

Let $\pi, \kappa$ be probability distributions over $\mathcal{X}$ such that $\kappa$ $\tau$-oversamples $\pi$. Suppose we have a collection of samples $x_1, x_2, \ldots$ from the distribution $D_2$. Consider the following strategy RejectionSample: iterate through samples $\{x_i\}_i$ and terminate if a sample $x_i$ is accepted with probability $\varrho(x_i) = \frac{\pi(x_i)}{\tau \kappa(x_i)}$.

It is not difficult to argue that an accepted sample comes from the distribution $\pi$. Moreover, for any $\varepsilon \in (0, 1)$ it takes at most $\tau \ln(1/\varepsilon)$ samples from $\kappa$ to accept a sample with probability at least $1 - \varepsilon$. Note that the above strategy may not need to know $\tau$ in advance (indeed, computing $\tau$ may be non-trivial), but only requires the ability to compute the quantity $\varrho(x_i)$. In the current form, our strategy does not have any control over the number of samples from $\kappa$ which it may need to examine. However, a bound on the number of samples to be examined can be obtained if we are content with sampling from a slightly perturbed distribution. Suppose we have another distribution $\nu$ over $\mathcal{X}$. This time we are allowed to use samples coming from $\kappa$ and $\nu$ and instead of a sample from $\pi$, we are content with obtaining a sample generated by a hybrid distribution $\pi_\delta(x) = (1-\delta)\pi(x) + \delta\nu(x)$ for some small enough $\delta \in (0, 1)$. For this we can modify the above strategy which we now call RejectionSample($m$): Iterate through $m$ samples $x_1, \ldots, x_m$ from $\kappa$, terminate if a sample $x_i$ is accepted with

probability $\varrho(x_i)$. If no sample is accepted, terminate with a sample from $\nu$. It can be shown that the *failure* probability $\delta$ diminishes with increasing $m$. Indeed, the following holds:

**Lemma 40.** *(Failure probability bound) The failure probability of* RejectionSample$(m)$ *satisfies* $\delta \leq e^{-m/\tau}$.

In our algorithm, we appropriately choose the distributions $\kappa$ and $\nu$ over $\mathcal{X}$ so that $\kappa$ $\tau$-*oversamples* $\pi$ (for a suitable $\tau$) and for which we obtain samples efficiently.

### C.5. Proof of Correctness for Rejection Sampling
We now prove that the rejection sampling procedure indeed produces samples from the desired distribution.

**Lemma 41.** *(Rejection sampling correctness) Let* $\pi, \kappa$ *be probability distributions over* $\mathcal{X}$ *such that* $\kappa$ $\tau$-*oversamples* $\pi$. *If we sample* $x$ *from* $\kappa$ *and accept it with probability* $\varrho(x) = \frac{\pi(x)}{\tau\kappa(x)}$, *then the accepted sample comes from distribution* $\pi$.

*Proof.* For any $x \in \mathcal{X}$, the probability that $x$ is the accepted sample is:

$$\mathbf{Pr}[\text{accept } x] = \mathbf{Pr}[\text{sample } x \text{ from } \kappa] \cdot \mathbf{Pr}[\text{accept} \mid \text{sampled } x]$$

$$= \kappa(x) \cdot \frac{\pi(x)}{\tau\kappa(x)} = \frac{\pi(x)}{\tau}$$

The total probability of accepting any sample is:

$$\mathbf{Pr}[\text{accept}] = \sum_{x \in \mathcal{X}} \mathbf{Pr}[\text{accept } x] = \sum_{x \in \mathcal{X}} \frac{\pi(x)}{\tau} = \frac{1}{\tau}$$

Therefore, the conditional probability that the accepted sample is $x$ is:

$$\mathbf{Pr}[x \mid \text{accept}] = \frac{\mathbf{Pr}[\text{accept } x]}{\mathbf{Pr}[\text{accept}]} = \frac{\pi(x)/\tau}{1/\tau} = \pi(x)$$

Thus, accepted samples follow distribution $\pi$. $\qquad\square$

**Lemma 42.** *(Expected number of samples) Using the rejection sampling procedure with* $\kappa$ $\tau$-*oversampling* $\pi$, *the expected number of samples from* $\kappa$ *needed to obtain one accepted sample is exactly* $\tau$.

*Proof.* Each sample from $\kappa$ is accepted with probability $1/\tau$. Therefore, the number of samples follows a geometric distribution with success probability $p = 1/\tau$, which has expected value $\tau$. $\qquad\square$

**Lemma 43.** *(Bounded rejection sampling) For* RejectionSample$(m)$ *with fallback distribution* $\nu$, *the output distribution is* $(1 - \delta)\pi + \delta\nu$ *where* $\delta \leq e^{-m/\tau}$.

*Proof.* The probability that no sample is accepted after $m$ trials is:

$$\delta = \left(1 - \frac{1}{\tau}\right)^m \leq e^{-m/\tau}$$

where we used the inequality $(1 - x) \leq e^{-x}$ for $x \geq 0$.
With probability $1 - \delta$, a sample is accepted and comes from $\pi$ (by the first lemma). With probability $\delta$, we fallback to sampling from $\nu$. Therefore, the output distribution is $(1 - \delta)\pi + \delta\nu$. $\qquad\square$

### C.6. Using rejection sampling for $Qk$-means++
In the context of our algorithm, we identify the proposal distribution as $\kappa = \kappa(\cdot|S)$, the target distribution as $\pi = \pi(\cdot|S)$ and the fallback distribution as $\nu = \mathrm{uniform}(\mathcal{X})$. We now show that $\kappa$ $\tau$-oversamples $\pi$ for a suitable $\tau$.

**Lemma 44.** *(Oversampling factor) The distribution* $\kappa(\cdot|s_1)$ $\tau$-*oversamples the distribution* $\pi^{\mathscr{L}}(\cdot|S)$ *for*

$$\tau = 2\varrho^{-1} \frac{\|\mathcal{X}\|_{\mathrm{F}}^2 + n\|s_1\|^2}{\mathrm{cost}(\mathcal{X}, S)}$$

*Proof.* The key to this observation is the following use of Cauchy-Schwartz inequality :

$$\texttt{cost}^{\mathscr{L}}(x,S) \leq \varrho^{-1}\texttt{cost}(x,S) \leq \varrho^{-1}\|x - s_1\|^2 \leq 2\varrho^{-1}(\|x\|^2 + \|s_1\|^2)$$

Dividing both sides by $\texttt{cost}^L(\mathcal{X}, S)$, we get

$$\frac{\texttt{cost}^{\mathscr{L}}(x,S)}{\texttt{cost}^{\mathscr{L}}(\mathcal{X},S)} \leq \frac{2\varrho^{-1}(\|x\|^2 + \|s_1\|^2)}{\texttt{cost}^{\mathscr{L}}(\mathcal{X},S)} \leq \frac{2\varrho^{-1}(\|x\|^2 + \|s_1\|^2)}{\texttt{cost}(\mathcal{X},S)}$$

from which we can see

$$\frac{\texttt{cost}^{\mathscr{L}}(x,S)}{\texttt{cost}^{\mathscr{L}}(\mathcal{X},S)} \leq \left(2\varrho^{-1}\frac{\|\mathcal{X}\|_{\mathrm{F}}^2 + n\|s_1\|^2}{\texttt{cost}(\mathcal{X},S)}\right)\frac{\|x\|^2 + \|s_1\|^2}{\|\mathcal{X}\|_{\mathrm{F}}^2 + n\|s_1\|^2}$$

This completes the proof. $\qquad\square$

Finally, let us relate this oversampling factor to the quantity $\beta_k(\mathcal{X}) := \frac{\texttt{opt}_1(\mathcal{X})}{\texttt{opt}_k(\mathcal{X})}$

**Lemma 45.** *For the oversampling factor $\tau$ in Lemma 44, the following hold:*

(i) $\mathbf{E}[\tau] \leq 4\varrho^{-1}\beta_k(\mathcal{X})$

(ii) $\mathbf{Pr}\left[\tau > 10\varrho^{-1}\beta_k(\mathcal{X})\right] \leq \dfrac{1}{4}$

*Proof.* Since $s_1 \sim \text{UNIFORM}(\mathcal{X})$, we have $\mathbf{E}[\|s_1\|^2] = \frac{1}{n}\|\mathcal{X}\|_{\mathrm{F}}^2$. From Lemma 44, the oversampling factor satisfies:

$$\tau \leq \frac{2\varrho^{-1}\left(\|\mathcal{X}\|_{\mathrm{F}}^2 + n\|s_1\|^2\right)}{\texttt{opt}_k(\mathcal{X})}$$

*Proof of (i).* Taking expectations:

$$\begin{aligned}
\mathbf{E}[\tau] &\leq 2\varrho^{-1}\frac{\|\mathcal{X}\|_{\mathrm{F}}^2 + n\mathbf{E}[\|s_1\|^2]}{\texttt{opt}_k(\mathcal{X})} \\
&= 2\varrho^{-1}\frac{\|\mathcal{X}\|_{\mathrm{F}}^2 + n \cdot \frac{1}{n}\sum_{x \in \mathcal{X}}\|x\|^2}{\texttt{opt}_k(\mathcal{X})} \\
&= \frac{4\varrho^{-1}\|\mathcal{X}\|_{\mathrm{F}}^2}{\texttt{opt}_k(\mathcal{X})} = \frac{4\varrho^{-1}\texttt{opt}_1(\mathcal{X})}{\texttt{opt}_k(\mathcal{X})} = 4\varrho^{-1}\beta_k(\mathcal{X})
\end{aligned}$$

where the last equality uses the fact that $\mathcal{X}$ is translated so that its centroid is at the origin, giving $\|\mathcal{X}\|_{\mathrm{F}}^2 = \texttt{opt}_1(\mathcal{X})$.
*Proof of (ii).* Define the event $\mathcal{E} := \left\{\|s_1\|^2 \leq \frac{4}{n}\|\mathcal{X}\|_{\mathrm{F}}^2\right\}$. By Markov's inequality:

$$\mathbf{Pr}\left[\|s_1\|^2 > \frac{4}{n}\|\mathcal{X}\|_{\mathrm{F}}^2\right] \leq \frac{\mathbf{E}[\|s_1\|^2]}{\frac{4}{n}\|\mathcal{X}\|_{\mathrm{F}}^2} = \frac{\frac{1}{n}\|\mathcal{X}\|_{\mathrm{F}}^2}{\frac{4}{n}\|\mathcal{X}\|_{\mathrm{F}}^2} = \frac{1}{4}$$

so $\mathbf{Pr}[\mathcal{E}] \geq 3/4$. Conditioned on $\mathcal{E}$:

$$\tau \leq \frac{2\varrho^{-1}\left(\|\mathcal{X}\|_{\mathrm{F}}^2 + n \cdot \frac{4}{n}\|\mathcal{X}\|_{\mathrm{F}}^2\right)}{\texttt{opt}_k(\mathcal{X})} = \frac{10\varrho^{-1}\|\mathcal{X}\|_{\mathrm{F}}^2}{\texttt{opt}_k(\mathcal{X})} = 10\varrho^{-1}\beta_k(\mathcal{X})$$

Therefore $\mathbf{Pr}\left[\tau > 10\varrho^{-1}\beta_k(\mathcal{X})\right] \leq \mathbf{Pr}[\mathcal{E}^c] \leq \dfrac{1}{4}$. $\qquad\square$

### C.7. Proof of main theorem

We finally need to combine all the results from the previous sections to prove Theorem 6, which claims existence of a fast rejection-sampling based $k$-means++ while giving an approximation bound of the clustering from the optimal one. Corollary 8 will then follow as a corollary, together with Assumption 1 and Theorem 5.

Let us see how QKMEANS($\mathcal{X}, k, m$) (Algorithm 3) satisfies the claims made by Theorem 6. As seen in Subsection C.1, the preprocessing step takes a time of $O(nD)$. Let us now see how it helps in fast sampling from the distribution $\kappa(\cdot|C)$ which is step 4 of SAMPLE($\mathscr{L}, C, m$).

Here is how this step can be performed:

---

**Procedure 7** `SampleDistribution`

---

**Input:** A set of centers $C \subseteq \mathcal{X}$

**Output:** A sample according to the distribution $\kappa(\cdot|C)$ defined as $\kappa(x|C) = \frac{\|x\|^2 + \|c_1\|^2}{\|\mathcal{X}\|^2 + n\|c_1\|^2}$

1: Generate $r \sim \text{UNIFORM}([0,1])$
2: **if** $r \leq \frac{\|\mathcal{X}\|_\text{F}^2}{\|\mathcal{X}\|_\text{F}^2 + n\|c_1\|^2}$ **then**
3:     Generate a sample $x$ with probability $\kappa(x)$
4: **else**
5:     Generate a sample $x \sim \text{UNIFORM}(\mathcal{X})$
6: **end if**
7: **output** $x$

---

**Lemma 46.** *Procedure 7 produces a sample from $\mathcal{X}$ according to the distribution $\kappa(\cdot|C)$. Moreover, it takes $O(\log n)$ time after a one time preprocessing time of $O(nD)$.*

*Proof.* The probability of a sampled point is as follows:

$$
\begin{aligned}
\mathbf{Pr}[x] &= \mathbf{Pr}\left[r \leq \frac{\|\mathcal{X}\|_\text{F}^2}{\|\mathcal{X}\|_\text{F}^2 + n\|c_1\|^2}\right] \mathbf{Pr}[x \sim \kappa(\cdot)] \\
&\quad + \mathbf{Pr}\left[r > \frac{\|\mathcal{X}\|_\text{F}^2}{\|\mathcal{X}\|_\text{F}^2 + n\|c_1\|^2}\right] \mathbf{Pr}[x \sim \text{UNIFORM}(\mathcal{X})] \\
&= \frac{\|\mathcal{X}\|_\text{F}^2}{\|\mathcal{X}\|_\text{F}^2 + n\|c_1\|^2} \frac{\|x\|^2}{\|\mathcal{X}\|_\text{F}^2} + \frac{n\|c_1\|^2}{\|\mathcal{X}\|_\text{F}^2 + n\|c_1\|^2} \frac{1}{n} \\
&= \frac{(\|x\|^2 + \|c_1\|^2)}{(\|\mathcal{X}\|_\text{F}^2 + n\|c_1\|^2)} = \kappa(x|C)
\end{aligned}
$$

The time complexity follows from Corollary 25. $\qquad\square$

Next, let us recall from Lemma 26 that the LSH-based ANNS data structure supports (conditioned on success with probability $1 - 1/t$) a $\texttt{Insert}(\mathcal{L}, p)$ operation in time $O\left(D \log \eta (t \log \eta)^\varrho\right)$ and a $\texttt{Query}(\mathcal{L}, p)$ operation in time $O\left(D \log \eta (t \log \eta)^\varrho\right)$, where $t$ is the size of the set of points $P$ to be inserted. In our case, the $\text{QKMEANS}(\mathcal{X}, k, m)$ algorithm only inserts the chosen cluster centers in the dataset, i.e. $t = k$ (Note that the $\texttt{Query}(\mathcal{L}, p)$ operation can be made for any $p \in \mathcal{X}$).
Also, recall that here, $D$ is the dimension of the dataset after applying the linear transformation from (Makarychev et al., 2019), so that $D = O(\log k)$, with a constant distortion in the clustering cost w.r.t. any centers.
Now, observe that the INSERT operation is invoked $k$ times, and the SAMPLE subroutine is invoked $k$ times from the QKMEANS algorithm. Within the SAMPLE subroutine, a QUERY operation and a sampling from $\kappa(\cdot|C)$ operation are invoked atmost $m \ln k$ times. This means that there are a total of $k$ INSERT operations, $mk \ln k$ QUERY operations and $mk \ln k$ sampling operations from $\kappa(\cdot|C)$.
Thus we obtain a time complexity (after the processing step) of

$$
O(kD \log \eta (t \log \eta)^\varrho) + O(mk(\ln k)D \log \eta (t \log \eta)^\varrho) + O(mk \ln k \log n)
$$

which after substituting values of $D, t$ can be simplified to $O(m(\log^2 k)(k \log \eta)^{1+\varrho} + mk \ln k \log n)$. This completes the time complexity analysis.
Now, let us analyse the approximation guarantee. From Theorem 38, the $(\varrho, \delta)$-$k$-means++ follows an approximation guarantee of:

$$
\mathbf{E}[\text{cost}(\mathcal{X}, C)] \in O(\varrho^{-2} \log k)\text{opt}_k(\mathcal{X}) + \delta \cdot O(k + \varrho^{-2} \log k)\text{opt}_1(\mathcal{X})
$$

where $\delta$ is the probability of choosing a center via the uniform distribution (which happens due to the failure of rejection sampling). It remains to bound $\delta$. From Lemma 43 and Lemma 44, we have $\delta \leq e^{-m \ln k/\tau}$ where $\tau = 2\varrho^{-1}\beta(\mathcal{X}, S)$. By part (ii) of Lemma 45, we have $\tau \leq 10\varrho^{-1}\beta_k(\mathcal{X})$ with probability at least $3/4$, where this bound is independent of the centers $S$ chosen by the algorithm. Let $c = \lceil \log(\text{opt}_1(\mathcal{X})/\text{opt}_k(\mathcal{X})) \rceil = \lceil \log \beta_k(\mathcal{X}) \rceil$. We repeat the entire seeding procedure $c$ independent times and return the best clustering. Since $\mathbf{Pr}\left[\tau \leq 10\varrho^{-1}\beta_k(\mathcal{X})\right] \geq 3/4$ independently across runs, at least one run satisfies the bound with probability at least $1 - \left(\frac{1}{4}\right)^c$. Recalling that we always have $\mathbf{E}[\text{cost}(\mathcal{X}, S)] \leq 2\text{opt}_1(\mathcal{X})$

since the first center is chosen uniformly at random, we can write the final expected cost, after taking the best clustering over $c$ repetitions as $\mathbf{E}[\texttt{cost}(\mathcal{X}, S)] \leq (1 - 4^{-c})\mathbf{E}[\texttt{cost}(\mathcal{X}, C)|\tau \leq 10\varrho^{-1}\beta_k(\mathcal{X})] + 4^{-c} \cdot 2\texttt{opt}_1(\mathcal{X})$. By our choice of $c$, this only adds a factor of $O(1)\texttt{opt}_k(\mathcal{X})$ to the approximation guarantee.

Thus the total expected cost satisfies:

$$\mathbf{E}[\texttt{cost}(\mathcal{X}, C)] \in \underbrace{O(\varrho^{-2})\log k \cdot \texttt{opt}_k(\mathcal{X})}_{k\text{-means++ guarantee}} + \underbrace{O(\varrho^{-2}) \cdot k^{-\Omega(\varrho m/\beta_k(\mathcal{X}))} \cdot \texttt{opt}_1(\mathcal{X})}_{\text{exponentially decaying additive term}}$$

which completes the proof of Theorem 6.

Corollary 7 follows as a corollary from Theorem 6; if we allow $m = \infty$ in $\textsc{Sample}(\mathscr{L}, C, m)$, we can see that the 'break' statement will be issued in expected time $O(\tau)$, since we need $\tau = 2\varrho^{-1}\beta(\mathcal{X}, C)$, $\mathbf{E}[\tau] \leq 4\varrho^{-1}\beta_k(\mathcal{X})$ (from Lemma 45) samples of distribution $\kappa(\cdot|C)$ in expectation to get a desired sample by Lemma 42.

Finally, Corollary 8 can be obtained as a corollary of Corollary 7 assuming the manifold hypothesis. Theorem 5 gives us $\beta_k(\mathcal{X}) \leq \beta_0 k^\varepsilon, \eta(\mathcal{X}) \leq \eta_0 n^{3\varepsilon/2}$ for quantization exponent $\varepsilon$. Substituting these in the time complexity bound of Corollary 7 gives Corollary 8, keeping in mind that we always have $\varepsilon \in O(1)$.

## D. Empirical Study

In this section, we empirically evaluate the extent to which the theoretical predictions developed in this paper are reflected in real-world data.

**Compute environment.** All experiments were conducted on a dual-socket machine equipped with IntelXeonGold5220R CPUs (48 physical cores total, 2.20 GHz) and 96 hardware threads, running Linux on x86-64 architecture. Unless otherwise stated, all experiments were executed in a CPU-only setting with a fixed number of threads. Implementations were compiled with aggressive compiler optimizations, and all reported runtimes are wall-clock times averaged over multiple random seeds.

**Datasets.** We use datasets across several domains for our empirical evaluation. We include popular image datasets, both as raw pixels as well as image embeddings which are produced through joint vision-language models such as CLIP-ViT-B/32; text datasets in the form of text embeddings generated by the All-MiniLM-L6-v2 model; tabular datasets which record sensor data as well as data generated by scientific experiments. The details of the datasets used are presented in Table 2.

| CATEGORY | DATASET | $n$ | $d$ |
|---|---|---|---|
| RAW IMAGES | MNIST | 60,000 | 784 |
| | FMNIST | 60,000 | 784 |
| | CIFAR10 | 50,000 | 3,072 |
| | CIFAR100 | 50,000 | 3,072 |
| CLIP VIT-B/32 IMAGE EMBEDDINGS | MNIST | 60,000 | 512 |
| | FMNIST | 60,000 | 512 |
| | CIFAR10 | 50,000 | 512 |
| | CIFAR100 | 50,000 | 512 |
| ALL-MINILM-L6-V2 TEXT EMBEDDINGS | REDDIT | 100,000 | 384 |
| | STACKEXCHANGE | 20,000 | 384 |
| TABULAR | SUSY | 5,000,000 | 18 |
| | HAR | 10,299 | 561 |

### D.1. Scaling Laws

We test the hypothesis that the data-dependent parameters $\beta_k = \mathtt{opt}_1/\mathtt{opt}_k$ and $\eta_k$ grow polynomially with the number of clusters $k$, as predicted by optimal quantization theory. For each dataset, we compute $k$-means clusterings for $k \in \{5, 10, 50, 100, 250, 500, 750, 1000\}$ using 10 independent runs. We fit linear regressions to $\log \beta_k$ and $\log \eta_k$ versus $\log k$ and report slopes, $R^2$, and 95% confidence intervals. Across datasets and modalities, both quantities exhibit reasonable linear behavior in log–log coordinates. The plots are shown in Figures 7 and 8. Table 4 provides the estimated quantization exponents and $R^2$ scores for the best fit estimates. The high $R^2$ scores ($\gtrsim 0.95$) show that the scaling laws fit well on the data.

### D.2. Intrinsic Dimension

We test the hypothesis that the empirically estimated quantization exponent $\widehat{\varepsilon}$ is inversely proportional to the intrinsic dimension. Independent intrinsic dimension estimates $\widehat{d}_{\mathrm{MLE}}$ are obtained by using the popular maximum likelihood method (MLE) (Levina & Bickel, 2005). For a dataset $\mathcal{X} = \{x_1, \ldots, x_n\}$, let $T_j(x_i)$ denote the distance of the $j$th nearest neighbor of $x_i$ from $x_i$. For a hyperparameter $k$, the ID estimate is calculated as :

$$\widehat{d}_{\mathrm{MLE}}(\mathcal{X}, k) := \frac{1}{n} \sum_{i=1}^{n} \left( \frac{1}{k-1} \sum_{j=1}^{k-1} \log \frac{T_k(x_i)}{T_j(x_i)} \right)^{-1}$$

For each dataset we take 10 independent samples each of size $n' = 10{,}000$ to estimate the ID for a particular value of $k$. This is repeated for $k \in \{5, 10, 20, 50, 100\}$ and the average is reported. We plot the estimates $\widehat{\varepsilon}$ of the quantization exponent against $\widehat{d}_{\mathrm{MLE}}$ in Figure 5, and the estimated dimension $\hat{d}_\varepsilon = 2/\hat{\varepsilon}$ against $\hat{d}_{\mathrm{MLE}}$. We observe significant agreement with Pearson's $R = 0.92$ and an $R^2$ score of 0.84. It is to be noted that ID estimation in itself is a challenging problem with several proposed estimators in the literature. We refer the reader to the surveys (Camastra & Staiano, 2016; Binnie et al.,

*Table 4.* Scaling law parameters across datasets. $\varepsilon$ is estimated from $\beta_k \sim k^\varepsilon$. $R^2_\beta$ and $R^2_\eta$ are the coefficients of determination for the $\beta$ and $\eta$ scaling fits respectively.

| Dataset | $\hat{\varepsilon}$ | $\hat{R}^2_\beta$ | $\hat{R}^2_\eta$ |
|---|---|---|---|
| MNIST | 0.1449 | 0.9998 | 0.9944 |
| FASHION-MNIST | 0.1808 | 0.9847 | 0.9702 |
| CIFAR-10 | 0.0862 | 0.9945 | 0.9661 |
| CIFAR-100 | 0.0961 | 0.9919 | 0.9659 |
| MNIST-CLIP | 0.1759 | 0.9980 | 0.9981 |
| FMNIST-CLIP | 0.1665 | 0.9887 | 0.9809 |
| CIFAR-10-CLIP | 0.1101 | 0.9898 | 0.9737 |
| CIFAR-100-CLIP | 0.1166 | 0.9988 | 0.9938 |
| REDDIT | 0.0622 | 0.9899 | 0.9814 |
| STACKEXCHANGE | 0.0565 | 0.9826 | 0.9898 |
| HAR | 0.1894 | 0.9984 | 0.8791 |
| SUSY | 0.2644 | 0.9993 | 0.9693 |

*Table 5.* Comparison of intrinsic dimension estimates. $\varepsilon$ is the scaling exponent from $\beta \sim k^\varepsilon$. $d_\varepsilon = 2/\varepsilon$ is the intrinsic dimension estimate from the scaling law. $d_{\text{MLE}}$ is the maximum likelihood estimate of intrinsic dimension from (Levina & Bickel, 2005).

| Dataset | $\varepsilon$ | $\hat{d}_\varepsilon$ | $\hat{d}_{\text{MLE}}$ |
|---|---|---|---|
| MNIST | 0.1449 | 13.80 | 14.25 |
| FASHION-MNIST | 0.1808 | 11.06 | 15.64 |
| CIFAR-10 | 0.0862 | 23.21 | 28.39 |
| CIFAR-100 | 0.0961 | 20.81 | 25.69 |
| MNIST-CLIP | 0.1759 | 11.37 | 14.43 |
| FMNIST-CLIP | 0.1665 | 12.01 | 16.22 |
| CIFAR-10-CLIP | 0.1101 | 18.16 | 22.05 |
| CIFAR-100-CLIP | 0.1166 | 17.15 | 20.13 |
| REDDIT | 0.0622 | 32.13 | 29.22 |
| HAR | 0.1894 | 10.56 | 14.75 |
| SUSY | 0.2644 | 7.56 | 7.88 |
| STACKEXCHANGE | 0.0565 | 35.40 | 28.48 |

2025) for further discussion on ID estimators. We have chosen the MLE estimator of (Levina & Bickel, 2005) since it is one of the most established and widely studied estimators in the literature.

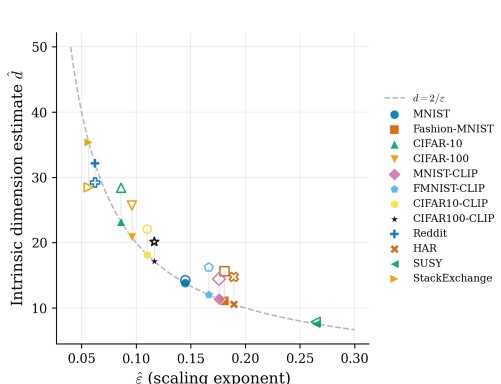

*Figure 5.* Relationship between $\varepsilon$ (from $\beta$ scaling) and $d_{\text{MLE}}$.

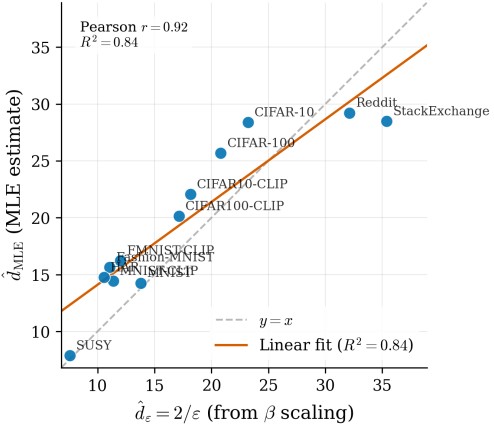

*Figure 6.* Comparison of $d_\varepsilon$ and $d_{\text{MLE}}$.

### D.3. Effect of manifold stucture

To assess the robustness of the quantization scaling law $\beta_k \sim k^\varepsilon$ under distributional perturbation, we inject isotropic Gaussian noise at 20 linearly-spaced noise-to-signal ratios (NSR) from 0 to 2 across four datasets (MNIST, MNIST-CLIP, HAR, STACKEXCHANGE), where $\mathcal{X}_{\text{noisy}} = X + \text{NSR} \cdot \sigma(\mathcal{X}) \cdot \mathcal{N}(0,1)$, and re-fit the power-law exponent $\varepsilon$ at each level. Across all datasets, $\varepsilon$ decays monotonically and convexly toward $2/D$, with the majority of degradation occurring at low noise levels (NSR $< 0.5$), after which the exponent plateaus well above the ambient-dimension limit $2/D$. The rate of decay varies with intrinsic structure: compact, low-dimensional representations such as MNIST-CLIP ($d \approx 11$, $D = 512$) lose most of their scaling signal by NSR $\approx 0.5$, while higher-dimensional datasets like StackExchange ($d \approx 39$, $D = 384$) degrade more gradually. This confirms that the scaling exponent captures genuine geometric structure rather than statistical artifacts, and suggests that $\varepsilon$ could serve as a noise diagnostic — its sensitivity to perturbation is itself informative about the signal-to-noise regime of the data. The plots are shown in Figure 10.

### D.4. Rejection sampling analysis

Rejection Rate Experiment. We investigate the rejection sampling behavior of QKMEANS across 12 datasets by varying the chain length $m \in \{1, 2, 3, 5, 7, 10, 15, 20, 30, 50\}$ and number of clusters $k \in \{10, 50, 100\}$, averaging over 5 independent

runs. For each center selection step, QKMEANS draws up to $m \ln(k + 1)$ proposals from the distribution $\kappa(x) \propto \|x\|^2 + \|c_1\|^2$ and accepts a proposal with probability proportional to its distance to the nearest existing center; if all proposals are rejected, the algorithm falls back to uniform random selection. Low-dimensional and well-separated datasets (MNIST, FashionMNIST, SUSY) exhibit low rejection even at $m = 1$ and reach zero rejection by $m \approx 5$–10. High-dimensional raw-pixel datasets (CIFAR-10, CIFAR-100) show moderate rejection at small $m$ ($\sim$20–30) but converge to zero by $m \approx 60$–80. CLIP embeddings present the most challenging setting: FMNIST-CLIP and MNIST-CLIP maintain elevated rejection rates ($\sim$75–90 at $m = 1$), with MNIST-CLIP retaining $\sim$10–15 rejection even at $m = 50$, reflecting the tighter concentration of norm-based proposal distributions in these embedding spaces. In contrast, text embedding datasets (StackExchange, Reddit) exhibit near-zero rejection across all $m$ values, indicating that their $\kappa$ distribution closely approximates the $D^2$ distribution. Across all datasets, larger $k$ generally yields higher rejection fractions at a given $m$, as later center selections must distinguish among increasingly fine-grained cluster structure. These results suggest that $m \approx 10$–20 suffices for most practical settings, while highly concentrated embedding datasets may benefit from larger chain lengths. The plots are shown in Figure 11.

### D.5. Effect of ANNS used

Although our theoretical analysis uses Locality Sensitive Hashing for implementing nearest neighbor search, there are several other methods for doing the same. Our algorithm uses the ANNS algorithm as a black-box and hence, it can be easily replaced by a different method. We test several ANNS methods on the datasets to study the runtime-solution quality tradeoffs. We include brute-force search, hierarchicaly navigable small worlds (Malkov & Yashunin, 2020), locality sensitive hasing (Indyk & Motwani, 1998), inverted file index (Jégou et al., 2011b) and random projection trees (Dasgupta & Freund, 2008).For small $k \approx 50$, Qkmeans is fastest with brute-force nearest-neighbor search, which outperforms LSH and tree-based ANNS by 2–4$\times$ even in very high dimensions, since ANNS overhead dominates at low k. As k increases, this trend reverses: tree-based methods (e.g., RP-trees, HNSW) become increasingly advantageous by amortizing query costs across many seeding steps. The results are in Table 6.

### D.6. Comparison with Prior Art

In our comparative analysis, we consider the following algorithms :

- `AFKMC`$^2$ from (Bachem et al., 2016a)

- `RejectionSampling` from (Cohen-Addad et al., 2020a)

- `PRONECoreset` from (Charikar et al., 2023)

- `FastKMeans` from (Draganov et al., 2024)

We followed the reference implementations from https://github.com/adriangoe/afkmc2 for `AFKMC`$^2$, from https://git.informatik.uni-hamburg.de/0moin/google-research/-/tree/masterfor `RejectionSampling`, from https://github.com/boredoms/prone for `PRONECoreset` and from https://github.com/Andrew-Draganov/Fast-Coreset-Generation for `FastKMeans`. We use the default hyperparameters which are specified in these implementations. For implementing Qkmeans, we use the HNSW ANNS algorithm.

Figure 9 shows the seeding time and seeding quality tradeoffs for various algorithms across datasets. These results are also tabulated in Tables 7-18.

### D.7. Summary of Empirical Findings

Across all datasets, we observe consistent evidence for quantization-theoretic scaling, a strong inverse relationship between intrinsic dimension and scaling exponents, and stable rejection-sampling behavior. These results collectively support the central premise that geometric structure governs the empirical behavior of fast $k$-means seeding algorithms.

*Table 6.* QKMEANS seeding time (ms) with different ANNS methods across all datasets and $k$ values.

| DATASET | $k$ | HNSW | BRUTEFORCE | LSH | IVF | RPTREE |
|---|---|---|---|---|---|---|
| MNIST ($d = 784$) | 50 | 12.6 | 9.4 | 16.8 | 10.5 | 12.7 |
| | 200 | 43.3 | 24.8 | 46.6 | 22.1 | 23.9 |
| | 1000 | 526.6 | 544.0 | 693.1 | 348.8 | 93.6 |
| FASHION-MNIST ($d = 784$) | 50 | 14.8 | 8.8 | 19.4 | 9.1 | 12.5 |
| | 200 | 73.1 | 48.4 | 99.0 | 46.2 | 43.3 |
| | 1000 | 852.6 | 1506.8 | 1961.1 | 1256.3 | 233.6 |
| CIFAR-10 ($d = 3072$) | 50 | 36.1 | 31.4 | 113.1 | 40.6 | 60.6 |
| | 200 | 341.5 | 301.6 | 681.5 | 327.4 | 231.3 |
| | 1000 | 5242.8 | 8899.1 | 10951.5 | 8596.0 | 1396.1 |
| CIFAR-100 ($d = 3072$) | 50 | 31.8 | 29.3 | 105.3 | 33.4 | 52.3 |
| | 200 | 264.6 | 241.6 | 556.7 | 253.8 | 189.6 |
| | 1000 | 4187.5 | 7065.1 | 8738.9 | 7012.6 | 1156.1 |
| MNIST-CLIP ($d = 512$) | 50 | 33.0 | 15.6 | 78.4 | 23.8 | 34.5 |
| | 200 | 405.5 | 193.5 | 878.2 | 233.9 | 207.6 |
| | 1000 | 4702.3 | 6664.2 | 26362.4 | 6985.7 | 1052.7 |
| FMNIST-CLIP ($d = 512$) | 50 | 21.9 | 10.6 | 28.4 | 11.9 | 16.8 |
| | 200 | 155.7 | 76.6 | 299.7 | 123.7 | 72.2 |
| | 1000 | 1828.8 | 2510.2 | 4885.6 | 2689.8 | 387.9 |
| CIFAR10-CLIP ($d = 512$) | 50 | 11.8 | 5.5 | 16.2 | 11.8 | 13.2 |
| | 200 | 73.0 | 37.9 | 65.5 | 38.9 | 27.9 |
| | 1000 | 735.5 | 825.2 | 960.7 | 856.4 | 141.6 |
| CIFAR100-CLIP ($d = 512$) | 50 | 13.9 | 7.2 | 13.0 | 7.2 | 10.4 |
| | 200 | 59.6 | 31.3 | 52.7 | 30.4 | 24.8 |
| | 1000 | 695.4 | 707.8 | 816.7 | 742.9 | 120.4 |
| REDDIT ($d = 384$) | 50 | 9.5 | 4.1 | 6.4 | 5.0 | 6.1 |
| | 200 | 21.2 | 12.2 | 14.4 | 9.7 | 9.7 |
| | 1000 | 221.7 | 93.9 | 40.8 | 96.1 | 26.2 |
| STACKEXCHANGE ($d = 384$) | 50 | 7.6 | 3.9 | 7.5 | 5.9 | 5.0 |
| | 200 | 24.0 | 10.2 | 13.9 | 12.6 | 9.5 |
| | 1000 | 225.4 | 84.8 | 39.1 | 87.6 | 24.4 |
| HAR ($d = 561$) | 50 | 19.6 | 8.6 | 37.0 | 13.5 | 20.2 |
| | 200 | 171.8 | 107.8 | 212.9 | 113.1 | 87.5 |
| | 1000 | 2186.7 | 3958.6 | 4748.6 | 3947.5 | 556.0 |
| SUSY ($d = 19$) | 50 | 6.5 | 2.5 | 3.4 | 1.5 | 3.0 |
| | 200 | 35.8 | 6.1 | 12.1 | 6.7 | 5.3 |
| | 1000 | 370.5 | 68.3 | 85.7 | 81.6 | 40.0 |

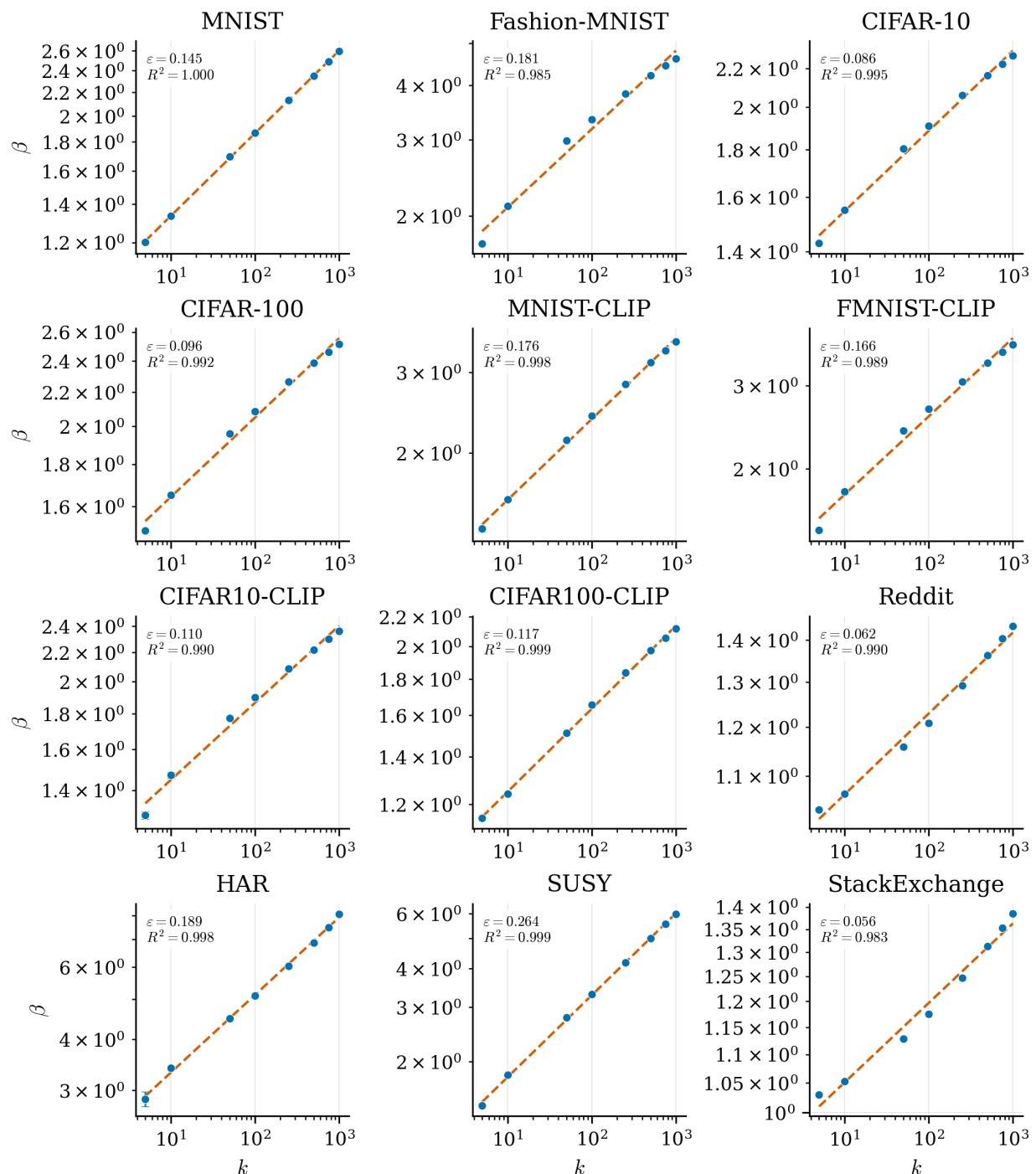

*Figure 7.* Scaling Laws for $\beta_k$ vs $k$

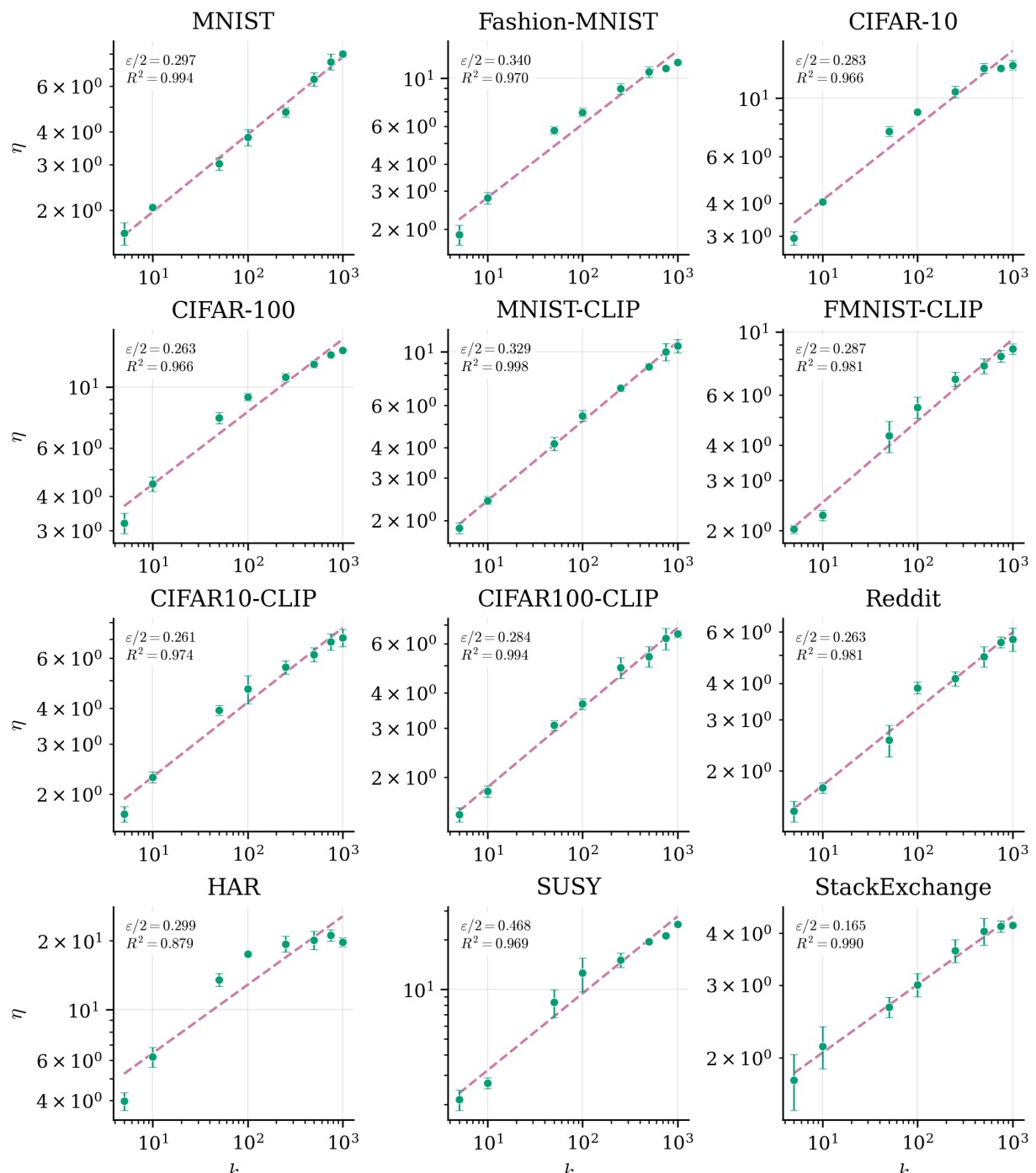

*Figure 8.* Scaling Laws for $\eta_k$ vs $k$

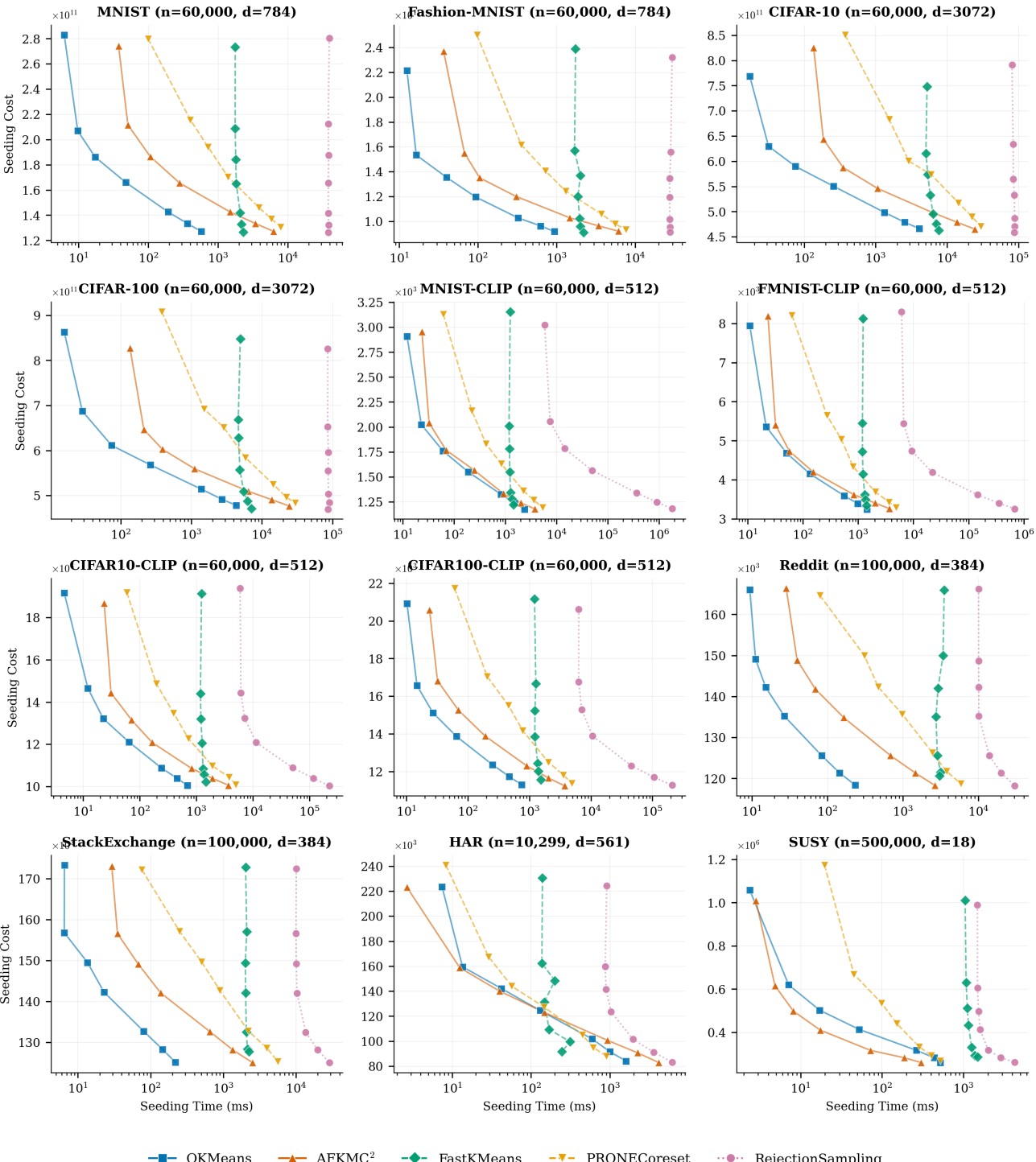

*Figure 9.* Plots for seeding time and seeding quality tradeoffs for various algorithms

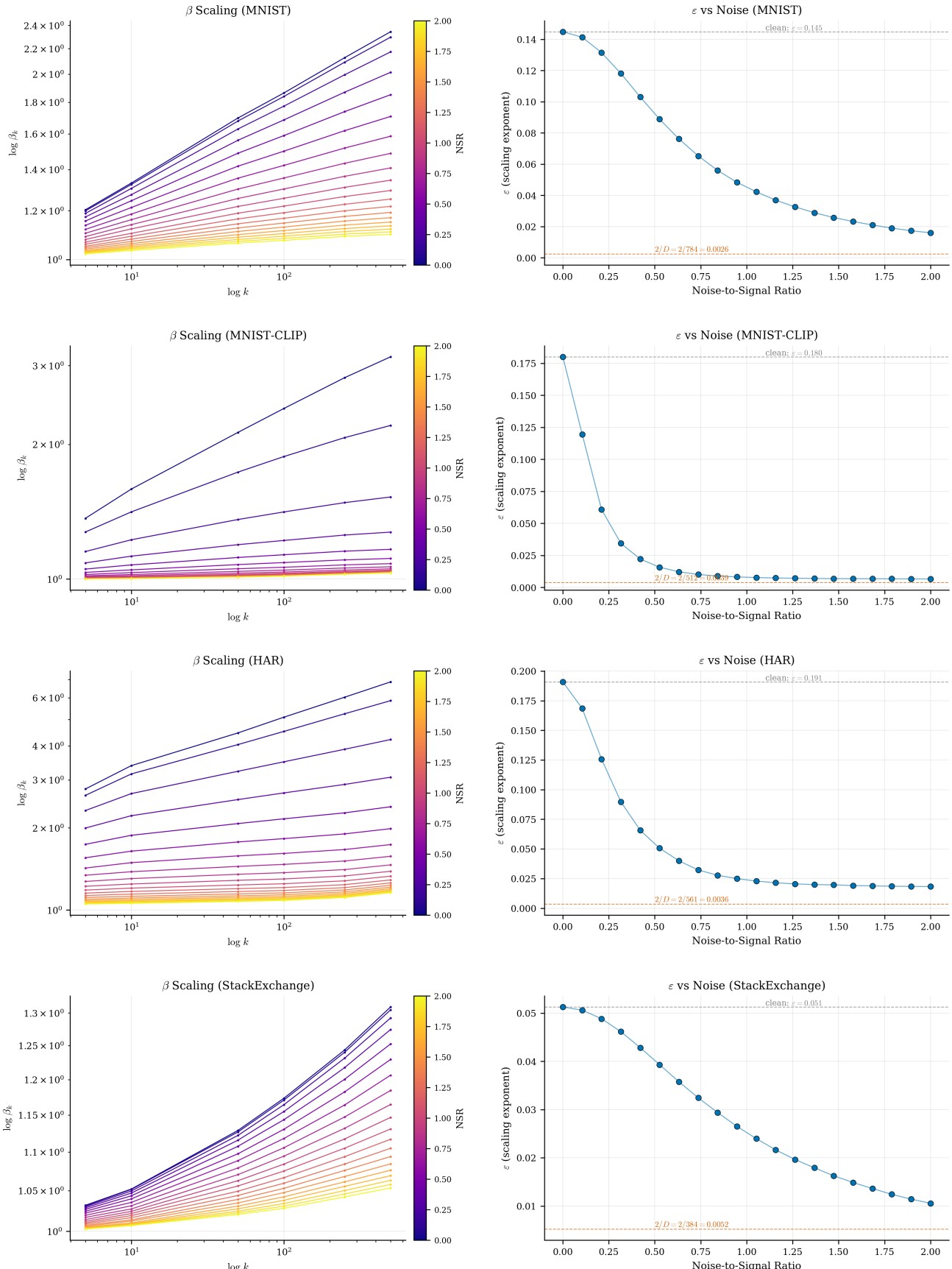

*Figure 10.* Effect of noise on the scaling laws, demonstrated on 4 datasets

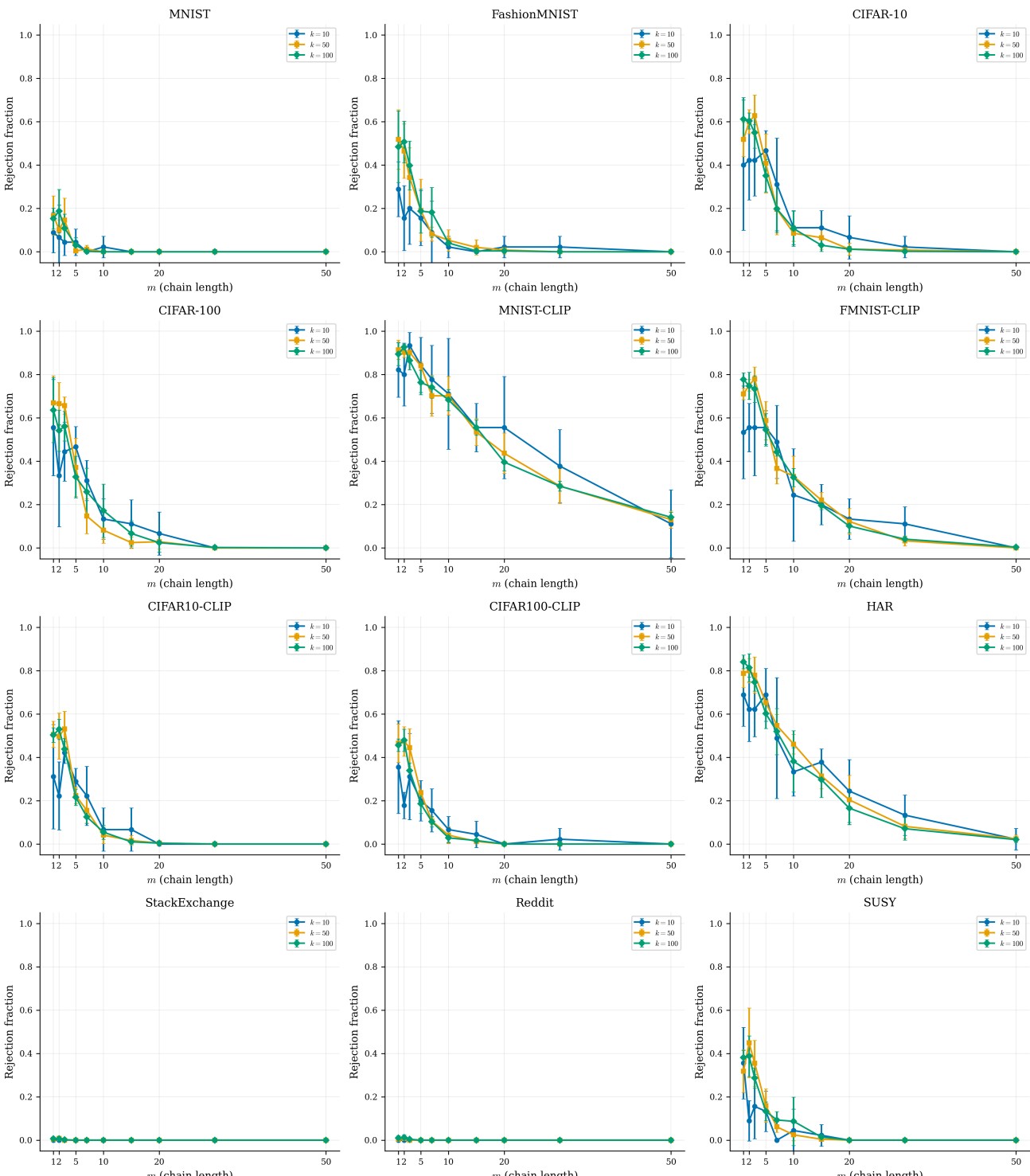

*Figure 11.* Fraction of samples rejected vs the chain length $m$

*Table 7.* Seeding benchmark on mnist. Time in ms, cost $\times 10^{11}$.

| Algorithm | $k = 10$ | | $k = 50$ | | $k = 100$ | | $k = 200$ | | $k = 500$ | | $k = 750$ | | $k = 1000$ | |
|---|---|---|---|---|---|---|---|---|---|---|---|---|---|---|
| | Time | Cost | Time | Cost | Time | Cost | Time | Cost | Time | Cost | Time | Cost | Time | Cost |
| AFKMC$^2$ | 38 | 2.74 | 51 | 2.12 | 107 | 1.86 | 281 | 1.65 | 1.5s | 1.43 | 3.4s | 1.33 | 6.2s | 1.27 |
| PRONECoreset | 99 | 2.80 | 400 | 2.16 | 719 | 1.94 | 1.4s | 1.70 | 3.9s | 1.46 | 5.8s | 1.37 | 7.9s | 1.31 |
| FastKMeans | 1.8s | **2.73** | 1.8s | 2.09 | 1.8s | **1.84** | 1.8s | **1.65** | 2.1s | 1.42 | 2.2s | 1.33 | 2.3s | 1.27 |
| RejectionSampling | 39.2s | 2.80 | 37.8s | 2.12 | 38.3s | 1.88 | 38.0s | 1.65 | 37.8s | **1.41** | 38.3s | **1.32** | 38.0s | **1.26** |
| Qkmeans | **6.2** | 2.83 | **9.7** | **2.07** | **17** | 1.86 | **48** | 1.66 | **192** | 1.43 | **365** | 1.33 | **573** | 1.27 |

*Table 8.* Seeding benchmark on fmnist. Time in ms, cost $\times 10^{11}$.

| Algorithm | $k = 10$ | | $k = 50$ | | $k = 100$ | | $k = 200$ | | $k = 500$ | | $k = 750$ | | $k = 1000$ | |
|---|---|---|---|---|---|---|---|---|---|---|---|---|---|---|
| | Time | Cost | Time | Cost | Time | Cost | Time | Cost | Time | Cost | Time | Cost | Time | Cost |
| AFKMC$^2$ | 37 | 2.37 | 67 | 1.55 | 105 | 1.35 | 308 | 1.20 | 1.5s | 1.03 | 3.4s | 0.96 | 6.1s | 0.92 |
| PRONECoreset | 97 | 2.50 | 356 | 1.62 | 722 | 1.41 | 1.3s | 1.24 | 3.7s | 1.06 | 5.6s | 0.98 | 7.7s | 0.93 |
| FastKMeans | 1.7s | 2.39 | 1.7s | 1.57 | 2.0s | 1.37 | 1.9s | 1.20 | 2.0s | 1.02 | 2.0s | 0.96 | 2.2s | **0.91** |
| RejectionSampling | 29.6s | 2.32 | 28.5s | 1.56 | 27.6s | **1.35** | 27.6s | **1.19** | 27.7s | **1.02** | 27.9s | **0.95** | 27.9s | 0.91 |
| Qkmeans | **13** | **2.21** | **16** | **1.53** | **40** | 1.35 | **94** | 1.20 | **325** | 1.03 | **627** | 0.96 | **941** | 0.92 |

*Table 9.* Seeding benchmark on cifar10. Time in ms, cost $\times 10^{11}$.

| Algorithm | $k = 10$ | | $k = 50$ | | $k = 100$ | | $k = 200$ | | $k = 500$ | | $k = 750$ | | $k = 1000$ | |
|---|---|---|---|---|---|---|---|---|---|---|---|---|---|---|
| | Time | Cost | Time | Cost | Time | Cost | Time | Cost | Time | Cost | Time | Cost | Time | Cost |
| AFKMC$^2$ | 136 | 8.25 | 186 | 6.44 | 352 | 5.87 | 1.1s | 5.46 | 6.2s | 4.99 | 13.8s | 4.79 | 24.6s | 4.65 |
| PRONECoreset | 374 | 8.50 | 1.6s | 6.83 | 2.9s | 6.00 | 6.0s | 5.74 | 14.5s | 5.17 | 22.4s | 4.90 | 29.7s | 4.70 |
| FastKMeans | 5.3s | **7.48** | 5.1s | **6.15** | 5.4s | 5.74 | 5.8s | 5.33 | 6.4s | 4.95 | 7.1s | 4.76 | 7.7s | 4.63 |
| RejectionSampling | 81.4s | 7.91 | 84.5s | 6.33 | 85.1s | **5.65** | 87.8s | **5.32** | 88.5s | **4.87** | 89.1s | **4.71** | 87.8s | **4.58** |
| Qkmeans | **17** | 7.68 | **32** | 6.30 | **75** | 5.90 | **258** | 5.50 | **1.3s** | 4.98 | **2.5s** | 4.79 | **4.1s** | 4.66 |

*Table 10.* Seeding benchmark on cifar100. Time in ms, cost $\times 10^{11}$.

| Algorithm | $k = 10$ | | $k = 50$ | | $k = 100$ | | $k = 200$ | | $k = 500$ | | $k = 750$ | | $k = 1000$ | |
|---|---|---|---|---|---|---|---|---|---|---|---|---|---|---|
| | Time | Cost | Time | Cost | Time | Cost | Time | Cost | Time | Cost | Time | Cost | Time | Cost |
| AFKMC$^2$ | 135 | 8.27 | 212 | **6.46** | 389 | 6.03 | 1.1s | 5.59 | 6.4s | 5.09 | 13.8s | 4.91 | 24.4s | 4.77 |
| PRONECoreset | 382 | 9.08 | 1.5s | 6.92 | 2.9s | 6.51 | 5.9s | 5.84 | 14.5s | 5.25 | 22.2s | 4.96 | 29.7s | 4.83 |
| FastKMeans | 4.9s | 8.48 | 4.6s | 6.69 | 4.7s | 6.28 | 4.9s | 5.57 | 5.5s | 5.09 | 6.3s | 4.87 | 7.2s | 4.71 |
| RejectionSampling | 85.9s | **8.26** | 85.5s | 6.53 | 87.7s | **5.96** | 86.7s | **5.54** | 87.9s | **5.04** | 90.2s | **4.84** | 86.6s | **4.70** |
| Qkmeans | **16** | 8.63 | **29** | 6.87 | **74** | 6.11 | **262** | 5.68 | **1.4s** | 5.14 | **2.7s** | 4.92 | **4.3s** | 4.78 |

*Table 11.* Seeding benchmark on mnist_clip. Time in ms, cost .

| Algorithm | $k = 10$ | | $k = 50$ | | $k = 100$ | | $k = 200$ | | $k = 500$ | | $k = 750$ | | $k = 1000$ | |
|---|---|---|---|---|---|---|---|---|---|---|---|---|---|---|
| | Time | Cost | Time | Cost | Time | Cost | Time | Cost | Time | Cost | Time | Cost | Time | Cost |
| AFKMC$^2$ | 24 | 2955 | 33 | 2041 | 70 | 1768 | 248 | 1566 | 927 | 1330 | 2.0s | **1239** | 3.8s | 1178 |
| PRONECoreset | 62 | 3131 | 224 | 2162 | 422 | 1830 | 838 | 1629 | 2.3s | 1361 | 3.6s | 1265 | 5.4s | 1191 |
| FastKMeans | 1.3s | 3156 | 1.2s | **2009** | 1.2s | 1781 | 1.2s | **1550** | 1.3s | 1340 | **1.4s** | 1279 | **1.5s** | 1221 |
| RejectionSampling | 5.9s | 3021 | 7.5s | 2055 | 14.4s | 1783 | 49.9s | 1563 | 363.8s | 1340 | 912.2s | 1246 | 1800.0s | 1180 |
| Qkmeans | **12** | **2907** | **23** | 2024 | **61** | **1759** | **190** | **1550** | **828** | **1325** | 1.5s | **1239** | 2.4s | **1175** |

*Table 12.* Seeding benchmark on fmnist_clip. Time in ms, cost .

| Algorithm | $k = 10$ | | $k = 50$ | | $k = 100$ | | $k = 200$ | | $k = 500$ | | $k = 750$ | | $k = 1000$ | |
|---|---|---|---|---|---|---|---|---|---|---|---|---|---|---|
| | Time | Cost | Time | Cost | Time | Cost | Time | Cost | Time | Cost | Time | Cost | Time | Cost |
| AFKMC$^2$ | 23 | 8185 | 32 | 5407 | 57 | 4732 | 153 | 4201 | 839 | 3620 | 2.0s | 3407 | 3.7s | 3262 |
| PRONECoreset | 63 | 8210 | 272 | 5646 | 498 | 5038 | 812 | 4339 | 2.0s | 3689 | 3.6s | 3430 | 4.9s | 3294 |
| FastKMeans | 1.2s | 8128 | 1.2s | 5445 | 1.2s | 4720 | 1.2s | **4143** | 1.3s | 3622 | 1.4s | 3490 | **1.4s** | 3348 |
| RejectionSampling | 6.1s | 8301 | 6.7s | 5437 | 9.3s | 4742 | 22.1s | 4188 | 145.5s | 3616 | 355.1s | 3401 | 685.6s | 3257 |
| Qkmeans | **11** | **7944** | **22** | **5361** | **51** | **4682** | **134** | 4155 | **555** | **3596** | **976** | **3394** | 1.4s | **3249** |

*Table 13.* Seeding benchmark on cifar10_clip. Time in ms, cost $\times 10^2$.

| Algorithm | $k = 10$ | | $k = 50$ | | $k = 100$ | | $k = 200$ | | $k = 500$ | | $k = 750$ | | $k = 1000$ | |
|---|---|---|---|---|---|---|---|---|---|---|---|---|---|---|
| | Time | Cost | Time | Cost | Time | Cost | Time | Cost | Time | Cost | Time | Cost | Time | Cost |
| AFKMC$^2$ | 24 | **187** | 31 | 144 | 72 | **131** | 165 | 121 | 823 | 108 | 1.9s | **104** | 3.7s | 101 |
| PRONECoreset | 59 | 192 | 195 | 149 | 396 | 135 | 720 | 123 | 1.9s | 110 | 3.8s | 104 | 5.0s | 101 |
| FastKMeans | 1.2s | 191 | 1.2s | **144** | 1.2s | 132 | 1.3s | **120** | 1.3s | **108** | 1.4s | 106 | 1.5s | 102 |
| RejectionSampling | 5.9s | 194 | 6.1s | 144 | 7.2s | 132 | 11.4s | 121 | 50.5s | 109 | 117.1s | 104 | 224.0s | **100** |
| Qkmeans | **4.6** | 192 | **12** | 146 | **23** | 132 | **65** | 121 | **240** | 109 | **453** | 104 | **695** | 100 |

*Table 14.* Seeding benchmark on cifar100_clip. Time in ms, cost $\times 10^2$.

| Algorithm | $k = 10$ | | $k = 50$ | | $k = 100$ | | $k = 200$ | | $k = 500$ | | $k = 750$ | | $k = 1000$ | |
|---|---|---|---|---|---|---|---|---|---|---|---|---|---|---|
| | Time | Cost | Time | Cost | Time | Cost | Time | Cost | Time | Cost | Time | Cost | Time | Cost |
| AFKMC$^2$ | 24 | **206** | 32 | 168 | 70 | 153 | 191 | 139 | 906 | **123** | 2.0s | **117** | 3.7s | **112** |
| PRONECoreset | 61 | 218 | 206 | 170 | 455 | 155 | 782 | 142 | 2.0s | 125 | 3.6s | 118 | 4.9s | 114 |
| FastKMeans | 1.2s | 212 | 1.3s | 167 | 1.2s | 152 | 1.2s | **138** | 1.4s | 124 | 1.4s | 120 | 1.5s | 116 |
| RejectionSampling | 6.3s | 206 | 6.3s | 168 | 7.1s | 153 | 10.6s | 139 | 45.4s | 123 | 106.6s | 117 | 208.3s | 113 |
| Qkmeans | **10** | 209 | **15** | **166** | **27** | 151 | **65** | 139 | **252** | 124 | **470** | 117 | **745** | 113 |

*Table 15.* Seeding benchmark on reddit. Time in ms, cost $\times 10^2$.

| Algorithm | $k = 10$ | | $k = 50$ | | $k = 100$ | | $k = 200$ | | $k = 500$ | | $k = 750$ | | $k = 1000$ | |
|---|---|---|---|---|---|---|---|---|---|---|---|---|---|---|
| | Time | Cost | Time | Cost | Time | Cost | Time | Cost | Time | Cost | Time | Cost | Time | Cost |
| AFKMC$^2$ | 29 | 1664 | 40 | 1489 | 69 | **1418** | 165 | **1348** | 687 | 1256 | 1.5s | 1214 | 2.7s | 1183 |
| PRONECoreset | 80 | **1647** | 309 | 1500 | 472 | 1423 | 982 | 1357 | 2.5s | 1263 | 3.8s | 1218 | 5.9s | 1187 |
| FastKMeans | 3.5s | 1659 | 3.4s | 1500 | 2.9s | 1420 | 2.8s | 1350 | 2.9s | 1256 | 3.2s | 1215 | 3.1s | 1207 |
| RejectionSampling | 10.1s | 1662 | 10.1s | **1486** | 10.2s | 1423 | 10.2s | 1352 | 14.0s | **1255** | 20.2s | **1213** | 30.5s | **1182** |
| Qkmeans | **9.4** | 1660 | **11** | 1491 | **15** | 1422 | **27** | 1352 | **85** | 1256 | **146** | 1213 | **235** | 1183 |

*Table 16.* Seeding benchmark on stackexchange. Time in ms, cost $\times 10^2$.

| Algorithm | $k = 10$ | | $k = 50$ | | $k = 100$ | | $k = 200$ | | $k = 500$ | | $k = 750$ | | $k = 1000$ | |
|---|---|---|---|---|---|---|---|---|---|---|---|---|---|---|
| | Time | Cost | Time | Cost | Time | Cost | Time | Cost | Time | Cost | Time | Cost | Time | Cost |
| AFKMC$^2$ | 29 | 1730 | 35 | **1566** | 68 | **1492** | 137 | 1421 | 655 | 1326 | 1.3s | 1282 | 2.5s | **1251** |
| PRONECoreset | 76 | **1722** | 250 | 1572 | 504 | 1497 | 900 | 1427 | 2.2s | 1328 | 4.0s | 1286 | 5.6s | 1253 |
| FastKMeans | 2.0s | 1728 | 2.1s | 1571 | 2.0s | 1494 | 2.0s | 1421 | 2.1s | 1326 | 2.1s | 1284 | 2.3s | 1277 |
| RejectionSampling | 10.1s | 1724 | 10.0s | 1566 | 10.1s | 1492 | 10.3s | **1420** | 13.6s | **1325** | 19.9s | **1282** | 28.7s | 1251 |
| Qkmeans | **6.5** | 1733 | **6.5** | 1568 | **14** | 1495 | **23** | 1423 | **80** | 1328 | **146** | 1283 | **220** | 1251 |

*Table 17.* Seeding benchmark on har. Time in ms, cost $\times 10^2$.

| Algorithm | $k = 10$ | | $k = 50$ | | $k = 100$ | | $k = 200$ | | $k = 500$ | | $k = 750$ | | $k = 1000$ | |
|---|---|---|---|---|---|---|---|---|---|---|---|---|---|---|
| | Time | Cost | Time | Cost | Time | Cost | Time | Cost | Time | Cost | Time | Cost | Time | Cost |
| AFKMC$^2$ | **2.7** | **2230** | **13** | **1587** | **40** | **1401** | 148 | **1230** | 931 | **1007** | 2.2s | **907** | 4.2s | **829** |
| PRONECoreset | 8.3 | 2408 | 29 | 1674 | 57 | 1440 | 147 | 1273 | 449 | 1049 | 617 | 949 | 905 | 880 |
| FastKMeans | 140 | 2305 | 138 | 1623 | 201 | 1483 | 147 | 1312 | **170** | 1094 | **313** | 996 | **247** | 917 |
| RejectionSampling | 913 | 2244 | 873 | 1596 | 895 | 1415 | 1.0s | 1236 | 2.0s | 1016 | 3.6s | 911 | 6.1s | 832 |
| Qkmeans | 7.5 | 2235 | 14 | 1594 | 42 | 1421 | **130** | 1245 | 595 | 1018 | 1.0s | 916 | 1.6s | 841 |

*Table 18.* Seeding benchmark on susy. Time in ms, cost $\times 10^6$.

| Algorithm | $k = 10$ | | $k = 50$ | | $k = 100$ | | $k = 200$ | | $k = 500$ | | $k = 750$ | | $k = 1000$ | |
|---|---|---|---|---|---|---|---|---|---|---|---|---|---|---|
| | Time | Cost | Time | Cost | Time | Cost | Time | Cost | Time | Cost | Time | Cost | Time | Cost |
| AFKMC$^2$ | 2.8 | 1.01 | **4.8** | 0.62 | **8.1** | 0.50 | **17** | **0.41** | **72** | 0.32 | 187 | **0.28** | 302 | 0.26 |
| PRONECoreset | 20 | 1.17 | 45 | 0.67 | 99 | 0.54 | 152 | 0.44 | 288 | 0.33 | 404 | 0.29 | 521 | 0.27 |
| FastKMeans | 1.1s | 1.01 | 1.1s | 0.63 | 1.1s | 0.51 | 1.2s | 0.43 | 1.3s | 0.33 | 1.4s | 0.29 | 1.5s | 0.29 |
| RejectionSampling | 1.5s | **0.99** | 1.5s | **0.61** | 1.5s | **0.50** | 1.6s | 0.41 | 2.0s | 0.32 | 2.9s | 0.28 | 4.3s | 0.26 |
| Qkmeans | **2.4** | 1.06 | 7.1 | 0.62 | 17 | 0.50 | 52 | 0.41 | 264 | 0.32 | 437 | 0.28 | 518 | **0.26** |

