# OpenReview forum: "Fast k-means Seeding Under The Manifold Hypothesis"
_ICML.cc/2026/Conference — ICML 2026 regular_

### Official Review · Reviewer_cgbx · 2026-03-08

**Soundness:** 3
**Presentation:** 2
**Significance:** 3
**Originality:** 3
**Overall Recommendation:** 4
**Confidence:** 4

**Summary:**

The $k$-means problem asks for a set of $k$ centers $C$ minimizing the sum of squared (typically Euclidean) distances of input points $X$ to their respectively nearest center in $C$. A popular heuristic is the $k$-means++ algorithm that iteratively constructs $C$ by sampling a point proportionate to its cost, until $k$ centers have been sampled. This algorithm achieves a $O(\log k)$ approximation on expectation, however the running time is $O(nDk)$, where $n$ is the number of data points, $D$ the dimension. Here the authors achieve a linear time algorithm and retain the multiplicative $\log k$ approximation and add an additive error that scales with the $1$-means cost. The additive error may be further controlled at the cost of an increase in running time or by assuming that the data lies on a low dimensional manifold. In a sense (see comments below), I believe that this paper can be considered an improvement of the running time bounds by Shah and Jaiswal by adding NNS data structures to improve dependencies on $k$.

**Compliance With Llm Reviewing Policy:**

Affirmed.

**Final Justification:**

The sampling algorithm is nice, the analysis a bit sloppy but ultimately correct, the learning results are perfunctory (and worse than the SOTA, which the authors could do a better job of surveying). Since there exist many papers with relatively similar seeding strategies, I believe that this paper comes in at weak accept.

**Key Questions For Authors:**

Please address the issue with the bounds in Theorem 5 mentioned above. If the running times depend on $\frac{opt_1(X)}{opt_k(X)}$, or indeed $\Delta$, then they should be stated in this way. My weak accept is conditioned on this being possible to resolve. I would also like to see clearly stated bounds in the abstract and the theorem. The bounds currently are hard to make sense of and in my opinion, $\Delta^2$ is the fairest comparison to previous work, which often would have also achieved a dependence in terms of $\frac{opt_1(X)}{opt_k(X)}$ if they cared to write it that way.

Also, is it clear that the LSH-type bounds work against an adaptive adversary (i.e. the algorithm), or do they require an oblivious adversary? This might also be an issue with previous work. The tree embedding part works, but the LSH part is unclear to me.

**Strengths And Weaknesses:**

Strengths:
- This is a solid result that fits nicely with a long line of work aimed at obtaining faster algorithms for $k$-means.
- Quantum inspired techniques are a nice addition to the toolbox of algorithm designers working in clustering, and I appreciate that this paper is introducing these techniques.
- Experiments are impressive.

Weaknesses:
- The writing could be better. A few references are broken and the results are very hard to parse (see below).
- Some of the statements are inaccurate. For example, $\eta(X , C)$ is claimed to be smaller than $\eta(X)$. This is only true if $C$ is drawn from the input.
- Other times, terms are not defined. For example the "Query" subroutine is not defined in the main body.
- The learning bounds seem a bit divorced from the rest of the paper and not particularly novel. I am not sure why they were included. The assumptions are stronger than I'd like and results not particularly impressive. In any case, the analysis could be tighter. For example, a rate of the order $\sqrt{1/n}$ is possible using chaining, instead of the $\sqrt{\log n / n}$ rate given by the authors. Also, no dependence on $D$ is necessary, if $D$ is assumed to be large. In fact, I think one could obtain a $\sqrt{k d /n}$ bound. Previous work already gave a $\sqrt{k \log k / n}$ rate. As such, the given convergence bounds are weaker than what is known in literature.
- If the authors wish to use a JL-embedding in a preprocessing step, then the running time is no longer $O(nD)$. I am also not sure where this is used, so I don't know why it was included in the first place.
- I am still not sure how to make sense of a term $\mathbb{E}[\text{cost}(X,S)] = ...k^{-\text{cost}(X,S)} ...$, which appears in the expected cost of the solution given in Theorem 5. From reading the proof, the authors seem to bound $\text{cost}(X,S)$ on the right hand side in terms of the expected cost of the previous iteration. The easiest way around this is to get a lower bound on $\text{cost}(X,S)$ in terms of the optimal solution. Unfortunately, this may also mean that the cost is not rapidly decreasing and in Corollary 6, I am not sure what the claimed running time is supposed to be. In the worst case, this seems to be of the order $\frac{opt_1(X)}{opt_k(X)}$, which can be as large as $n\Delta^2$, though this dependency can be improved to $\Delta^2$ with more care and I do not see how this can be avoided. In addition, from the sketch of the proof, the running time is on expectation, and the RHS still contains a random variable. In any case, this should be clarified and I believe more readable bounds that minimize the number of parameters are preferable over what is currently written.

---

> ### Author Rebuttal · Authors · 2026-03-31
>
> We thank the reviewer for their constructive comments and hope that the following addresses the concerns raised.
>
> > Some of the statements are inaccurate . . .
>
> We agree that this is inaccurate in general, and the oversight happened since the context of our discussion was k-means++ seeding, where the centres are chosen from the data points. We will correct this.
>
> > Other times, terms are not defined . . .
>
> We will include a short section that defines the LSH operations in the main paper instead of the appendix.
>
> > The learning bounds seem a bit divorced from the rest of the paper . . .
>
> As we mention in Appendix A lines 667-669, the techniques
> used here are well developed in optimum quantization theory (Zador, 1982; Gray & Neuhoff, 1998; Graf & Luschgy, 2000; Gruber, 2004) and we provide an elementary proof for the sake of being self-contained.
>
> The main reason for including this was to transfer the scaling laws from the population quantization error to the sample quantization error. We will be happy to include citations and references to the improved bounds if suggested by the reviewer.
>
> Regarding improving the dependence from $\sqrt{kD \log n / n}$ to $\sqrt{kd \log n / n}$, we think that it is a great idea and can be shown formally as well. We provide a brief sketch here.  It is well known that the optimal cost when the centers are restricted to be chosen from the dataset is atmost a factor of 2 larger from the optimal cost in the unrestricted setting. The factor of $kD$ comes from the covering number argument (Lemma 18&19 in appendix A). If the centres are restricted to lie on the manifold, this can be improved to $kd$ from which the improved bounds follow, losing a multiplicative factor of 2.  For a smooth $d$-dimensional manifold, the covering number of the space of centers scales with the intrinsic dimension $d$ rather than $D$.
>
> > If the authors wish to use a JL-embedding in a preprocessing step . . .
>
> We agree with the reviewer on our oversight that the runtime should be mentioned as $O(nd \log k)$. Since JL-type dimensionality reduction is well known for k-means (Makarychev et al, 2019), we will add it as a remark that these techniques can be applied so that the dependence of $D$ can be replaced by $O(\log k)$ after the dimensionality reduction step.
>
> > I am still not sure how to make sense of a term . . .
>
> We will make sure to make our main results more clear and precise. We agree with the reviewer that the way that it is written currently is a bit confusing.
>
> Suppose the first center which is chosen is $c_1$ and that $\delta$ is an upper bound on the failure probability of the Rejection sampling procedure. We show that $\delta \leq e^{-m / \tau}$ where $\tau \leq 2 \rho^{-1} (\|X\|_F^2 + n\|c_1\|^2) / opt_k$. Taking expectations yields $E[\tau] \leq 4\rho^{-1} opt_1/opt_k$. We can avoid using expectation by conditioning on the fact that $\|c_1\|^2 \leq 8 \|X\|_F^2$ which happens with prob. atleast $3/4$ (using Markov's inequality). This gives $\tau \leq 18 \rho^{-1} opt_1/opt_k$, independent of the centers $C$ chosen by the algorithm. This conditioning can be removed by repeating the procedure $c\log (opt_1/opt_k)$ times and then taking the best clustering, so that there is negligible (atmost $1+(1/4)^c = 1+o(1)$) factor loss in the approximation guarantee, using Section 2, Theorem 15 .
>
>
> > In the worst case, this seems to be of the order  . . .
>
> We agree with the reviewer that in the worst case, the parameter $\beta$ could be as large as $\Delta^2$. However, we feel that this misses the point of the paper: the upper bound is too pessimistic for relevant instances, and much tighter bounds can be shown under beyond-worst-case assumptions, which we take to be the manifold hypothesis. We also refer to our experimental study, which shows this to be the case for a variety of datasets across research domains.
>
> We would be happy to include this point in the abstract as well as the main theorem by explicitly writing $opt_k/opt_1$ and then arguing why this quantity scales well.
>
> **Please see our response to hGft for further discussion on how to estimate these parameters**.
>
>
> > Also, is it clear that the LSH-type bounds work against an adaptive adversary  . . .
>
> Corollary D.6 of previous work [1] shows that once the LSH initialisation is successful (which happens with high probability), each insert and query operation works correctly.
>
> **Please see our response to zJDn for further discussion on LSH issues**, where we show how the initialisation can be repeated so as to have a negligible effect on the expected clustering cost output by our method.
>
> We also clarify that doing this repetition of sampling hash functions is necessary for each dataset, since the same choice of hash functions may lead to the data structure not being successful for another dataset.
>
> [1] Cohen-Addad et al., Fast and Accurate k-means++ via Rejection Sampling, NeurIPS 2020.

---

> > ### Author Rebuttal · Reviewer_cgbx · 2026-03-31
> >
> > Thank you. I will maintain my score.

---

> > > ### Author Response · Authors · 2026-04-06
> > >
> > > We are glad that we were able to clarify all questions during the discussion phase.

---

### Official Review · Reviewer_hGft · 2026-03-11

**Soundness:** 3
**Presentation:** 3
**Significance:** 3
**Originality:** 3
**Overall Recommendation:** 4
**Confidence:** 4

**Summary:**

This paper studies the classic k-means problem under a beyond-worst-case analysis (with Manifold Hypothesis), aiming to design faster k-means seeding algorithms. An important contribution for this paper is the link between low intrinsic dimension, quantization-style scaling laws, and improved runtime-quality tradeoffs for k-means++ seeding. More specific, this paper proposes to model clustering data as samples from a density on a low-dimensional smooth manifold, and then uses the properties from manifold to design fast k-means seeding methods by combining with a rejection-sampling-based scheme to retain approximation guarantees. The paper also includes a empirical study across image, embedding, text, and tabular datasets.

**Compliance With Llm Reviewing Policy:**

Affirmed.

**Final Justification:**

After considering both the paper and the rebuttal, I maintain my weak accept recommendation. The paper is technically solid, well motivated, and offers an interesting connection between manifold-based scaling behavior and faster k-means seeding, while the rebuttal adequately addressed most of my concerns by clarifying the role of the manifold hypothesis, the interpretation of the theoretical quantities, and the empirical methodology. Although I still think some of these explanations should have been stated more clearly in the original submission, the rebuttal strengthened my confidence in the work rather than changing my overall evaluation.

**Key Questions For Authors:**

1. Theorem 5 depends on $\beta(X,C)$ and $\eta(X,C)$, where $C$ is the returned solution. Can the authors restate the guarantee using quantities observable before execution, or prove upper bounds on these quantities under transparent assumptions?

2. Since Theorem 5 already holds for arbitrary datasets, what is the precise algorithmic role of the manifold hypothesis beyond justifying favorable scaling of the data-dependent parameters?

3. The experiments fit scaling exponents using heuristic $k$-means solutions. How sensitive are the estimated exponents to initialization, number of Lloyd restarts, or the range of $k$ values used in the regression?

**Limitations:**

yes

**Strengths And Weaknesses:**

The strengths can be summarized as follows.

1. The problem studied in this paper is classic and well motivated. Rather than assuming separability or stability properties of the optimal clustering, the paper argues for using the manifold hypothesis as a data-side structural assumption, which is somewhat interesting for modern ML datasets. It combines quantization-theoretic scaling laws with a rejection-sampling rule of $D^2$-sampling, and then further accelerates sampling with approximate nearest-neighbor search plus a rejection loop.

2. The empirical study is reasonably broad. The study covers raw images, CLIP embeddings, text embeddings, and tabular/sensor datasets, and the paper reports both scaling-law behavior and seeding-runtime comparisons against multiple baselines.

The main weakness can be summarized as follows.

1. My main concern is that the theoretical guarantees are somewhat hard to interpret explicitly. In Theorem 5 and Corollary 6 and Corollary 7, the runtime depends on $\rho, m, \beta, opt_1(\mathcal{X})$ and $\epsilon$. Especially, in Corollary 7, $\gamma$ is also related to $\epsilon$ and $\rho$. This makes the guarantee feel partly self-referential: the bound is stated using quantities that are not known before the algorithm finishes, which weakens its practical interpretability.

2. The role of the manifold hypothesis feels somewhat indirect. The main seeding theorem is stated for arbitrary datasets, and the manifold assumption is mainly used afterward to argue that the data-dependent parameters scale favorably.

3. The empirical validation of the scaling laws is not fully aligned with the theoretical objects. The experiments estimate scaling by running standard k-means++ plus Lloyd iterations for a finite grid of k values, then fitting regressions in log-log space. This is suggestive, but it does not directly validate the optimal quantities in the theory, and it leaves open how robust these fitted exponents are to initialization, local minima, or different k-ranges.

---

> ### Author Rebuttal · Authors · 2026-03-31
>
> We thank the reviewer hGft for the insightful comments, which have helped us improve both the clarity and positioning of our contributions.
>
> > My main concern is that the theoretical guarantees are somewhat hard to interpret explicitly . . .Can the authors restate the guarantee using quantities observable before execution, or . . .
>
> $\beta(X,C),\eta(X,C)$ can be upper bounded by the center independent quantitites $\beta(X),\eta(X)$. Please see Lines 114-130. To estimate them, we outline three complementary approaches:
>
> 1)
> A conservative approach is to upper bound the parameters using the dataset aspect ratio $\Delta$, which can be approximated in $O(n \log n)$ time [1]. This yields: $\eta \leq \Delta, \quad \beta \leq \Delta^2$
> providing fully observable, worst-case bounds that hold for any dataset.
>
> 2)
> Under the manifold hypothesis, the parameters exhibit predictable scaling behavior. In particular, we estimate the intrinsic dimension $d$ using the maximum likelihood estimator of Levina and Bickel [2], and obtain an estimate of the quantization exponent $\epsilon = 2/d$. This yields: $\beta_k \lesssim k^{2/d}$
> which provides significantly tighter and more informative bounds than worst-case estimates. As shown in Section 3 (lines 415–432), these estimates are stable across datasets.
>
> 3)
> A practical alternative is to compute k-means costs for a small set of values $k \in O(1)$ , fit the scaling laws, and extrapolate to larger $k$. Our experiments show that the resulting fits are highly consistent (typically $R^2 \geq 0.95$), indicating that these estimates provide reliable upper bounds in practice.
>
> 4)
> Using this estimate of $\beta$, we can set $m = O(\beta)$ to ensure negligible effect from the additive term.
>
> We will revise the paper to explicitly include these bounds and estimators to improve clarity.
>
> Please also refer to our reply to Reviewer cgbx.
>
>
> > The role of the manifold hypothesis feels somewhat indirect . . . what is the precise algorithmic role of the manifold hypothesis  . . .
>
> As pointed out, at a high level, our approach separates algorithm design from structural analysis: Theorem 5 provides guarantees for arbitrary datasets in terms of natural data-dependent quantities, while the manifold hypothesis is used to make these quantities predictable and well-behaved.
>
> More concretely, the manifold hypothesis plays a  role in two key ways:
>
> 1. Controlling the efficiency of rejection sampling.
> The runtime of our seeding procedure depends on the Rényi divergence between the proposal and target distributions. Under the manifold hypothesis, we show (Lemma 13) that this divergence scales as $O(\log k / d)$. Without this structure, it is not clear that such a bound holds.
>
> 2. Converting data-dependent guarantees into explicit scaling laws.
> Theorem 5 depends on $\beta$ and $\eta$, which are instance-specific. The manifold hypothesis implies that these quantities follow predictable scaling laws (Theorem 4), e.g., $\beta \sim k^{2/d}$.This allows us to derive Corollary 7, where both runtime and approximation guarantees are expressed explicitly in terms of $k$ and intrinsic dimension
> $d$.
>
> More broadly, we view this as part of a general paradigm: algorithms may admit data-dependent guarantees in full generality, while assumptions such as the manifold hypothesis enable these guarantees to become quantitatively predictive.
>
>
>
> > The empirical validation of the scaling laws is not fully aligned with the theoretical objects . . . How sensitive are the estimated exponents to initialization . . .
>
> We thank the reviewer for this important observation. We agree that our experiments use k-means++ + Lloyd as a proxy for the optimal quantities $\mathrm{opt}_k(X)$, which are intractable to compute exactly. This is standard practice in empirical studies of k-means.
>
> Importantly, k-means++ provides an $O(\log k)$-approximation, i.e., $\mathrm{cost}_{k\text{-means++}}(X) \leq O(\log k)\,\mathrm{opt}_k(X)$
> which implies that the empirical quantities yield valid **upper bounds** on the theoretical ones. Thus, the estimated is also an upper bound up to an $O(\log k)$ factor, preserving the scaling exponent.
>
> Our goal is to validate the *scaling behavior*, not exact optimal values. We observe that the fitted exponents are stable across runs (10 restarts) with consistently high goodness-of-fit ($R^2 \geq 0.95$). Increasing the number of Lloyd restarts improves absolute cost but has minimal effect on scaling exponents (as ratios cancel multiplicative factors). We also use a broad k-range [5-1000]and will include additional analysis showing robustness across different k-ranges.
>
> We will clarify the role of empirical proxies and include these robustness checks in the final version.
>
> [1] Michael Ian Shamos and Dan Hoey. Closest-point problems. FOCS'1975.
>
> [2] Elizaveta Levina and Peter J. Bickel. 2004. Maximum Likelihood estimation of intrinsic dimension. NIPS'04.

---

> > ### Author Rebuttal · Reviewer_hGft · 2026-04-01
> >
> > Thank you for the thoughtful response. The rebuttal addresses most of my concerns. However, the original submission did not provide such detailed discussion of the theoretical guarantees, and for this reason I would prefer to maintain my initial evaluation at this stage.

---

> > > ### Author Response · Authors · 2026-04-06
> > >
> > > We are glad that we were able to clarify all questions during the discussion phase.
> > >
> > > We assure the reviewer that we will add the discussions and clarifications which happened during the rebuttal to the camera ready version of our submission.

---

### Official Review · Reviewer_zJDn · 2026-03-12

**Soundness:** 3
**Presentation:** 2
**Significance:** 3
**Originality:** 2
**Overall Recommendation:** 4
**Confidence:** 3

**Summary:**

This paper studies the problem of $k$-means clustering under the manifold hypothesis. They introduce a new algorithm, QKmeans, which exhibits competitive guarantees against SOTA methodology. In particular, this method is on the faster end, while maintaining nearly optimal clustering guarantees. They emphasize the geometric placement of their algorithm, with guarantees depending on the intrinsic data dimensionality.

**Compliance With Llm Reviewing Policy:**

Affirmed.

**Final Justification:**

The proposed method combines ideas present in the literature for a fast and effective k-means clustering procedure. It is my impression at this time that the paper would be better structured by emphasizing the specifics of its contribution and the papers that motivate these ideas in the introductory sections of the paper. Presently, the writing does not quickly get to this point, and instead has many distracting superficial details. My opinion is that the strong manifold hypothesis assumption is not a driving factor for the results presented, and is more accurately a special case of the possible applications of interest. I do however think there is merit in the proposed methodology.

**Key Questions For Authors:**

1. Would accounting for LSH failure significantly change your bounds?

2. When the manifold hypothesis is violated, and data is fully high dimensional, how do your guarantees compare to other seeding algorithms throughout the literature? My impression is that many of the proposed guarantees would still be intact and perhaps SOTA in some contexts, which again makes me question the heavy dependence on the manifold assumption in the presentation.

3. Similarly, what about situations where data only concentrates near a manifold? Can you recover a finite sample bound in terms of something like a covering number, so that in appropriate data regimes the guarantees scale intrinsically/extrinsically?

**Limitations:**

Yes

**Strengths And Weaknesses:**

I think there are many strengths and insights of this paper, but they are ultimately bogged down by an unclear presentation which obfuscates the placement of the method in the literature. Much time is spent in the introduction introducing and justifying basic notions in $k$-means clustering and manifold geometry, which are already standard in the literature/related literatures. Further, the claim that it is a novelty of this paper to introduce the manifold hypothesis is ill-placed, with many related works capturing a similar intrinsic dimensional dependence, see https://arxiv.org/pdf/1707.00087 for example.

I believe that much of the paper should be revised to match publication quality, with the most substantial revision being the overall flow and presentation of the proposed methodology. The QKmeans algorithm combines ideas from several recent $k$-means papers in a fairly transparent way, primarily leveraging approximate nearest neighbor searches, and a dequantized rejection sampling algorithm. While these are mentioned in the text, I believe they could be more clearly attributed earlier in the presentation along with a corresponding pictorial chart or so depicting these components. This would better help the reader place what the exact contribution of this paper is (leveraging these ideas in tandem) from the beginning of the presentation. Along these lines, some theoretical ideas of the paper are slightly sloppy. For one, it is assumed that the dequantized "LSH-based ANNS data structure" is succesful, despite being utilized repeatedly throughout the algorithm, opening up a large probabilistic chance of failure in the proposed bounds. Secondly, it is mentioned that other methods have a negative dependence on the data aspect ratio, however Lemma 23 demonstrates a similar vulnerability to QKmeans.

The empirical results are encouraging, and it may indeed be the case that the algorithmic combination presented in the paper is worthy of some celebration. My overall opinion however is that the paper requires further revision in the framing of the methodology, its overall presentation, as well as some minor theoretical issues. It is not helped that there are a great deal of competing $k$-means algorithms, listed and unlisted, which often exhibit very similar guarantees and this further adds to the confusion. I am flexible to improving my evaluation if I have significantly overlooked some aspect of this paper.

---

> ### Author Rebuttal · Authors · 2026-03-31
>
> We would like to thank reviewer zJDn for pointing out various issues, and addressing those will surely improve the clarity of our paper.
>
> > “Much time is spent in the introduction introducing and justifying basic notions…”:
>
> We agree with the reviewer that the introduction does cover a lot of background, but we decided to include it so that our work may be more accessible to the broader ML community by describing the need of beyond-worst-case algorithm analysis. We would be happy to incorporate any particular suggestions that the reviewer may have regarding the presentation.
>
> > “...claim that it is a novelty of this paper to introduce the manifold hypothesis…”:
>
> We are NOT claiming novelty for introducing the manifold hypothesis (indeed, it has been widely studied and established by now). We also clearly attribute the origin of the scaling laws to the quantisation theory community (Zador’s and Gruber’s seminal works, Section 1.1, Fact 2).
> Rather, our contribution is to identify the geometric structure it imposes and to use it to design a simple sampling-based algorithm. Moreover, we perform an empirical study to verify these assumptions. We would ensure we cite additional relevant work (as suggested by the reviewer) to clarify this point in the introduction. We believe that using this background from a beyond-worst-case analysis perspective and the QKmeans algorithm for k-means is novel.
>
> > “ …more clearly attributed earlier in the presentation along with a corresponding pictorial chart…”:
>
> We will incorporate the suggestion of attributing the prior work along with a diagrammatic illustration more upfront in the introduction section.
>
> > “Would accounting for LSH failure significantly change your bounds?”:
>
> Given the dataset, the hash tables which constitute the LSH data structure are chosen in the beginning in such a manner that with probability $1 - 1/n$, they follow the desirable property (farther pairs of points map to distinct hashes, closer points map to same hash). See Corollary D.6 of previous work [1]. Assuming this property successfully holds for all $\binom{n}{2}$ points in the dataset, the Insert and Query operations will always be successful. So the failure probability of $1/n$ is valid only during the initialization phase; henceforth we can work with the assumption that all operations are successful with probability $1-1/n$.
> Also, see the footnote on Page 6 of previous work [1]. Suppose we repeat our LSH initialization followed by the seeding procedure $M$ times and take the best clustering. Then with probability atleast $1 - p = 1 - 1/n^M$, one of the intializations would be successful. In that case, the expected cost is bounded by $O(\log k) opt_k$. In the case of failure, since we choose the first center uniformly at random, the expected cost is bounded by $2 opt_1$. Hence, the total expected cost can be bounded as :
> $\mathbb{E}[ALG] \leq (1-p)\cdot O(\log k) opt_k + 2p\cdot opt_1 \leq O(\log k)\cdot opt_k (1 + 2p \beta_k / \log k) $. Choosing $M = \log_n(2n \beta_k)$ shows that $\mathbb{E}[ALG] \leq O(\log k) opt_k (1 + O(1/n))$. Hence, the effect of LSH failure has negligible impact on the output quality of our method.
>
> >  “other methods have a negative dependence on the data aspect ratio … similar vulnerability to QKmeans”:
>
> We agree that our methods also have a dependence on the aspect ratio, but we only require a dependence on the aspect ratio of the cluster centres (rather than that of the complete dataset), As shown in Section 3, Figure 1(b), this quantity scales favourably in our experiments.
>
> > “...situations where data only concentrates near a manifold?”:
>
> We agree with the reviewer that formally extending the scaling laws to more general settings is a very exciting future direction, however there are some non trivial barriers to doing so, and we refer them to our reply to reviewer KD6v for further discussion.
>
> > “...heavy dependence on the manifold assumption in the presentation”:
>
> The role of the manifold hypothesis is more towards justifying the existence of the proposed scaling laws. Since it is very established in the ML community, relying on it makes our assumptions backed by prior theoretical and empirical research.
> We believe that the novelty and relevance of our works lies in leveraging the manifold structure to show that even though some algorithms have bad dependency on data-dependent parameters in the worst case, for realistic data, they scale favourably and give way to very simple and efficient implementations. We hope that our initial work may lead to further investigation in exploiting such structures in the ML-algorithms community.
>
> [1] Cohen-Addad et al., Fast and Accurate k-means++ via Rejection Sampling, NeurIPS 2020.

---

> > ### Author Rebuttal · Reviewer_zJDn · 2026-04-01
> >
> > Thank you for the thoughtful responses to my questions. My impression is that the proposed methodology is indeed novel and is strong enough that I will revise my score to a 4. My overall concerns regarding the quality of presentation are more difficult to address. In particular, I still think the introductory pages need large revision, shifting more emphasis toward comparison with the literature and methodological advances, as opposed to justifying auxiliary notions such as the manifold hypothesis. It is my opinion that the results should be stated in such a way so that the intrinsic dimensional dependence is a corollary rather than a central focus of the paper.

---

> > > ### Author Response · Authors · 2026-04-06
> > >
> > > Thank you for raising the score from 3 to 4.
> > >
> > > Our thinking while writing the paper was that the manifold hypothesis and scaling laws provide a strong theoretical basis for beyond-worst-case guarantees and offer a compelling justification for our methods. That is why we chose to devote space to this material. However, your point that the paper should place greater emphasis on comparisons with other methods is well taken. Making the discussion of dimensionality reduction more concise by omitting some details should not compromise clarity or flow and may help highlight our key contributions. We are willing to make these adjustments in the presentation style in the final version.
> > >
> > > We have attempted to add comparisons with relevant prior work, both theoretical and empirical, in Section 1.2. If we have inadvertently missed any important reference, please let us know.

---

### Official Review · Reviewer_KD6v · 2026-03-12

**Soundness:** 3
**Presentation:** 3
**Significance:** 3
**Originality:** 3
**Overall Recommendation:** 4
**Confidence:** 4

**Summary:**

Traditional analysis of $k$-means algorithms often focuses on worst-case scenarios, creating a significant gap between theoretical bounds and empirical performance. Since such pathological instances are rare in real-world datasets, it is more effective to model input data under reasonable assumptions. The authors propose the manifold hypothesis that high-dimensional data actually concentrates around low-dimensional smooth manifolds.

Leveraging this assumption, the authors introduce Qkmeans, a fast seeding algorithm designed to bridge this gap. Qkmeans provides strong theoretical guarantees and significant practical speedups over standard methods like $k$-means++. A key contribution of the paper is the derivation of a scaling law induced by the manifold hypothesis. Specifically, the authors demonstrate that when data adheres to this structure, the ratio of the optimal k-cluster cost to the 1-cluster cost follows a specific proportionality:$\frac{\mathtt{opt}_k(X)}{\mathtt{opt}_1(X)} \propto k^{-2/d}.$

Unlike standard $k$-means++, which requires traversing all n points during initialization, Qkmeans only computes distances for a small set of candidates at each step. It utilizes an accept/reject filter to efficiently mimic the k-means++ distribution. Because the data is clustered near a manifold, this filtering process becomes exceptionally fast, allowing for high-quality seeding without the linear scan overhead. The authors perform abundant experiments to validate the theoretical corollary and the algorithm’s performance.

**Compliance With Llm Reviewing Policy:**

Affirmed.

**Final Justification:**

After considering the rebuttal and the other reviewers' comments, I have decided to maintain my original rating. The authors have successfully clarified the connection between their assumptions and the proposed algorithms. However, a minor gap between the theory and practical implementation remains. Consequently, I recommend a weak acceptance.

**Key Questions For Authors:**

Does the scaling law still hold if the manifold hypothesis is weakened? I doubt that real-world data satisfies such a restrictive condition. However, since the experimental results demonstrate the law's validity, I suspect the scaling law persists even under this more relaxed assumption.

**Limitations:**

no potential negative societal impact

**Strengths And Weaknesses:**

### Strengths

- The presented manifold hypothesis is more natural than the assumptions in previous works, which typically relied on specific properties of the optimal solution. Furthermore, the authors verify its validity through an empirical study.
- The writing of this paper is very clear. The algorithm is actually very simple and easy to implement.

### Weaknesses

- The manifold hypothesis requires the probability of a data point not lying exactly on the manifold to be zero. This seems overly restrictive. A more reasonable, albeit weaker, assumption is that data points lie near the submanifold rather than strictly upon it.

---

> ### Author Rebuttal · Authors · 2026-03-31
>
> We thank the reviewer for the insightful comments, which have helped us improve both the clarity and positioning of our contributions.
>
> > “Does the scaling law still hold if the manifold hypothesis is weakened?...”:
>
> We agree with the reviewer that a real dataset need not lie exactly on a smooth manifold. Indeed, this is the reason for including our extensive empirical study to verify whether the scaling laws hold true for typical datasets across domains.
>
> In Appendix D.3 , Figure 10, we study the effect of isotropic ambient noise on the scaling laws, and our experiments suggest that there is a graceful transition from the intrinsic dimension (d)  to ambient dimension (D) as the noise-to-signal ratio grows.
>
> However, we were able to formally prove the scaling laws only in the exact manifold case. This is because current quantization theorems do not directly allow for such a transition, as the noise makes the support of the distribution D-dimensional. Extending this to a more general setting is a promising future direction.
>
> Moreover, the manifold hypothesis acts as a justification for the scaling laws, and our focus was more on how to utilise this particular geometric structure of the data to design beyond worst-case algorithms for clustering.

---

> > ### Author Rebuttal · Reviewer_KD6v · 2026-04-03
> >
> > While proving a stronger theorem remains challenging, empirical results validate the underlying assumption. The current theorem serves to explain the phenomenon rather than provide a rigorous guarantee.

---

> > > ### Author Response · Authors · 2026-04-06
> > >
> > > Since the acknowledgments tags (a), (b), (c) are new to us in the review process, we want to make sure that we are on the same page with the reviewer regarding the acknowledgment choice "(b) Partially resolved - I have follow-up questions for the authors," and that it does not cause any confusion during the discussion phase. As discussed above, the follow-up question regarding the weakening of the Manifold Hypothesis is part of our future research and not one we can answer at this point.

---

### Decision · Program_Chairs · 2026-04-30

**Decision:**

Accept (regular)

**Comment:**

The reviewers universally recognise the theoretical contributions of the paper being valuable inclusions in the literature, and that the empirical validation of the results/assumptions underlying the results are sufficiently extensive and thorough. Perhaps the main weakness of the initial submission relates to clarity in some of the technical constructs relevant to the statements of the theory. There are also some concerns over the overall structure, with arguably too much space dedicated to somewhat basic background; and also the language usage i.t.o. what is actually novel. Perhaps it is unlucky for the authors that these sorts of concerns are the hardest to satisfy in a rebuttal, but I believe they have done a reasonable job in doing so. I am recommending acceptance, and should the paper be accepted I would suggest the authors consider the following two points in preparing their final camera ready submission:
- Use simpler sentences when discussing the contributions of the paper. Understandably the scientific community is precious about intellectual ownership of ideas and results. In reading the paper myself I can see, for example, why it could be misconstrued that you are introducing the manifold hypothesis itself, especially by readers whose first language is not English. This is not meant to be a critical comment; only that I can understand some of the comments from the reveiwers in this regard. This is especially true of Contribution 1. Has not the manifold hypothesis previously be empirically validated across real world clustering instances?
- In terms of how much basic background to include; ultimately this is up to the authors, and I appreciate their argument for how the paper was structured. However, perhaps the short-form conference proceedings papers are not the place for people who don't know what kmeans is.